# Plasma proteomic associations with genetics and health in the UK Biobank

Benjamin B. Sun[1✉], Joshua Chiou[2,26], Matthew Traylor[3,26], Christian Benner[4,26], Yi-Hsiang Hsu[5,26], Tom G. Richardson[3,6,26], Praveen Surendran[6,26], Anubha Mahajan[4,26], Chloe Robins[7,26], Steven G. Vasquez-Grinnell[8,26], Liping Hou[9,26], Erika M. Kvikstad[8,26], Oliver S. Burren[10], Jonathan Davitte[7], Kyle L. Ferber[11], Christopher E. Gillies[12], Åsa K. Hedman[13], Sile Hu[3], Tinchi Lin[14], Rajesh Mikkilineni[15], Rion K. Pendergrass[4], Corran Pickering[16], Bram Prins[10], Denis Baird[1], Chia-Yen Chen[1], Lucas D. Ward[17], Aimee M. Deaton[17], Samantha Welsh[16], Carissa M. Willis[17], Nick Lehner[18], Matthias Arnold[18,19], Maria A. Wörheide[18], Karsten Suhre[20], Gabi Kastenmüller[18], Anurag Sethi[21], Madeleine Cule[21], Anil Raj[21], Alnylam Human Genetics*, AstraZeneca Genomics Initiative*, Biogen Biobank Team*, Bristol Myers Squibb*, Genentech Human Genetics*, GlaxoSmithKline Genomic Sciences*, Pfizer Integrative Biology*, Population Analytics of Janssen Data Sciences*, Regeneron Genetics Center*, Lucy Burkitt-Gray[16], Eugene Melamud[21], Mary Helen Black[9], Eric B. Fauman[2], Joanna M. M. Howson[3], Hyun Min Kang[12], Mark I. McCarthy[4], Paul Nioi[17], Slavé Petrovski[10,22], Robert A. Scott[6], Erin N. Smith[23], Sándor Szalma[23], Dawn M. Waterworth[24], Lyndon J. Mitnaul[12], Joseph D. Szustakowski[8,27], Bradford W. Gibson[5,27], Melissa R. Miller[2,27] & Christopher D. Whelan[1,25,27✉]

The Pharma Proteomics Project is a precompetitive biopharmaceutical consortium characterizing the plasma proteomic profiles of 54,219 UK Biobank participants. Here we provide a detailed summary of this initiative, including technical and biological validations, insights into proteomic disease signatures, and prediction modelling for various demographic and health indicators. We present comprehensive protein quantitative trait locus (pQTL) mapping of 2,923 proteins that identifies 14,287 primary genetic associations, of which 81% are previously undescribed, alongside ancestry-specific pQTL mapping in non-European individuals. The study provides an updated characterization of the genetic architecture of the plasma proteome, contextualized with projected pQTL discovery rates as sample sizes and proteomic assay coverages increase over time. We offer extensive insights into *trans* pQTLs across multiple biological domains, highlight genetic influences on ligand–receptor interactions and pathway perturbations across a diverse collection of cytokines and complement networks, and illustrate long-range epistatic effects of *ABO* blood group and *FUT2* secretor status on proteins with gastrointestinal tissue-enriched expression. We demonstrate the utility of these data for drug discovery by extending the genetic proxied effects of protein targets, such as PCSK9, on additional endpoints, and disentangle specific genes and proteins perturbed at loci associated with COVID-19 susceptibility. This public–private partnership provides the scientific community with an open-access proteomics resource of considerable breadth and depth to help to elucidate the biological mechanisms underlying proteo-genomic discoveries and accelerate the development of biomarkers, predictive models and therapeutics[1].

Genetic studies of human populations are increasingly used as research tools for drug discovery and development. These studies can facilitate the identification and validation of therapeutic targets[2,3], help to predict long-term consequences of pharmacological intervention[4], improve patient stratification for clinical trials[5] and repurpose existing drugs[6]. Several precompetitive biopharmaceutical consortia have recently invested in population biobanks to accelerate genetics-guided drug discovery, enhancing massive-scale phenotype-to-genotype studies such as the UK Biobank (UKB)[7,8] with comprehensive multi-omics profiling of biological samples[9–11].

Ongoing private–public investments in biobank-based genetics are supported, in part, by a series of systematic analyses of historical drug development pipelines, all indicating that drugs developed with supporting evidence from human genetics are at least twice as likely

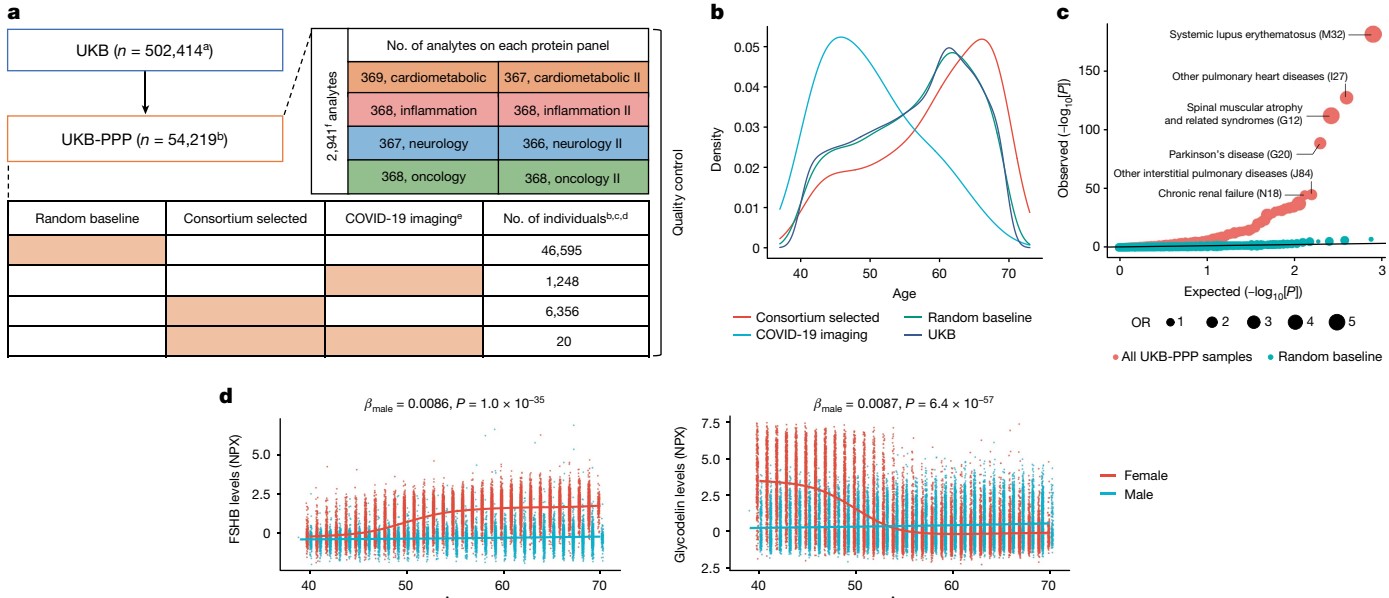

**Fig. 1 | Overview of UKB-PPP. a**, Sample set-up and protein measurements. The number of individuals comprising each cohort (random baseline, consortium selected, COVID-19 imaging, or a combination) is represented by the orange boxes. **b**, The age distribution between different subcohorts. **c**, Q–Q plot showing enrichment *P* values of the full UKB cohort compared against all of the UKB-PPP samples and UKB-PPP randomly selected baseline samples. Statistical analysis was performed using two-sided, unadjusted Fisher's exact tests. **d**, Follicle-stimulating hormone beta subunit (FSHB) and glycodelin (PAEP) levels by age and sex. Linear regression coefficients and two-sided unadjusted *P* values for males are shown. [a]The number is based on the October 2021 release

of the UKB. [b]Samples from individuals who have withdrawn from the study are excluded except in the sample-processing schematic. [c]Samples ($n = 13$) and plates ($n = 4$) that were damaged/contaminated were not included in the summaries except for in the sample-processing schematic. [d]Multiple measurements include a combination of blind duplicate samples and bridging samples. [e]Participants selected for COVID-19-positive status measured at baseline ($n = 1,230$), visit 2 ($n = 1,209$) and visit 3 ($n = 1,261$). Visit 2 and 3 measurements were performed together in batch 7. [f]2,923 unique proteins; 6 proteins were measured across 4 protein panels. NT-proBNP and BNP, IL-12A and IL-12 are treated as separate proteins. NPX, normalized protein expression.

to be approved[12,13]. Nonetheless, human genetics remains an imprecise instrument for biopharmaceutical research and development, as genome-wide association studies (GWAS) frequently implicate genetic variants without clear causal genes mediating their impact(s)[14] or map to genes implicating putative drug targets with poorly understood biology or unclear mechanisms of modulation[2]. Combining human genetics with high-throughput, population-scale proteomics could help to bridge the gap between the human genome and human diseases[1]. Circulating proteins could also provide insights into the current state of human health[15] and partially capture the influences of lifestyle and environment on disease pathogenesis[16].

To date, large-scale proteogenomic studies have identified upwards of 12,000 independent associations between genetic variants and plasma protein concentrations (pQTLs) using samples typically sourced from studies lacking participant-level access or linkage to deep phenotyping[17–23]. The open-access framework[24], deep phenotypic characterization[7] and long-term development[9,10] of population studies like the UKB offer a unique opportunity to expand proteogenomics to a massive scale, broaden research use of high-throughput proteomic data, build more extensive pQTL databases, and accelerate the discovery of biomarkers, diagnostics and medicines. To fulfil these aims, we formed the UK Biobank Pharma Proteomics Project (UKB-PPP)—a precompetitive consortium of 13 biopharmaceutical companies funding the generation of multiplex, population-scale proteomic data. Here we describe the measurement, processing and downstream analysis of 2,941 blood plasma analytes measured across 54,219 UKB participants using the antibody-based proximity extension assay (PEA)[25].

## Overview of UKB-PPP characteristics

We conducted proteomic profiling on blood plasma samples collected from 54,219 UKB participants using the antibody-based

Olink Explore 3072 PEA, measuring 2,941 protein analytes and capturing 2,923 unique proteins (Fig. 1a, Supplementary Information and Extended Data Fig. 1). This included a randomly selected subset of 46,595 UKB participants at the baseline visit (randomly selected baseline), 6,376 individuals at baseline selected by the UKB-PPP consortium members (consortium selected) and 1,268 individuals who participated in the COVID-19 repeat-imaging study at multiple visits (Fig. 1a and Methods).

The randomly selected baseline participants were highly representative of the overall UKB population for various demographic characteristics, except for a slightly higher deprivation index (Townsend index, 0.51 higher, $P = 1.4 \times 10^{-5}$), different distribution of recruitment centres ($P = 2.1 \times 10^{-95}$) and a minimal difference in time since recruitment ($P = 0.00072$) (Supplementary Table 1). Compared with the UKB participants overall, the consortium-selected participants were on average older (by 2.5 years, $P = 5.0 \times 10^{-117}$) and had a lower proportion of women (by 3.2%, $P = 4.1 \times 10^{-7}$), a higher body mass index (BMI; 0.6 kg m$^{-2}$ higher, $P = 1.3 \times 10^{-16}$), a lower prevalence of never smokers (2.4% lower, $P = 2.1 \times 10^{-6}$) and different composition of self-reported ethnic background (UKB data field 21000) ($P = 3.8 \times 10^{-296}$), with a higher proportion (6.3% higher) of self-reported non-white participants (Fig. 1b and Supplementary Table 1). The COVID-19 imaging participants had a younger age distribution (by 6.3 years, $P = 1.2 \times 10^{-162}$), lower BMI (1.1 kg m$^{-2}$ lower, $P = 1.7 \times 10^{-20}$) and lower smoking prevalence (7.7% lower, for individuals who have never smoked, $P = 2.1 \times 10^{-9}$), but were comparable to the overall UKB participants in sex, ethnic background and blood group (Supplementary Table 1). The consortium-selected and COVID-19 imaging participants showed widespread differences across medication use patterns, haematology measures and biochemistry markers, reflecting their non-random sampling, whereas the randomly selected baseline participants remained highly representative of the UKB overall (Supplementary Table 1).

Compared with the full UKB cohort, UKB-PPP participants were enriched for 119 diseases, spanning multiple systems, at a Bonferroni-corrected threshold of $P < 6.7 \times 10^{-5}$ (0.05/746 diseases), with no significant depletion in the diseases tested after adjustment for multiple comparisons (Fig. 1c and Supplementary Table 2). This enrichment was largely driven by the inclusion of consortium-selected and COVID-19 imaging participants (Methods) as the enrichments were mostly attenuated when considering randomly selected baseline samples only (Fig. 1c and Supplementary Table 2).

## Data processing and quality control

A total of 2,923 unique proteins was measured across eight protein panels (cardiometabolic, cardiometabolic II, inflammation, inflammation II, neurology, neurology II, oncology and oncology II; Fig. 1a and Extended Data Fig. 1). Detailed descriptions of the antibody-based Olink Explore 3072 platform, study-wide protein measurement, processing and quality control are outlined in Fig. 1a, Supplementary Figs. 1–12, Supplementary Table 31 and Extended Data Fig. 1. Analyses and results are based on data after quality control; all of the exclusions are described in the Supplementary Information.

Globally, we did not observe batch effects, plate effects or abnormalities in protein coefficients of variation (CVs) (Supplementary Information). 100% and 99.5% of proteins had variabilities of less than 10% attributable to batch and plate, respectively (Supplementary Information). Protein CVs, representing intraindividual variability across duplicate samples, ranged from 1.8% to 27.2%, with a median of 6.7% (Supplementary Table 3 and Supplementary Information). We observed that, on balance, CVs varied across the dilution factors. Proteins with lower expected concentrations (dilution factor 1:1) exhibited higher CVs, whereas proteins with the highest expected concentrations (dilution factor 1:100,000) exhibited lower CVs. Most (73.2%) protein analytes had <30% of samples below the limit of detection (LOD), and the majority (67.3%) had <10% below the LOD (Supplementary Information). The proportion of samples below the LOD exhibited a similar trend across dilution factors (Extended Data Fig. 1c and Supplementary Table 3).

We observed high correlations between the same protein targets assayed across multiple panels (Supplementary Information and Extended Data Fig. 2a). We also compared seven protein measures acquired using the antibody-based Olink assay with their corresponding protein measurements obtained using independent assays in the UKB; further details are provided in the Supplementary Information and Supplementary Table 4.

## Proteomic links with health and disease

We investigated proteomic associations with demographic factors (age, sex, BMI), health burden, prevalent diseases and markers of renal and liver function (Methods). The results, summarized in the Supplementary Information, Supplementary Tables 5–8 and Extended Data Fig. 3, validate several established proteomic associations, such as elevated N-terminal pro-brain natriuretic peptide (NT-proBNP) in ischaemic heart disease[26]. We also uncovered physiological age and sex interactions for proteins such as follitropin subunit beta and glycodelin (Fig. 1d) and demonstrated that plasma proteomic measures can infer age, sex, BMI, blood groups, and renal and liver function with high predictive accuracy (Extended Data Fig. 3c, Supplementary Table 8 and Supplementary Information).

## Discovery of pQTLs

The UKB demographic composition and its potential impact on research projects, especially those involving genetics, is well characterized[27]. To enable appropriate breadth and robustness of pQTL discovery, we performed proteo-genomic analyses in independent discovery/replication subgroups and in non-European, ancestry-specific subgroups.

Discovery pQTL analyses were performed in participants of European ancestry from the randomly selected baseline cohort ($n = 34,557$), which was broadly representative of the full UKB cohort, with the remaining samples ($n = 17,806$) used as a replication cohort (Fig. 1b,c and Supplementary Tables 1 and 2). We performed pQTL mapping of up to 16.1 million imputed variants for 2,922 proteins after quality control. We identified 14,287 significant primary associations across 3,760 independent genetic regions at a multiple testing-corrected threshold of $P < 1.7 \times 10^{-11}$ (Fig. 2a and Supplementary Table 9). At a less stringent, single-phenotype genome-wide significance threshold of $P < 5 \times 10^{-8}$, we found 14,731 additional associations across 2,519 proteins. The results are based on associations that remained significant after adjustment for multiple testing, unless otherwise indicated.

Globally, 2,414 of the 2,922 proteins tested (82.6%) had at least one pQTL at $P < 1.7 \times 10^{-11}$, with 66.9% of proteins tested (1,954 of 2,922 proteins) having a *cis* association (within 1 Mb from the gene encoding the protein). When stratified by dilution levels, we found that the least-abundant dilution section (1:1) had significantly lower proportions of proteins with ≥1 total pQTLs (74.5%) and *cis* pQTLs (55.3%) compared with the more-abundant dilution sections (1:10 to 1:100,000, 98.6% and 89.6% of proteins had ≥1 total and *cis* pQTLs, respectively) (Extended Data Fig. 4a). Concordantly, we found a significant negative relationship between the number of pQTLs and the proportion of samples that were below the LOD (Spearman's $\rho = -0.69$, $P < 10^{-300}$; Extended Data Fig. 4b), whereby 81.4% of proteins without a pQTL (compared with 10.0% of proteins with pQTLs) have more than 50% of samples below the LOD (Extended Data Fig. 4c). For proteins with *cis* pQTLs, we found significant enrichments ($P < 0.01$, correcting for five categories) in the proportions of proteins that are secreted (odds ratio (OR) = 1.52, $P = 1.1 \times 10^{-11}$), whereas we found depletions in cytoplasmic (OR = 0.76, $P = 1.6 \times 10^{-5}$) and nuclear (OR = 0.55, $P = 2.2 \times 10^{-12}$) proteins compared with the assay background. These enrichments and depletions were attenuated when considering proteins with any pQTL, but enrichment for secreted proteins (OR = 1.22, $P = 6.5 \times 10^{-4}$) and depletion for nuclear proteins (OR = 0.80, $P = 0.0032$) remained.

We observed a median of 4 primary associations (5th–95th quantiles: 1–17) per protein, with 62 proteins (2.6%) having at least 20 associations (Fig. 2b). Genomic inflation was well controlled, with median $\lambda_{GC} = 1.02$ (s.d. = 0.019). The general inverse trend between effect-size magnitudes and minor allele frequency (MAF) remained for both *cis* and *trans* associations, with *trans* associations showing smaller magnitudes of effect sizes compared with *cis* associations (Fig. 2c). Approximately 5.5% (783 out of 14,287) and 1.6% (235 out of 14,287) of the primary associations had MAF < 1% and 0.5%, respectively.

A total of 1,955 of the 14,287 primary associations were in *cis* and 12,332 were in *trans* (>1 Mb from the gene encoding the protein). In total, 60%, 86% and 92% of the *cis* associations were within the gene, 50 kb and 100 kb from the gene start site, respectively. We found no systematic enrichment of *trans* pQTLs occurring on the same chromosomes as the protein tested after adjusting for chromosome lengths (Fisher's test, $P = 0.56$). Of the *trans* pQTLs located on the same chromosome as the gene encoding the protein, all but two were located more than 2 Mb away from their corresponding gene (93% were further than 5 Mb away and 81% were further than 10 Mb away).

In total, 62% (2,326 out of 3,760) of the non-overlapping genetic regions were associated with a single protein, whereas 11% were associated with at least 5 proteins (pleiotropic region), and 16 loci were extremely pleiotropic, associated with at least 100 proteins (Fig. 2a). These included well-established, previously identified[17–20] pleiotropic loci such as *MHC*, *ABO*, *ZFPM2*, *ARHGEF3*, *GCKR*, *SERPINA1*, *SH2B3* and *ASGR1*.

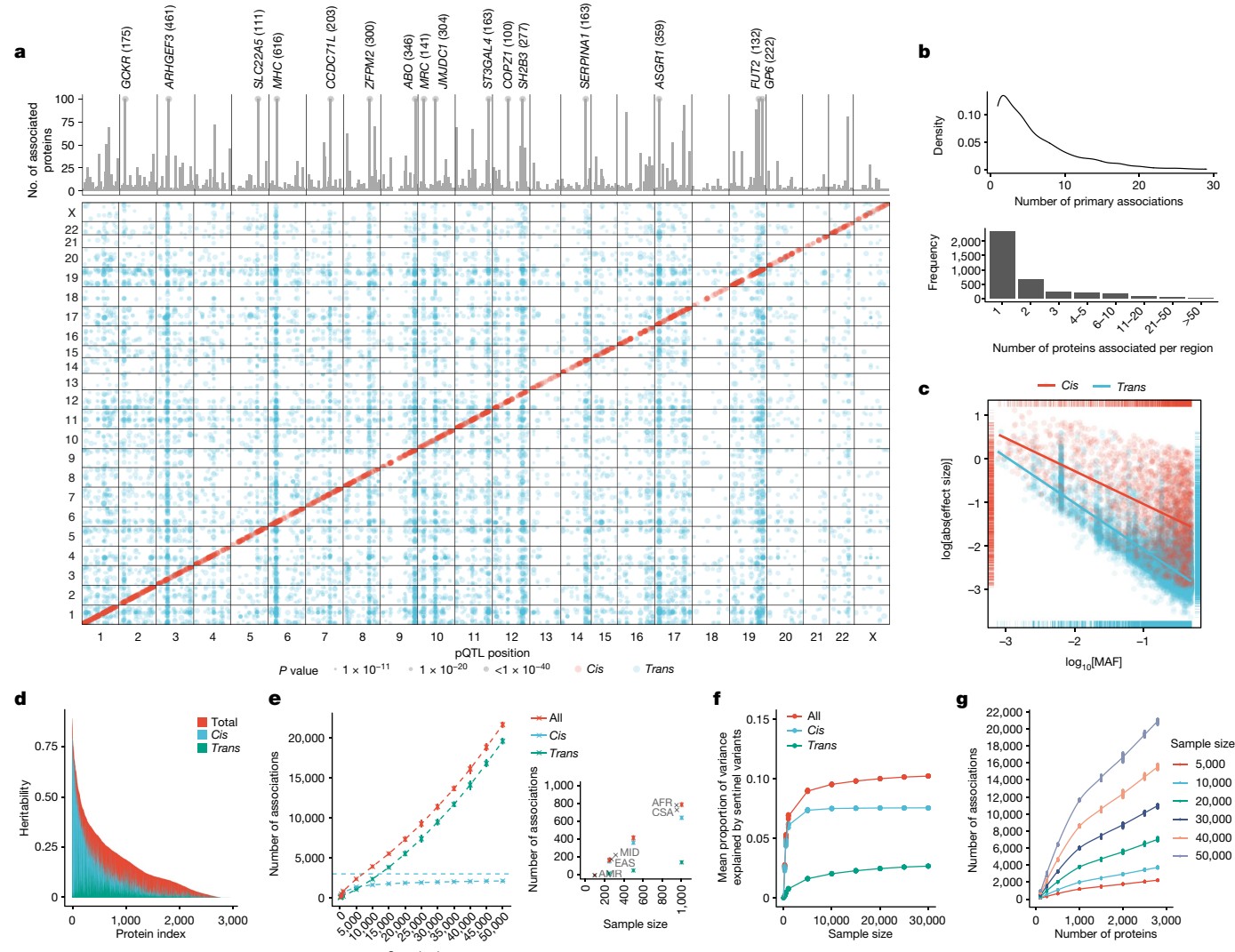

**Fig. 2 | The genetic architecture of pQTLs. a**, Summary of pQTLs across the genome. Bottom, genomic locations of pQTLs against the locations of the gene encoding the protein target. Red, *cis* pQTLs; blue, *trans* pQTLs. Top, the number of associated protein targets for each genomic region (the axis is capped at 100; regions with >100 number of associated proteins are labelled, with the number in parenthesis). **b**, The number of primary pQTLs per protein (top) and the number of associated proteins per genomic region (bottom). **c**, The log absolute effect size against log[MAF] by *cis* and *trans* associations.

The lines indicate the linear regression slope for *cis* (red) and *trans* (blue) associations. **d**, The distribution of heritability and contributions from primary *cis* and *trans* pQTLs. **e**, The number of primary associations against sample size. Data are mean ± 3 s.d. of *n* = 10 independent sets of random subsamples at each sample size strata. **f**, The mean proportion of variance explained by primary pQTLs against sample size. **g**, The number of primary associations against the number of proteins assayed.

## Replication of pQTLs

A total of 95.6% (1,869 out of 1,955) of *cis* and 64.1% (7,906 out of 12,332) of *trans* associations remained significant ($P < 1.3 \times 10^{-5}$ after adjusting for the number of associated unique genomic regions) and directionally concordant in the replication cohort (*n* = 17,806) (Supplementary Table 9), consistent with previous large-scale studies[17–20]. Effect sizes were well-aligned between the discovery and replication sets (*r* = 0.99, $P < 10^{-300}$; Extended Data Fig. 5a). Moreover, we observed good concordance of replicated (*P* < 0.05) genetic associations, where available, between proteins measured across multiple protein panels (CXCL8, interleukin 6 (IL-6), TNF, IDO1 and LMOD1; Extended Data Fig. 2b), reflecting their phenotypic correlations (Extended Data Fig. 2a).

To maximize power and variant coverage, we also performed pQTL mapping for the full UKB-PPP cohort (*n* = 52,363) of up to 23.8 million imputed variants, identifying 23,588 putative primary associations ($P < 1.7 \times 10^{-11}$; Supplementary Table 10).

## Non-European pQTL mapping

We performed ancestry-specific pQTL analyses within five ancestry groups as defined by pan-UKBB: African (AFR, *n* = 931), Central/South Asian (CSA, *n* = 920), Middle Eastern (MID, *n* = 308), East Asian (EAS, *n* = 262) and admixed American (AMR, *n* = 97), with minor allele count (MAC) > 10 corresponding to a minimum MAF of 0.5% in AFR to 5% in AMR. At $P < 1.7 \times 10^{-11}$, we found 785 (AFR), 732 (CSA), 227 (MID) and 179 (EAS) primary associations (Supplementary Table 11). Given the small sample size for AMR, we observed only 9 associations, all *cis*, at $P < 5 \times 10^{-8}$. The number of significant associations for each ancestry was consistent with those expected in similar, predominantly (95%) European ancestry sample sizes (Fig. 2e), suggesting that there is no major global enrichment of total pQTLs for any ancestry. Effect sizes were highly aligned (Extended Data Fig. 5b) between European and non-European ancestries.

In total, 531 (AFR, 68%), 712 (CSA, 97%), 221 (MID, 97%) and 174 (EAS, 97%) of the primary pQTLs were also significant in the European

ancestry group (EUR) at $P < 1.7 \times 10^{-11}$. Despite the similar sample sizes between AFR and CSA, the median MAF enrichment of the lead primary ancestry associations compared with EUR was much higher for AFR (median[$MAF_{AFR}/MAF_{EUR}$] = 2.00) than for CSA (median[$MAF_{AFR}/MAF_{EUR}$] = 1.07). Thus, higher allelic enrichment in participants of AFR ancestry is an important driver of associations not detected in participants of EUR ancestry.

Of the primary associations found in non-European ancestries, 212 out of 254 (AFR), 16 out of 20 (CSA), 5 out of 6 (MID) and 3 out of 4 (EAS) had MAF < 0.07% (corresponding to MAC < 50) in EUR. For example, the rare variant rs202092481, which leads to a premature stop codon in *CD1C* (Arg43Ter), is associated with strongly decreased CD1C levels in CSA ($\beta = -2.23$, $P = 1.4 \times 10^{-15}$). This variant is almost exclusively seen in CSA ($MAF_{CSA} = 0.7\%$) and is nearly absent in EUR and other ancestries[28]. The missense variant rs72938840 (Arg59Gln, $MAF_{AFR} = 7.9\%$) in the gene encoding rhomboid protease, *RHBDL2*, is mostly absent in EUR and other ancestries[28] and is associated in *trans* with concentrations of one of its cleavage substrates[29], SPINT1 ($P = 5.8 \times 10^{-12}$). We annotated the consequences of other potential high-impact pQTLs, as summarized in the Supplementary Information and Supplementary Tables 12–14.

Notably, at the *SERPINA12* locus, we found a *cis* primary association in all four non-European ancestries, in proxy with the intronic functional variant rs77060950[30] ($r^2 = 1$) where the minor T allele was associated with higher SERPINA12 (vaspin) levels ($MAF_{AFR} = 10.0\%$, $P_{AFR} = 5.8 \times 10^{-83}$; $MAF_{CSA} = 6.2\%$, $P_{CSA} = 4.8 \times 10^{-72}$; $MAF_{MID} = 7.8\%$, $P_{MID} = 3.2 \times 10^{-17}$; $MAF_{EAS} = 3.8\%$, $P_{EAS} = 3.4 \times 10^{-15}$) (Extended Data Fig. 5c). Liver expression of *SERPINA12* was also higher in rs77060950-T carriers ($P = 2.6 \times 10^{-8}$)[31]. The rs77060950 variant is depleted in EUR (MAF = 1.2%) and did not reach genome-wide significance in EUR despite much larger sample sizes ($P = 0.0047$).

## Comparisons with previous pQTL studies

We cross-referenced pQTLs identified in this study with previously published pQTL results (Methods and Supplementary Table 32), finding that 81% of primary associations from the discovery cohort (11,521 out of 14,287) had not been identified by previous pQTL studies (Supplementary Table 15). A larger percentage of *trans* pQTLs were previously undescribed (89%; 11,002 out of 12,332) compared with *cis* pQTLs (27%; 518 out of 1,954). When comparing with previously identified pQTL results from antibody-based studies, 84% (934 out of 1116) of the previous associations were replicated in our discovery study. For aptamer-based pQTL results, limiting to the set of proteins common to both platforms, 38% of previous associations replicated in our discovery study (1,877 out of 4,978 associations, across 1,982 proteins).

## Identification and fine-mapping of independent signals

We identified 29,420 independent pQTL signals with SuSiE regression of individual-level protein levels on genotype dosages and confirmed statistical independence using multivariable linear regression models (Supplementary Table 16). This included 10,750 and 18,670 signals that mapped to *cis* and *trans* regions, respectively. In total, 87% (1,717 out of 1,967) of *cis* regions contained more than one signal (mean, 5.5 signals per *cis* region) (Extended Data Fig. 6a). We also performed fine-mapping using SuSiE to narrow down 95% credible sets of causal variants for each pQTL signal (Supplementary Table 16). Credible sets contained an average of 20.5 variants (range, 1–3,189) and were generally better resolved for *cis* signals than for *trans* signals (mean credible set size, 9.7 (*cis*) and 26.7 (*trans*)) (Extended Data Fig. 6b, Supplementary Information and Supplementary Tables 17 and 18).

## SNP-based heritability of proteins

We estimated single-nucleotide polymorphism (SNP)-based heritability as the sum of contributions from significant independent pQTLs identified by SuSiE (pQTL component) and the remaining SNPs across the genome (excluding the pQTL region), which assumes a polygenic model (polygenic component) using an approach that was described previously[32] (Methods and Supplementary Table 19). The mean total SNP-based heritability was 0.16 (5–95th quantiles, 0–0.50). On average, *cis* primary pQTLs accounted for 20.5% of the overall heritability, whereas *trans* pQTLs accounted for 10.4% (Fig. 2d and Extended Data Fig. 7a). We found a significant correlation between the lead pQTL component and the polygenic component (Spearman's $\rho = 0.68$, $P < 10^{-300}$; Extended Data Fig. 7b), with stronger correlations between the polygenic component and *trans* pQTL component ($\rho = 0.74$, $P < 10^{-300}$) compared with the *cis* pQTL component ($\rho = 0.56$, $P = 1.7 \times 10^{-243}$).

## Protein interactions and pathways at *trans* loci

*Trans* associations may reflect interactions between the protein products of genes at the *trans* locus and the target protein (Fig. 3a). Moreover, genes at/near *trans* loci may operate within the same pathway as the target protein and modulate target protein levels. We used the Human Integrated Protein–Protein Interaction Reference (HIPPIE)[33] to test whether *trans* pQTL loci contained at least one gene that encoded proteins interacting with the target protein tested. Overall, we found an interacting partner at *trans* loci for 861 proteins, which is enriched by 1.16 times compared with the permuted background ($n = 100$ times, empirical $P < 0.01$) (Methods and Supplementary Table 20). We found different gene products at the same pleiotropic *trans* loci interacting with different proteins with associations in those regions, potentially explaining certain pleiotropic effects. For 1,055 *trans* associations, we found a single, specific interacting protein candidate (Supplementary Table 20). We also found 27 cases of reciprocal interactions, where the protein tested interacted with a protein in one of its *trans* loci and vice versa, indicating strong coupled interactions (Supplementary Table 21). For pQTLs associated with multiple independent regions (≥5), gene set enrichment analysis revealed enriched biological functions and pathways for 254 proteins, implicating pathways involved in cellular activation, survival and signalling relevant to immune cells (summarized in Supplementary Table 22 and the Supplementary Information).

Notably, in addition to the HSPB6 *trans* pQTL at the *BAG3* locus (rs2234962; Cys151Arg), we found *trans* associations for MB, MYOM3, MYBPC1, MYL3, proBNP and NT-proBNP. BAG3 functions through BAG3–HSP70–HSPB complexes, which have an important role in heart failure and cardiomyopathies[34], including the same *BAG3* signal (rs2234962) identified in previous GWAS of cardiomyopathies[35,36]. ProBNP and NT-proBNP are established biomarkers of heart failure and cardiac damage[26], whereas MB, MYOM3, MYBPC1 and MYL3 are all myocyte (MB)/myofibrillar proteins. The rs2234962 pQTL is an independent secondary *cis* pQTL for BAG3 levels from the primary *cis* pQTL (rs35434411; Arg71Gln; Supplementary Table 16), for which we did not find significant evidence of association with the aforementioned proteins ($P > 1.5 \times 10^{-5}$). The rs2234962 missense variant sits in between two conserved IPV motifs, which are essential for HSPB6/8 binding[37] and may modulate interactions with HSP/HSPBs in vitro[38]. Taken together, these results provide evidence of different mechanisms of effect driven by different variants in *BAG3* (Extended Data Fig. 8), with the rs2234962 missense variant potentially affecting both BAG3 levels and BAG3–HSPB6 complexing, leading to downstream perturbations in cardiac muscle proteins, downstream blood biomarkers of heart failure and, potentially, risk of cardiomyopathies.

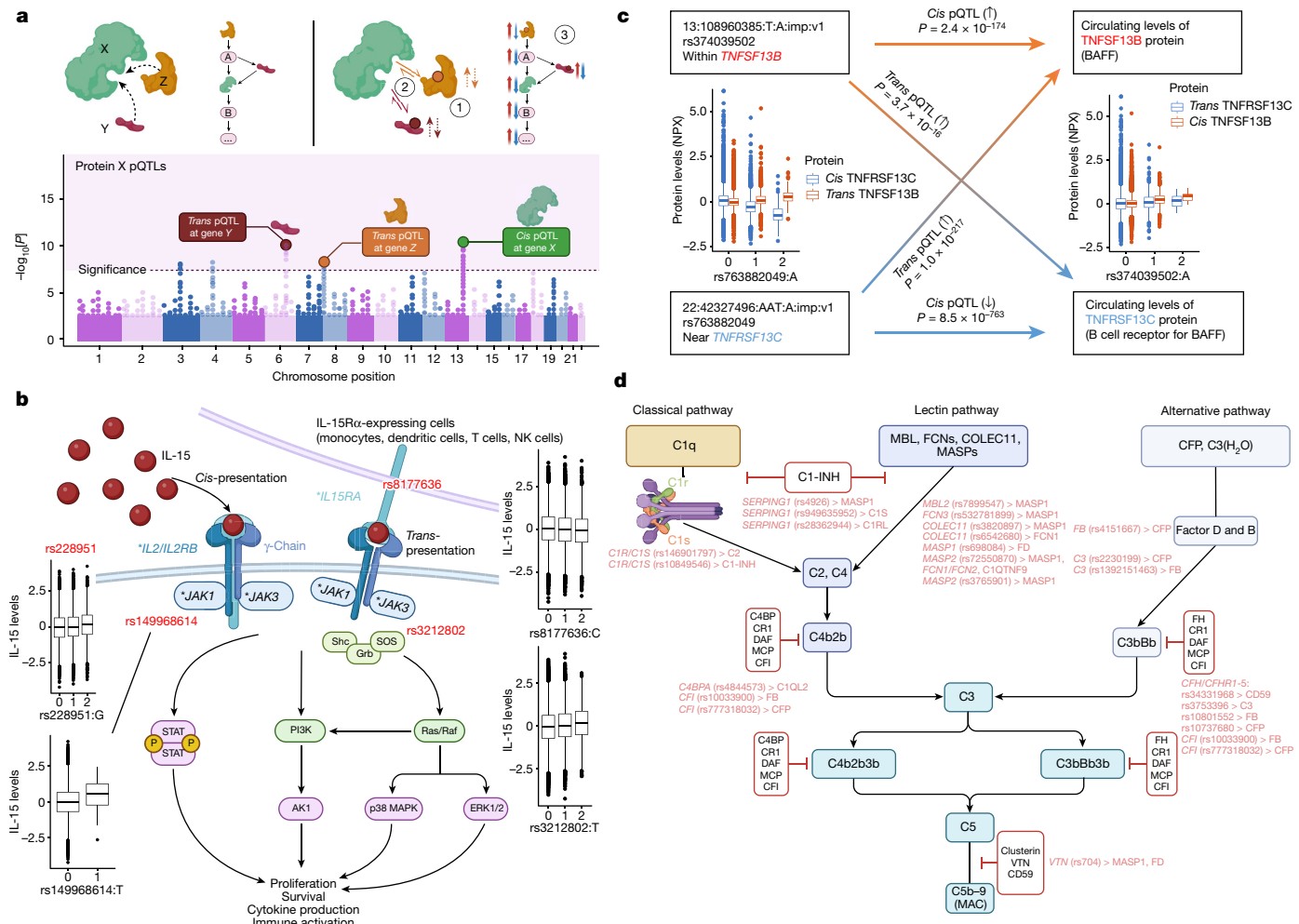

**Fig. 3 | Examples of pathway networks highlighted by *trans* pQTLs.**
**a**, Schematic of how *trans* pQTLs function as part of the same protein–protein interaction or pathway as the protein tested (protein X). Top left, proteins involved may be directly interacting or indirectly involved as part of the same pathway. Bottom, *trans* pQTLs found for corresponding genes in *trans* (in addition to potentially other signals and *cis* associations regulating protein X). Top right, some of the mechanisms by which the *trans* pQTLs may regulate the target protein (protein X), including: (1) regulating the levels of the binding partners (Y, Z), which in turn affects protein X levels; (2) altering the interaction between Y/Z with X; (3) modulating components of the pathway in which Y/Z may be upstream/downstream of protein X. The figure was created using BioRender, including adaptations from 'The principle of a genome-wide association study'. **b**, The IL-15-signalling pathway. The asterisks indicate genes with *trans* pQTLs for IL-15 (the primary association SNP is shown in red). The figure was created using BioRender, including adaptations from 'Thrombopoietin receptor signaling'. NK, natural killer. **c**, Example of a bidirectional *trans* pQTL pair. *P* values were derived from REGENIE regression GWAS (two-sided, unadjusted). Orange and blue solid arrows represent *cis* pQTLs for TNFSF13B and TNFRSF13C; gradient lines represent *trans* effects of *TNFSF13B* variants on TNFRSF13C protein levels and *trans* effects of *TNFRSF13C* variants on TNFSF13B levels. **d**, The complement pathway. *Trans* pQTLs and the associated proteins are shown in red. The figure was created using BioRender. The box plots in **b** and **c** show the median (centre line), first and third quartiles (box limits), and 1.5× the interquartile range above and below the third and first quartiles (upper and lower whiskers). *n* = 52,363 independent samples.

## Insights into cytokine interactions

The increased detection of *trans* pQTLs facilitated by this study provided an opportunity to uncover causal biological networks. We found multiple instances of receptor–ligand interactions at *trans* loci for circulating cytokines and TNF-superfamily proteins/receptors (Supplementary Table 23). In addition to *trans* pQTLs for IL-15 at genes encoding its receptor components (IL-15RA and IL-15RB), we also found *trans* pQTLs at both *JAK1* and *JAK3*, which are proximal components of IL-15 signalling (Fig. 3b and Supplementary Table 20); notably, the *trans* pQTL at *JAK1* is a rare missense mutation (rs149968614, MAF = 0.2%, Val651Met). Furthermore, we found that the variant rs4985556-A, which causes a premature stop gain in *IL34*, is associated with lower levels of IL-34 in *cis* ($\beta = -1.07$, $P = 4.5 \times 10^{-1,787}$) and lower CD207—a protein marker expressed in Langerhans cells—levels in *trans* ($\beta = -0.08$, $P = 2.7 \times 10^{-15}$).

Although IL-34 and CD207 do not directly interact, this result is highly consistent with the crucial role of IL-34 in the development and survival of Langerhans cells[39].

We uncovered proof-of-concept evidence for bidirectional *trans* pQTL pairs; that is, where a locus is both a *cis* pQTL for protein A and a *trans* pQTL for protein B, and a second locus is a *cis* pQTL for protein B and a *trans* pQTL for protein A. B-cell-activating factor (BAFF) and the BAFF receptor present such a pair (Fig. 3c). The variant rs374039502 on chromosome 13, near the gene encoding BAFF (*TNFSF13B*), is both a *cis* pQTL for the cytokine BAFF and *trans* pQTL for its receptor. The variant rs763882049 on chromosome 22 is both a *cis* pQTL for the BAFF receptor (encoded by *TNFRSF13C*) and *trans* pQTL for its ligand, BAFF (Fig. 3c). This locus on chromosome 13 is well-known for its association with blood cell traits and autoimmune diseases[40–42]. BAFF has a well-established role in B cell survival and function and is the drug target

for belimumab, a monoclonal antibody approved for the treatment of systemic lupus erythematosus[43]. These results demonstrate that bidirectional *trans* pQTL pairs can help to identify and characterize biological networks relevant to health traits, disease biology and drug targets.

## Complement cascade *trans* pQTL networks

In the complement pathway, we found multiple *trans* pQTLs in genes for various constituents within the same complement pathway as the protein tested (Fig. 3d). Notably, for the protein MASP1, we found that 6 out of the 13 *trans* associations lie in genes encoding other components of the complement pathway (including the lectin-pathway genes *MASP2, MBL2, FCN3, COLEC11, SERPING1* (encoding C1-inhibitor, also known as C1-INH) and *VTN*), all of which, except for *VTN*, showed direct interactions with MASP1 (Fig. 3d). Furthermore, the MASP1 *trans* pQTL rs6118 is a missense variant in the gene encoding protein C inhibitor (*SERPINA5*, Ala55Val), which has a key regulatory role in the coagulation pathway closely linked to complement. We also found that a variant in *MASP1* (rs698084) is associated in *trans* with factor D levels. Notably, the *trans* pQTL at *FCN3* is a low-frequency frameshift variant (rs532781899, MAF = 1.4%) leading to FCN3 deficiency[44–46] and, here, to reduced MASP1 levels ($\beta = -1.16$, $P = 3.8 \times 10^{-312}$). Similarly, we found a low-frequency missense variant in *MASP2* (rs72550870, Asp120Gly, MAF = 3.0%), previously linked to MASP2 deficiency[47–49], associated in *trans* with higher MASP1 levels ($\beta = 0.17$, $P = 2.5 \times 10^{-15}$; independent to the primary association), reduced FCN2 levels ($\beta = -0.22$, $P = 2.6 \times 10^{-32}$), higher FCN1 levels ($\beta = 0.17$, $P = 1.4 \times 10^{-17}$) and lower C1QTNF9 levels ($\beta = -0.12$, $P = 6.6 \times 10^{-14}$). We also found that C2 and C1-INH levels are associated with *trans* pQTLs at *C1R/C1S*; C1S and C1RL levels with *trans* pQTLs at *SERPING1*; C1QL2 levels in *trans* with a missense *C4BPA* variant (rs4844573, Ile300Thr); properdin (CFP) levels in *trans* with a missense factor B (*CFB*) variant (rs4151667, Leu9His); factor B and CFP levels with *trans* pQTLs at CFI and also C3; FCN1 levels with a *trans* pQTL at *COLEC11*; and C3, factor B, CFP and CD59 levels with *trans* pQTLs at the *CFH-CFHR1-5* locus (Fig. 3d). Most *trans* genetic effects occurred upstream of the proteins forming the membrane attack complex (C5b–C9). All proteins with *trans* pQTLs described here also have *cis* associations locally, demonstrating a complex, intricate network of local and long-range genetic perturbations on various proteins within a pathway system.

## Scaling of pQTL associations

Previous studies have performed pQTL mapping across different sample sizes and varying numbers of proteins. Here, through subsampling of participants and proteins, we investigated how the number of associations scaled with sample size and number of proteins assayed. We provide a detailed summary in Fig. 2e–g and in the Supplementary Information. In brief, the rate of increase in *cis* pQTL detection with increasing sample size plateaued after 5,000 samples, whereas *trans* pQTLs continued to increase (Fig. 2e). We observed corresponding trends for the mean variances explained by pQTLs (Fig. 2f). Accounting for the lowering abundance in proteins assayed, we began to see reducing yields of pQTL findings with additional proteins measured (Fig. 2g). We also investigated the detectability of pleiotropic and *trans* loci harbouring protein interactions (summarized in the Supplementary Information and Extended Data Fig. 9).

## Sensitivity analyses of pQTLs

We examined the impact of blood cell composition, BMI, season and fasting time before blood collection on pQTL effects and protein variances; overall, we found a limited impact of these variables on the majority of pQTLs (Supplementary Figs. 13 and 14 and Supplementary Tables 24, 25 and 33).

## Co-localization with expression QTLs

We performed colocalization analyses using coloc with the SuSiE framework to identify shared genetic associations between circulating protein QTLs and tissue-level expression QTLs from the GTEx consortium (v8)[31]. Of genes with a significant eQTL in at least one tissue, 65% (1,220 out of 1,889) shared casual variants with direct effects on both gene and protein expression levels, including 41% (503 out of 1,220) that had multiple colocalized signals (Supplementary Table 26, Supplementary Information and Extended Data Fig. 10).

## Drug targets and disease biology applications

Through a series of exemplar deep-dives, details of which are provided in the Supplementary Information, we showcase how this proteomic dataset can be used to provide insights into protein and pathway perturbations in health and disease and inform therapeutic target discovery and development. In particular, we show how (1) functional genetic interactions between ABO blood group and FUT2 secretor status affect proteins enriched for gastrointestinal tissue expression across humans and mice, which may be perturbed in gastrointestinal diseases (Fig. 4 and Supplementary Table 27); (2) multi-trait colocalization can be applied to COVID-19-associated loci to disentangle shared and distinct protein pathways (Extended Data Fig. 11 and Supplementary Table 28); (3) common genetic variation has a subtle, but significant, role in inflammasome-mediated innate immune responses (Supplementary Table 29); and (4) large-scale proteogenomics studies can increase the power and availability of genetic instruments for Mendelian randomization, mimicking drug target effects observed in clinical trials (Extended Data Fig. 12 and Supplementary Table 30).

## Interactive web portal

To facilitate interactive queries, visualizations and bulk downloads of summary statistics for pQTL results, we created an interactive web portal, which is accessible at http://ukb-ppp.gwas.eu.

## Discussion

High-throughput proteomic profiling of population biobanks holds the potential to accelerate our understanding of human biology and disease. Here we present findings from one of the largest proteomic studies conducted to date—constructing an updated genetic atlas of the plasma proteome, revealing biological insights into prevalent illnesses and providing the scientific community with an open-access, population-scale proteomics resource.

To date, most large-scale, broad-capture proteogenomics studies have used high-throughput, aptamer-based assays (comprehensively listed in the Supplementary Information). Antibody-based genetic studies have focused on narrower collections of proteins at large scale[22] or broad collections of proteins at smaller scale[23]. Certain comparative evaluations have suggested that aptamer-based assays offer broader biological coverage and higher precision, whereas antibody-based assays offer greater protein target specificity and stronger correlations with certain diseases and immunoassays[50]. Other studies have indicated that antibody- and aptamer-based assays are affected by different sources of biological, technical and genetic variation, capturing distinct features of protein chemistry; thus, both technologies may be used as complementary tools for biological discovery[51]. We formed UKB-PPP to complement the existing, extensive library of DNA sequence variants affecting aptamer levels in blood plasma with a comparably sized library of variants influencing antibody-based measurements.

Our analysis identified approximately twentyfold more associations than all previous antibody-based studies. This reflects both

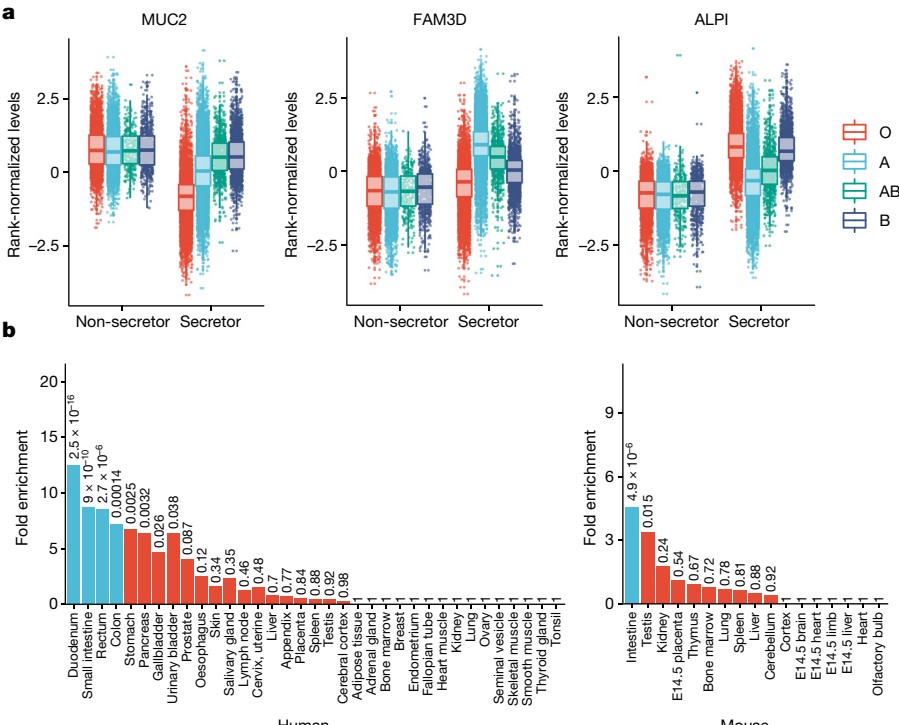

**Fig. 4 | ABO blood group FUT2 secretor status interaction. a**, Protein levels by blood group and secretor status for four proteins with the most significant interaction effects. The box plots show the median (centre line), first and third quartiles (box limits), and 1.5× the interquartile range above and below the third and first quartiles (upper and lower whiskers). $n$ = 52,363 independent samples. **b**, Enrichment of genes encoding proteins with significant interactions ($P < 1.7 \times 10^{-5}$) for expression in various human (left) and mouse (right) tissues. The numbers above the bars represent unadjusted $P$ values calculated using one-sided hypergeometric enrichment tests; the blue bars indicate significance after multiple-testing correction. E14.5, embryonic day 14.5.

the increased sample size of the present study as well as the recently expanded coverage of antibody-based assays. Most previously identified antibody-based pQTLs were replicated in our analysis (84%). When we compared our findings to aptamer-based studies, limiting to the set of common proteins between the two platforms, a much smaller percentage of proteins replicated (38%). This modest overlap is unsurprising, given that correlations between proteins measured by both platforms are highly variable[52,53], and shared genetic signals can be masked by extreme, assay-specific binding affinities[51].

The size and breadth of this study enabled us to estimate how pQTL discoveries may scale with increasing sample size and proteome coverage. We found that the discovery of *cis* pQTLs is saturated to the number of proteins tested after around 10,000 samples. Although *trans* pQTL discoveries continued to increase, the variance explained by *trans* loci increased at a slower rate beyond 10,000 samples. We anticipate that most gains from larger-scale studies will be driven by the detection of *trans* associations with smaller polygenic effects, rare associations, associations with proteins not previously tested, and associations in tissues or sample matrices beyond blood.

The predominantly white European ancestral composition of UKB does not capture the full genetic and phenotypic diversity of the human population. However, in a small, underpowered group of non-European UKB participants, we highlighted how ancestry-specific allelic enrichments can enhance the detection of certain pQTLs that may be absent or very rare in Europeans. These findings further underline the value of prioritizing under-represented and genetically diverse populations for future pQTL mapping studies[54,55]. Future studies should also prioritize resampling initiatives and longitudinal analyses, facilitating more systematic evaluations of assay analytical performance(s), consistencies of personal health signatures, and the effects of disease incidence, prevalence and severity on marker stability and pQTL detection.

Our results expand the catalogue of genetic instruments for downstream Mendelian randomization and associated genomic loci for multi-trait colocalization, facilitating more systematic causal inference and therapeutic target discovery studies, which were beyond the scope of this pre-competitive industry collaboration. As population-scale proteogenomic investigations expand, orthogonal comparisons of different proteomics assays applied across the same samples will help to decipher complementarities and differences between antibody, aptamer and emerging, high-throughput mass-spectrometry-based techniques[50,51]. Further technological advances will enable more comprehensive population-scale investigations incorporating protein isoforms, proteoforms generated by post-translational modifications and single-cell proteomic resolution.

Following on from the successful exome sequencing and the ongoing whole-genome sequencing of the UKB, the Pharma Proteomics Project builds on the precompetitive industry collaboration framework in generating high-dimensional, population-scale data for the advancement of science and medicine. The wider research community will be able to leverage this open-access resource to test hypotheses that are crucial to the development of improved diagnostics and therapeutics for human disease.

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

[1]Translational Sciences, Research & Development, Biogen, Cambridge, MA, USA. [2]Internal Medicine Research Unit, Worldwide Research, Development and Medical, Pfizer, Cambridge, MA, USA. [3]Human Genetics Centre of Excellence, Novo Nordisk Research Centre Oxford, Oxford, UK. [4]Genentech, San Francisco, CA, USA. [5]Amgen Research, Cambridge, MA, USA. [6]Genomic Sciences, GlaxoSmithKline, Stevenage, UK. [7]Genomic Sciences, GlaxoSmithKline, Collegeville, PA, USA. [8]Bristol Myers Squibb, Princeton, NJ, USA. [9]Population Analytics, Janssen Research & Development, Spring House, PA, USA. [10]Centre for Genomics Research, Discovery Sciences, BioPharmaceuticals R&D, AstraZeneca, Cambridge, UK. [11]Biostatistics, Research and Development, Biogen, Cambridge, MA, USA. [12]Regeneron Genetics Center, Tarrytown, NY, USA. [13]External Science and Innovation Target Sciences, Worldwide Research, Development and Medical, Pfizer, Stockholm, Sweden. [14]Analytics and Data Sciences, Biogen, Cambridge, MA, USA. [15]Data Science Institute, Takeda Development Center Americas, Cambridge, MA, USA. [16]UK Biobank, Stockport, UK. [17]Alnylam Human Genetics, Discovery & Translational Research, Alnylam Pharmaceuticals, Cambridge, MA, USA. [18]Institute of Computational Biology, Helmholtz Zentrum München, German Research Center for Environmental Health, Neuherberg, Germany. [19]Department of Psychiatry and Behavioral Sciences, Duke University, Durham, NC, USA. [20]Bioinformatics Core, Weill Cornell Medicine-Qatar, Doha, Qatar. [21]Calico Life Sciences, San Francisco, CA, USA. [22]Department of Medicine, University of Melbourne, Austin Health, Melbourne, Victoria, Australia. [23]Takeda Development Center Americas, San Diego, CA, USA. [24]Immunology, Janssen Research & Development, Spring House, PA, USA. [25]Neuroscience Data Science, Janssen Research & Development, Cambridge, MA, USA. [26]These authors contributed equally: Joshua Chiou, Matthew Traylor, Christian Benner, Yi-Hsiang Hsu, Tom G. Richardson, Praveen Surendran,

Anubha Mahajan, Chloe Robins, Steven G. Vasquez-Grinnell, Liping Hou, Erika M. Kvikstad. [27]These authors jointly supervised this work: Joseph D. Szustakowski, Bradford W. Gibson, Melissa R. Miller, Christopher D. Whelan. *A list of authors and their affiliations appears at the end of the paper. ✉e-mail: bbsun92@outlook.com; christopherdwhelan@outlook.com

**Alnylam Human Genetics**

**Lucas D. Ward[17], Aimee M. Deaton[17], Carissa M. Willis[17] & Paul Nioi[17]**

A full list of members and their affiliations appears in the Supplementary Information.

**AstraZeneca Genomics Initiative**

**Oliver S. Burren[10], Bram Prins[10] & Slavé Petrovski[10,22]**

A full list of members and their affiliations appears in the Supplementary Information.

**Biogen Biobank Team**

**Benjamin B. Sun[1], Denis Baird[1], Chia-Yen Chen[1], Kyle L. Ferber[11] & Tinchi Lin[14]**

A full list of members and their affiliations appears in the Supplementary Information.

**Bristol Myers Squibb**

**Steven G. Vasquez-Grinnell[8,26], Erika M. Kvikstad[8,26] & Joseph D. Szustakowski[8,27]**

A full list of members and their affiliations appears in the Supplementary Information.

**Genentech Human Genetics**

**Anubha Mahajan[4,26], Rion K. Pendergrass[4] & Mark I. McCarthy[4]**

A full list of members and their affiliations appears in the Supplementary Information.

**GlaxoSmithKline Genomic Sciences**

**Tom G. Richardson[3,6,26], Praveen Surendran[6,26], Chloe Robins[7,26], Jonathan Davitte[7] & Robert A. Scott[6]**

A full list of members and their affiliations appears in the Supplementary Information.

**Pfizer Integrative Biology**

**Joshua Chiou[2,26], Åsa K. Hedman[13], Eric B. Fauman[2] & Melissa R. Miller[2,27]**

A full list of members and their affiliations appears in the Supplementary Information.

**Population Analytics of Janssen Data Sciences**

**Liping Hou[9,26] & Mary Helen Black[9]**

A full list of members and their affiliations appears in the Supplementary Information.

**Regeneron Genetics Center**

**Christopher E. Gillies[12], Hyun Ming Kang[12] & Lyndon J. Mitnaul[12]**

A full list of members and their affiliations appears in the Supplementary Information.

# Methods

## UKB participants

The UKB is a population-based cohort of approximately 500,000 participants aged 40–69 years recruited between 2006 and 2010. Participant data include genome-wide genotyping, exome sequencing, whole-body magnetic resonance imaging, electronic health record linkage, blood and urine biomarkers, and physical and anthropometric measurements. Further details are available online (https://biobank.ndph.ox.ac.uk/showcase/). All of the participants provided informed consent.

## UKB-PPP sample selection and processing

Details of UKB participant selection and sample handling are provided in the Supplementary Information.

## Proteomic measurement, processing and quality control

Details of the Olink proteomics assay, data processing and quality control are provided in the Supplementary Information. One protein (GLIPR1) had >80% of data failing quality control (99.4% failing quality control; Supplementary Table 3) and was excluded from analyses. We did not perform further NPX processing after the quality-control procedures described in the Supplementary Information. Each protein level was inverse-rank normalized, including NPX data below the LOD, before analyses and association testing.

## Non-genetic associations

For associations between age, sex and BMI, we used multiple linear regression with all three variables fitted in the same model along with technical factors: batch, UKB centres, UKB array type, UKB-PPP subcohort (randomly selected baseline/consortium/COVID-19 imaging participants), and 20 genetic principal components, along with the time between blood sampling and protein measurement. Interactions between age, sex and BMI were tested as scaled interaction terms with the same covariate adjustments.

For the association between protein levels and liver function enzymes log[ALT] (field 30620); log[AST] (field 30650); estimated glomerular filtration rate (eGFR) calculated using the combined creatinine-cystatin C equation from the CKD-EPI study[56], with relevant parameters obtained from fields 30700 (creatinine), 30720 (cystatin C), 21000 (ethnicity) in addition to age and sex; smoking status (field 20116); the top 20 most prevalent diseases (by 2 digit ICD10 code fields); and number of medications (field 137), regression models were individually fitted with age, sex and BMI along with technical factors as covariates.

## Proteomic prediction models

Proteomic prediction models were trained using 80% of the UKB-PPP data randomly subsetted as training. Least absolute shrinkage and selection operator (LASSO) models were trained for age, sex, BMI, AST, ALT, eGFR and ABO blood groups (genetic ascertainment of blood groups is described in the 'ABO blood group and FUT2 secretor status analysis' section) separately using glmnet (R package v.4.1-4)[57] to tune the lambda.1se parameter with tenfold cross validation for 100 lambdas between $10^{-5}$ and 1,000. For AST and eGFR models, we excluded AST and cystatin C, respectively, as the same proteins are either measured (AST) or used in deriving eGFR (cystatin C). Performance was evaluated in the held out 20% test data. Proteins with more than 20% missingness due to quality control were excluded in the predictor models, with the remainder of missing measurements mean-imputed.

## Genomic data processing

UKB genotyping and imputation (and quality control) were performed as described previously[7]. In addition to checking for sex mismatch, sex chromosome aneuploidy and heterozygosity checks, imputed genetic variants were filtered for INFO > 0.7 and chromosome positions were lifted to the hg38 build using LiftOver[58]. Participant ancestries were defined using the pan-UKBB definitions of genetic ancestry in the UKB return dataset 2442 (for example, "pop = EUR").

## Genetic association analyses

GWAS analyses were performed using REGENIE v.2.2.1 through a two-step procedure to account for population structure detailed previously[59]. In brief, the first step fits a whole-genome regression model for individual trait predictions based on genetic data using the leave one chromosome out (LOCO) scheme. We used a set of high-quality genotyped variants: MAF > 1%, MAC > 100, genotyping rate > 99%, Hardy–Weinberg equilibrium test $P > 10^{-15}$, <10% missingness and linkage-disequilibrium (LD) pruning (1,000 variant windows, 100 sliding windows and $r^2 < 0.8$). The LOCO phenotypic predictions were used as offsets in step 2, which performs variant association analyses using standard linear regression.

We limited genetic association analyses to variants with INFO > 0.7 and MAC > 50 to minimize spurious associations. For ancestry-specific analyses, we limited variants to INFO > 0.7 and MAC > 10 to maintain comparable MAF with the EUR-only analysis in view of the smaller sample sizes.

In the discovery cohort ($n = 34,557$), we included participants of European ancestry from batches 0–6, excluding the plates that were normalized separately, and batch 7 (COVID-19 imaging longitudinal samples and baseline samples showing increased variability and mixed with COVID-19 imaging samples). Participants who were not included in the discovery cohort were included in the replication cohort, which consisted of individuals of European ($n = 10,840$), African ($n = 931$), Central/South Asian ($n = 920$), Middle Eastern ($n = 308$) East Asian ($n = 262$) and admixed American ($n = 97$) ancestries.

Individual protein levels (NPX) were inverse-rank normalized before analysis including NPX data below the LOD. For the discovery cohort, association models included the following covariates: age, $age^2$, sex, age × sex, $age^2$ × sex, batch, UKB centre, UKB genetic array, time between blood sampling and measurement and the first 20 genetic principal components. The covariates in the replication and full cohort along with genetic ancestry-specific analyses also included whether the participant was preselected, either by the UKB-PPP consortium members or as part of the COVID-19 repeat-imaging study.

To ensure reproducibility of the analysis protocol, the same proteomic quality-control and analysis protocols were independently validated across two additional sites using the same initial input data on three proteins measured across multiple protein panels (CXCL8, IL-6, TNF, IDO1, LMOD1, SCRIB).

## Definition and refinement of significant loci

We used a conservative multiple-comparison-corrected threshold of $P < 1.7 \times 10^{-11}$ ($5 \times 10^{-8}$ adjusted for 2,923 unique proteins) to define significance. We defined primary associations through clumping ±1 Mb around the significant variants using PLINK[60], excluding the HLA region (chromosome 6: 25.5–34.0 Mb), which is treated as one locus owing to complex and extensive LD patterns. Overlapping regions were merged into one, deeming the variant with the lowest $P$ value as the sentinel primary associated variant. To determine regions associated with multiple proteins, we iteratively, starting from the most significant association, grouped together regions associated with proteins containing the primary associations that overlapped with the significant marginal associations for all proteins ($P < 1.7 \times 10^{-11}$). In cases in which the primary associations contained marginal associations that overlapped across multiple groups, we grouped together these regions iteratively until convergence.

## Variant annotation

Annotation was performed using Ensembl Variant Effect Predictor (VEP), WGS Annotator (WGSA) and UCSC Genome Browser's variant annotation integrator (http://genome.ucsc.edu/cgi-bin/hgVai).

The gene/protein consequence was based on RefSeq and Ensembl. We reported exon and intron numbers that a variant falls in as in the canonical transcripts. For synonymous mutations, we estimated the rank of genic intolerance and consequent susceptibility to disease based on the ratio of loss of function. For coding variants, SIFT and PolyPhen scores for changes to protein sequence were estimated. For non-coding variants, transcription-factor-binding sites, promoters, enhancers and open chromatin regions were mapped to histone marks chip-seq, ATAC-seq and DNase-seq data from The Encyclopedia of DNA Elements Project (ENCODE, https://www.encodeproject.org) and the ROADMAP Epigenomics Mapping Consortium (https://www.ncbi. nlm.nih.gov/geo/roadmap/epigenomics/). For intergenic variants, we mapped the 5′ and 3′ nearby protein-coding genes and provided distance (from the 5′ transcription start site of a protein-coding gene) to the variant. The combined annotation dependent depletion score (https://cadd.gs.washington.edu) was estimated for non-coding variants. An enrichment analysis hypergeometric test was performed to estimate enrichment of the associated pQTL variants in specific consequence or regulatory genomic regions.

## Cross-referencing with previously identified pQTLs

To evaluate whether the pQTLs in the discovery set were previously undescribed, we used a list of published pQTL studies (http://www. metabolomix.com/a-table-of-all-published-gwas-with-proteomics/) and the GWAS Catalog to build a comprehensive list of previously published pQTL studies. A total of 34 studies was included (Supplementary Information). Using a $P$-value threshold of $1.7 \times 10^{-11}$, we identified the sentinel variants and associated protein(s) in the previously published studies and queried those against our discovery pQTLs. If a previously associated sentinel variant–protein pair fell within a 1 Mb window of the discovery set pQTL sentinel variant for the same protein and had an $r^2 \geq 0.8$ with any significant SNPs in the region, it was considered a replication.

## Identification and fine mapping of independent signals

We used sum of single-effects regression (SuSiE, v.0.12.6)[61] to identify and fine-map independent signals using individual-level genotypes and protein-level measurements from discovery-set participants. Our inputs for SuSiE were mean-centred and unit variance genotype and phenotype residuals accounting for the same covariates as for the marginal association analysis. We subtracted REGENIE LOCOs from the phenotype residuals to account for polygenic effects and sample relatedness.

To create dynamic test regions that accounted for potential long-range LD, we performed a two-step clumping procedure using PLINK with the parameters (1) --clump-r2 0.1 --clump-kb 10000 --clump-p1 1.7x10$^{-11}$ --clump-p2 0.05 on the marginal association summary statistics and (2) --clump-kb 500 on the results of the first clumping step. For each clump, we extended the coordinates of the left- and right-most variants to a minimum size of 1 Mb, merged overlapping clumps and defined these as the test regions.

For each test region, we applied SuSiE regression using the initial parameters min_abs_corr=0.1, L = 10, max_iter=1000. For test regions in which SuSiE found the maximum number of independent credible sets, which was initially set at L = 10, we incremented L by 1 until no additional credible sets were detected. We applied a post hoc filter to remove credible sets in high LD with another credible set in the same region (lead variants $r^2 > 0.8$). For regions with multiple credible sets, we assessed statistical independence by performing multiple linear regression with the most probable variants for each credible set and the same genotype and phenotype residuals.

## Heritability analysis

We estimated the SNP-based heritability as a sum of variance explained from the independent pQTLs through the SuSiE analyses for each

protein at each loci (pQTL component) and the polygenic component using the genome-wide SNPs excluding the pQTL regions of each protein. The polygenic component, which mostly likely satisfies the polygenic model of small genetic contributions across the genome, was estimated using LD-score regression[62]. We used the discovery-cohort associations to maintain consistent LD used in SuSiE and LD-score regression based on EUR.

## Pathway enrichment and protein interactions

For pleiotropic pQTL loci and multiple associated *trans* pQTL proteins, gene-set enrichment analyses were performed by ingenuity pathway analysis to identify enrichment of biological functions relevant to cell-to-cell signalling, cellular development, development and process. Gene pathways and networks annotated based on STRING-db and KEGG pathway databases were also used for enrichment analyses. Hypergeometric tests were performed to estimate statistical significance and hierarchical clustering trees and networks summarizing overlapping terms/pathways were generated. To correct for multiple testing, the false discovery rate (FDR) was estimated. FDR < 0.01 was considered to be statistically significant.

To test if *trans* pQTL loci contained at least one gene (within 1 Mb of the *trans* pQTL) that encoded proteins interacting with the tested protein, we used the curated protein interaction database: Human Integrated Protein-Protein Interaction Reference (HIPPIE)[33] release v.2.3 (http://cbdm-01.zdv.uni-mainz.de/~mschaefer/hippie/download. php). To get an estimate of background protein interactions by chance, we permuted the proteins against the sentinel pQTLs ($n$ = 100 times) and tested for protein interactions in HIPPIE.

## Subsampling analysis

To estimate how the number of associations scaled with sample size, we took random samples without replacement of [100, 250, 500, 1,000, 5,000, 10,000, 15,000, 20,000, 25,000, 30,000, 35,000, 40,000, 45,000 and 50,000] from the full cohort, then performed the association testing and examined the proteomic variance explained in the exact same manner as for the main analyses described above. We also examined how associations scaled with the number of proteins measured, accounting for the likelihood that additional proteins measured would be of decreasing abundance in plasma. We performed random subsampling of [100, 250, 500, 1,000, 1,500, 2,000, 2,500, 2,800] proteins starting preferentially from the most expected abundant dilution, a priori, (1:100,000) to the least abundant dilution (1:1). We also performed multiple samples ($n$ = 10) to check consistency and stability of subsampling results across runs.

## Sensitivity analyses

The variables for sensitivity analyses were chosen a priori to avoid post hoc biases.

## Effects of blood cell counts

We investigated the effect of blood cell composition on the genetic association with plasma proteins through sensitivity analyses of pQTLs from the discovery analyses. The top hits from the discovery analyses were reanalysed adjusting for the following blood cell covariates: monocyte count; basophil count; lymphocyte count; neutrophil count; eosinophil count; leukocyte count; platelet count; haematocrit percentage; and haemoglobin concentration. These blood cell covariates were selected to represent blood cell composition due to their common clinical use. Before the analyses, we followed the previously described methods[63] to exclude blood cell measures from individuals with extreme values or relevant medical conditions. Relevant medical conditions for exclusion included pregnancy at the time the complete blood count was performed, congenital or hereditary anaemia, HIV, end-stage kidney disease, cirrhosis, blood cancer, bone marrow transplant and splenectomy. Extreme measures were defined as

leukocyte count, >200 × 10^9 per l or >100 × 10^9 per l with 5% immature reticulocytes; haemoglobin concentration, >20 g dl^{-1}; haematocrit, >60%; and platelet count, >1,000 × 10^9 per l. Following these exclusions and quality control, genetic analyses of the sentinel variant–protein associations adjusted for blood cell covariates were performed using the same approach as described for the main analysis.

We further tested whether blood cell composition is partially or fully mediating variant–protein associations (genotype → blood cell measure → protein) for genetic associations that were significant within the discovery ($P < 1.7 × 10^{-11}$) and not in the sensitivity analyses ($P > 1.7 × 10^{-11}$). For each variant–protein association, we first identified the blood cell phenotypes that were associated with protein levels at $P < 1.7 × 10^{-11}$ within a multivariable linear regression model including blood cell phenotypes as the predictors, protein as the outcome and adjusted for all other covariates included in the discovery analysis. We then confirmed whether there was an association between the genetic variant (dosage) and each of the blood cell phenotypes (genotype → blood cell) and between blood cell phenotype and the protein (blood cell → protein) before testing for mediation. In the final test, we compared the strength of associations, genotype → protein, to that of the genotype → protein in a multivariable model (protein - dosage + blood cell phenotype + discovery covariates) to establish whether the variant–protein association is either fully ($P > 0.01$) or partially ($P < 1.7 × 10^{-11}$) mediated by the blood cell phenotype.

### Effects of BMI
We investigated the effect of BMI on the genetic association with plasma proteins through sensitivity analyses of pQTLs from the discovery analyses. The primary associations from the discovery analyses were reanalysed using the same approach as described for the main analysis including BMI (field: 21001) as an additional covariate.

### Effects of season and amount of time fasted at blood collection
To assess the effects of season and amount of time fasted at blood collection on variant associations with protein levels, we reanalysed all sentinel pQTLs identified in the main discovery analyses including season and fasting time as two additional covariates. Blood collection season (summer/autumn (June to November) versus winter/spring (December to May)) was defined on the basis of the blood collection date and time (field: 3166). Participant-reported fasting time was derived from field 74 and was standardized (Z-score transformation) before analysis.

### Co-localization analyses
We investigated evidence of shared genetic associations between variants directly affecting circulating protein expression levels and tissue-level gene expression using the coloc with SuSiE framework[61]. For genes with significant results in the marginal eQTL associations, we applied SuSiE regression using individual-level genotype and phenotype data for 49 tissues from GTEx[31] v.8 to define independent eQTL signals, using the same samples, variants, covariates, ±1 Mb window around TSS and normalized gene expression matrices as the GTEx consortium flagship paper. We then conducted pairwise colocalization analyses between independent *cis* pQTL and eQTL signals using default priors and considered a posterior probability of colocalization (PP.H4) ≥ 0.8 as shared genetic associations. For pairs of colocalized pQTL–eQTL signals, we used the top variants of each pQTL signal to compare the directionality of conditional effect estimates on protein and gene expression.

For colocalization with COVID-19 loci, the top loci reported by the COVID-19 Host Genetics consortium (https://app.covid19hg.org/variants) were updated with estimates from the R7 summary results (https://www.covid19hg.org/results/r7/) for hospitalized cases of COVID-19 and reported COVID-19 infections compared with population controls. We used HyprColoc[64] with a region association threshold of 0.8 to perform multi-trait colocalization across all significant proteins with each disease loci.

### ABO blood group and FUT2 secretor status analysis
ABO blood group was imputed through the genetic data using three SNPs in the *ABO* gene (rs505922, rs8176719 and rs8176746) according to the blood-type imputation method in the UKB (https://biobank.ndph.ox.ac.uk/ukb/field.cgi?id=23165), developed previously[65–68]. FUT2 secretor status was determined by the inactivating mutation (rs601338), with genotypes GG or GA as secretors and AA as non-secretors. Interaction term between blood group (O as the reference group) and secretor status was tested adjusting for the same covariates as in the main pQTL analyses for each protein separately. A multiple-testing threshold of $P < 1.7 × 10^{-5}$ (0.05/2,923 proteins) for the interaction terms was used to define statistically significant interaction effects.

### Enrichment for gene expression in tissues
Tissue enrichment of associated proteins was tested using the Tissue-Enrich R package (v.1.6.0)[69], using the genes encoding proteins on the Olink panel as the background. For enrichment in human genes, we used the RNA dataset from the Human Protein Atlas[70] using all genes that were found to be expressed within each tissue, whereas, for orthologous mouse genes, we used data from a previous study[71]. The enrichment *P*-value thresholds were corrected for multiple comparisons based on the number of tissues tested where applicable ($n = 35$ in human and $n = 17$ in mouse tissues).

### PCSK9 Mendelian randomization
**Instrument selection and outcomes.** Instruments to proxy for altered PCSK9 abundance were generated using variants associated in *cis* (within 1 Mb of the PCSK9 gene-coding region) at genome-wide significance ($P < 5 × 10^{-8}$) to minimize pleiotropic effects. We performed LD clumping to ensure that SNPs were independent ($r^2 < 0.01$) by using in-sample UKB participants. We removed SNPs with a *F*-statistic of less than 10 to avoid weak instrument bias.

Outcomes of interest were measurements of cholesterol, including low-density lipoprotein cholesterol, high-density lipoprotein cholesterol, triglycerides and total cholesterol; coronary heart disease and myocardial infarction; ischaemic stroke large artery atherosclerosis and small-vessel subtypes. Data for these outcomes were extracted from the OpenGWAS project[72,73]. *PCSK9* pQTL effects were harmonized to be on the same effect allele. If the variant was not present in the outcome dataset, we searched for a proxy SNP ($r^2 > 0.8$) as a replacement, if available.

**Mendelian randomization analysis.** We performed two-sample Mendelian randomization on the harmonized effects to estimate the effect of genetically proxied PCSK9 abundance on genetic liability to the outcomes of interest. We estimated the effects for each individual variant using the two-term Taylor series expansion of the Wald ratio and the weighted delta inverse-variance weighted method to meta-analyse the individual SNP effects to estimate the combined effect of the Wald ratios. Results from the Mendelian randomization analyses were analysed using standard sensitivity analyses. We used Steiger filtering to provide evidence of whether the estimated effect was correctly orientated from PCSK9 abundance to the outcome and not due to reverse causation.

### Inclusion and ethics statement
The inclusion and ethics standards have been reviewed where applicable.

### Reporting summary
Further information on research design is available in the Nature Portfolio Reporting Summary linked to this article.

## Data availability

Proteo-genomic results and summary association data are available through an interactive portal (http://ukb-ppp.gwas.eu). Underlying NPX measures are available through the UK Biobank Research Analysis Portal (https://www.ukbiobank.ac.uk/enable-your-research). UKB has catalogued the dataset in Category 1839, under 'Field 30900', described in greater detail online (https://biobank.ndph.ox.ac.uk/showcase/label.cgi?id=1839).

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

**Acknowledgements** We thank the participants, contributors and researchers of the UKB for making data available for this study, with special thanks to L. Carson, J. Busby, N. Allen and R. Collins for making the study possible; the members of the research & development leadership teams at the 13 participating UKB-PPP member companies (Alnylam Pharmaceuticals, Amgen, AstraZeneca, Biogen, Bristol Myers Squibb, Calico, Genentech, GlaxoSmithKline, The Janssen Pharmaceutical Companies of Johnson & Johnson, Novo Nordisk, Pfizer, Regeneron and Takeda) for funding the study; the legal and business development teams at each company for overseeing the contracting of this complex, precompetitive collaboration, with particular thanks to E. Olson of Amgen, A. Walsh of GSK and F. Middleton of AstraZeneca; M. Koprulu for compiling regional sentinel variants from genome-wide summary statistics referred to in her recent paper[23], facilitating a more comprehensive cross-referencing between our study and previously published pQTL studies; H. McLaughlin for her project management support; and the members of the team at Olink Proteomics (P. Pettingill, K. Diamanti, C. Lawley, L. Jung, I. Grundberg and J. Heimer) for their consistent logistic support throughout the project and creating the Olink workflow diagram, with special thanks to E. Mills for co-championing the project and leading internal activities at Olink. This research has been conducted using the UKB resource under approved application numbers 65851, 20361, 26041, 44257, 53639 and 69804.

**Author contributions** Study conceptualization, project coordination and consortium leadership: C.D.W. Study design and methodology: B.B.S., C.D.W., C.B., J.C., L.H., Y.-H.H., E.M.K., A.M., T.G.R., C.R., P.S., M.T., O.S.B., J.D., K.L.F., C.E.G., Å.K.H., S.H., T.L., R.M., R.K.P., B.P., S.G.V.-G., L.D.W., C.M.W., M.H.B., H.M.K., E.N.S., J.D.S., B.W.G. and M.R.M. Proteomic data quality control: B.B.S., K.L.F. and T.L. Phenotype harmonization: B.B.S., T.L., K.L.F., L.B.-G., S.W. and C.P. Analysis: B.B.S., C.D.W., C.B., J.C., L.H., Y.-H.H., E.M.K., A.M., T.G.R., C.R., P.S., M.T., O.S.B., J.D., K.L.F., C.E.G., Å.K.H., S.H., T.L., R.M., R.K.P., B.P., S.G.V.-G., L.D.W., C.M.W., M.H.B., H.M.K., E.N.S., J.D.S., B.W.G., M.R.M. and E.B.F. Genetic association analyses: B.B.S. Independent replication of genetic analyses: A.M., C.B. and E.M.K. Mendelian randomization: D.B. and C.-Y.C. Fine-mapping analyses: J.C. Epitope mapping analysis: A.M. and C.B. Co-localization with eQTLs: T.G.R., M.T. and J.C. Sensitivity analysis: A.M., C.B., C.R., P.S. and L.H. Variant annotation: Y.-H.H. Writing first draft of the manuscript: B.B.S. and C.D.W. Development of the interactive data portal: K.S., G.K., M.A., N.L. and M.A.W. Writing the second draft of manuscript: B.B.S., A.M., M.H.B., S.G.V.-G., Å.K.H., A.M.D., S.P., Y.-H.H., B.W.G., S.S., J.D.S., C.R., P.S., E.B.F., L.D.W., C.M.W., E.M.K., J.M.M.H., P.N., H.M.K., C.E.G., E.N.S., L.B.-G., S.W., M.R.M. and C.D.W. All of the authors reviewed the manuscript.

**Competing interests** L.D.W., P.N., C.M.W. and A.M.D. are employees and/or stockholders of Alnylam. Y.-H.H. and B.W.G. are employees and/or stockholders of Amgen. S.P., O.S.B. and B.P. are employees and/or stockholders of AstraZeneca. B.B.S., T.L., K.L.F., D.B. and C.-Y.C. are employees and/or stockholders of Biogen. E.M.K., J.D.S. and S.G.V.-G. are employees and/or stockholders of Bristol Myers Squibb. M.C., A.R., A.S. and E.M. are employees and/or stockholders of Calico. R.K.P., M.I.M., A.M. and C.B. are employees of Genentech and holders of Roche stock. C.R., P.S., R.A.S., T.G.R. and J.D. are employees and/or stockholders of GlaxoSmithKline. M.H.B., L.H., D.M.W. and C.D.W. are employees and/or stockholders of Janssen Research & Development. J.M.M.H., S.H. and M.T. are employees and/or stockholders of Novo Nordisk. Å.K.H., E.B.F., J.C. and M.R.M. are employees and/or stockholders of Pfizer. H.M.K., L.J.M. and C.E.G. are employees and/or stockholders of Regeneron. E.N.S., S.S. and R.M. are employees and/or stockholders of Takeda. L.B.-G., C.P. and S.W. are employees of the UK Biobank. The other authors declare no competing interests.

**Additional information**
**Correspondence and requests for materials** should be addressed to Benjamin B. Sun or Christopher D. Whelan.

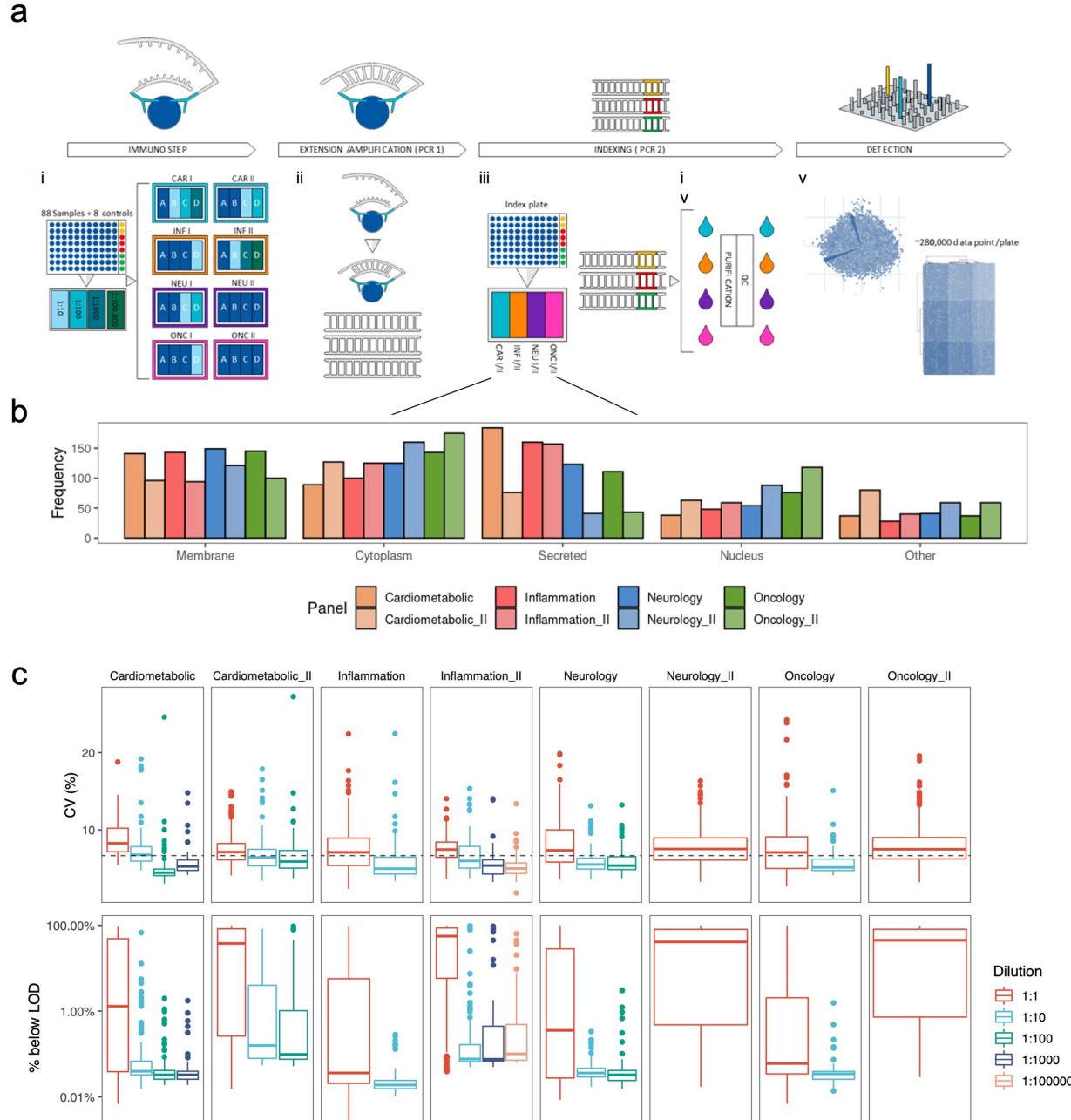

**Extended Data Fig. 1 | Summary of the Olink Explore proteomics assay.**
(a) Summary of the Olink proteomic assay workflow. (i) Assays are run in a
96-well format, each plate consists of 88 UKB samples and 8 external control
samples in column 12: sample controls (yellow) are used to determine precision
within and between plates, triplicate negative controls samples (red) set the
limit of detection (LOD) and triplicate plate controls (green) are used to
standardize protein levels within a plate. The Explore 3072 product consists of
eight 384-plex panels; Cardiometabolic (CAR) I and II, Inflammation (INF) I and
II, Neurology (NEU) I and II and Oncology (ONC) I and II, and each panel consists
of 4 abundance blocks, with plasma sample run 1:1 or diluted 1:1 (least expected
abundance), 1:10, 1:100, 1;1000 and 1:100,000. (ii) Extension and amplification
step: only matched PEA probes bind to their respective target and via PCR (PCR1)
generate dsDNA amplicons, containing assay information. (iii) Indexing: all
amplicons for a given sample in a single panel are pooled and unique index
primers are added and are integrated into the amplicon via PCR (PCR2).
(iv) All amplicons for all samples within a panel are combined to generate four
sequencing libraries; the libraries are purified and quality controlled before
(v) detection and being sequenced on an Illumina Novaseq 6000 instrument
generating ~280,000 data points per sample plate (b) Cell compartment
distribution of measured proteins by protein panel. (c) Boxplot of coefficients
of variation (CVs) and % of samples with measurements below LOD by dilutions.
Each box plot presents the median, first and third quartiles, with upper and
lower whiskers representing 1.5x inter-quartile range above and below the third
and first quartiles respectively; n = 2,941 independent protein analytes.

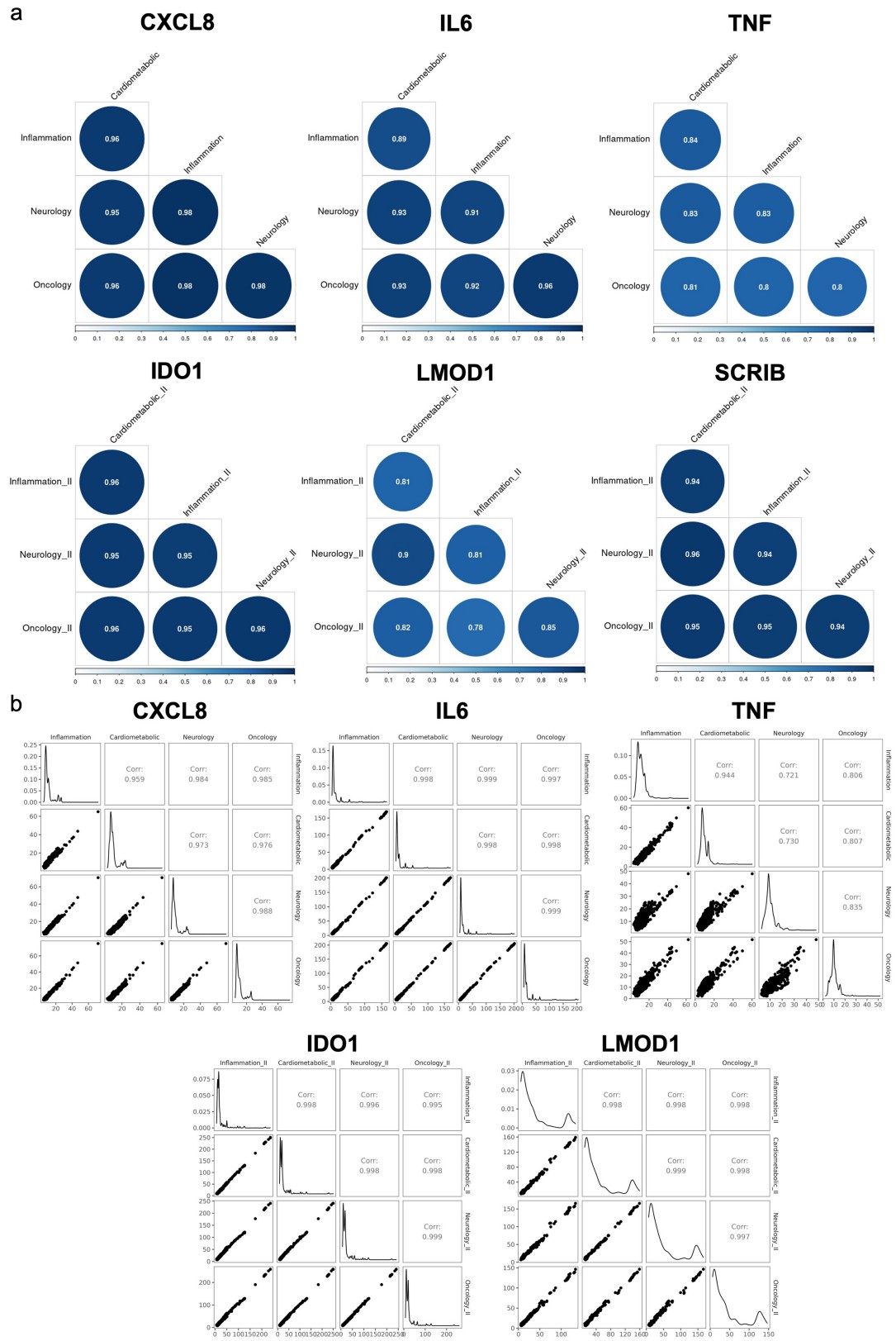

**Extended Data Fig. 2 | (a) Phenotypic correlation (Pearson's *r*) between the same protein targets (CXCL8, IL6, TNF, IDO1, LMOD1, SCRIB) measured across protein panels.** (b) Correlation (Pearson's *r*) of significant genetic associations ($p < 1.7 \times 10^{-11}$) between the same protein targets.

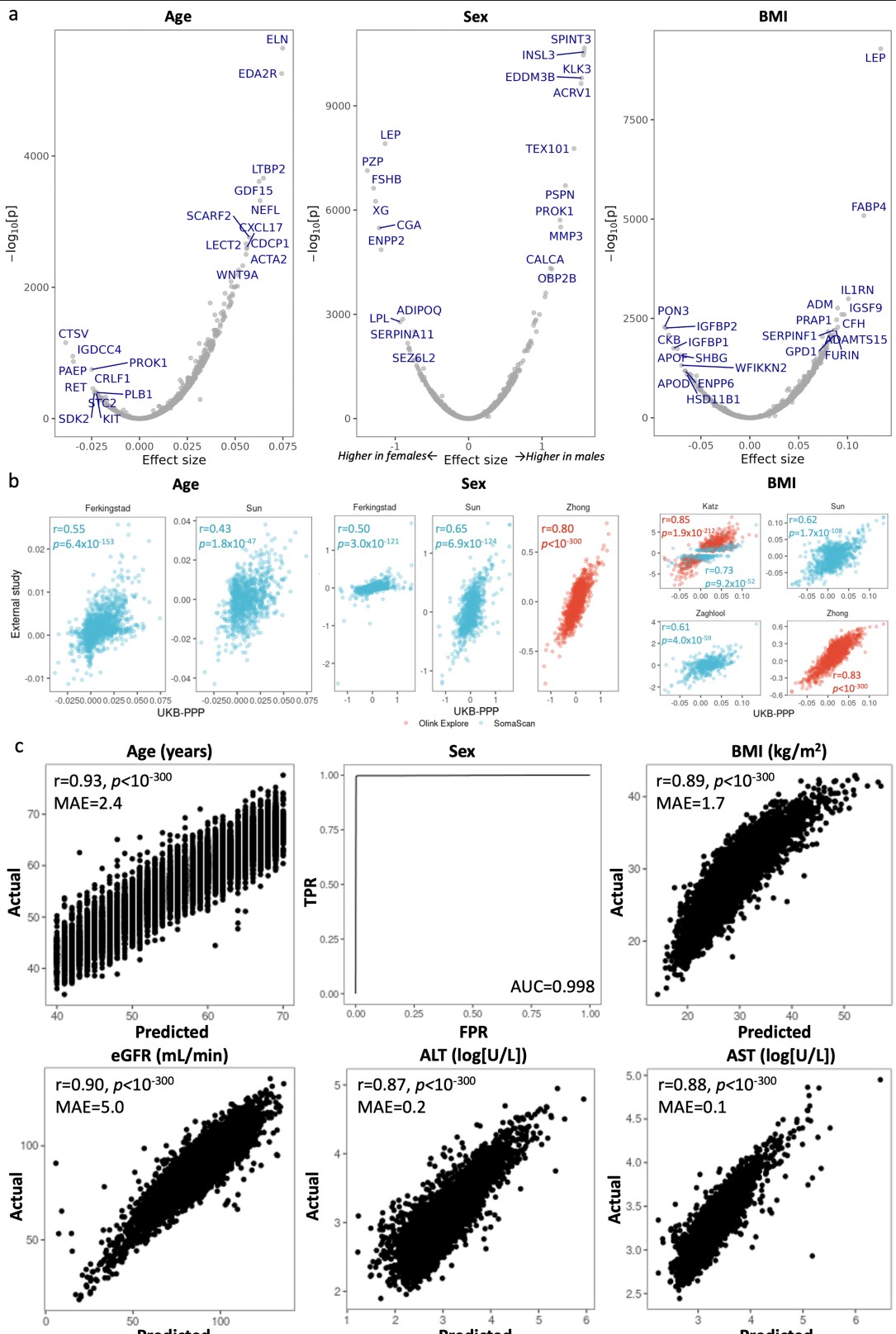

**Extended Data Fig. 3 | (a) Volcano plot of associations with age, sex and BMI.** Top 10 proteins with the largest positive and negative associations are labelled. *P*-values (two-sided, unadjusted) derived from multivariable linear regression. (b) Comparison of effect sizes between UKB-PPP and published multiplex proteomic studies for protein associations with age, sex and BMI.

(c) Performance of trained proteomic predictor models against true values in a held-out test data set. (b) and (c), *p*-values (unadjusted) for Pearson's correlation test (two-sided). *r: Pearson's correlation coefficient. MAE: mean absolute error, eGFR: estimated glomerular filtration rate. ALT: alanine aminotransferase. AST: aspartate aminotransferase.*

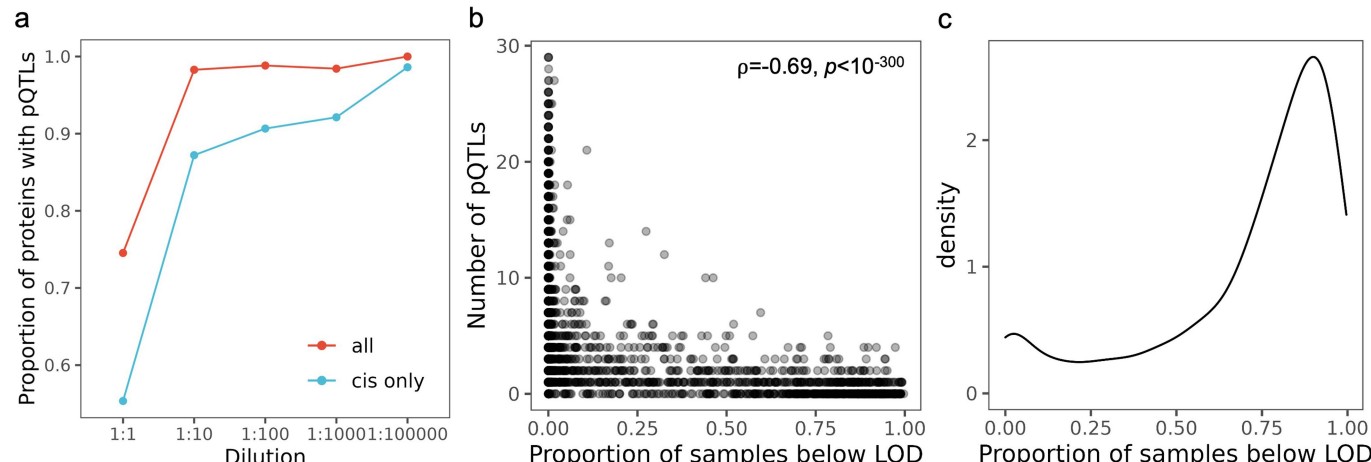

**Extended Data Fig. 4 | (a) Proportion of proteins with pQTLs across different dilution sections.** (b) Comparison of the number of pQTLs *vs* the proportion of samples with measurements below LOD for each protein. *P*-values (unadjusted) for Spearman's correlation test (two-sided). (c) Density plot of the proportion of samples with measurements below LOD for proteins with no significant pQTLs ($p < 1.7 \times 10^{-11}$). LOD: limit of detection. $\rho$: *Spearman's correlation coefficient*.

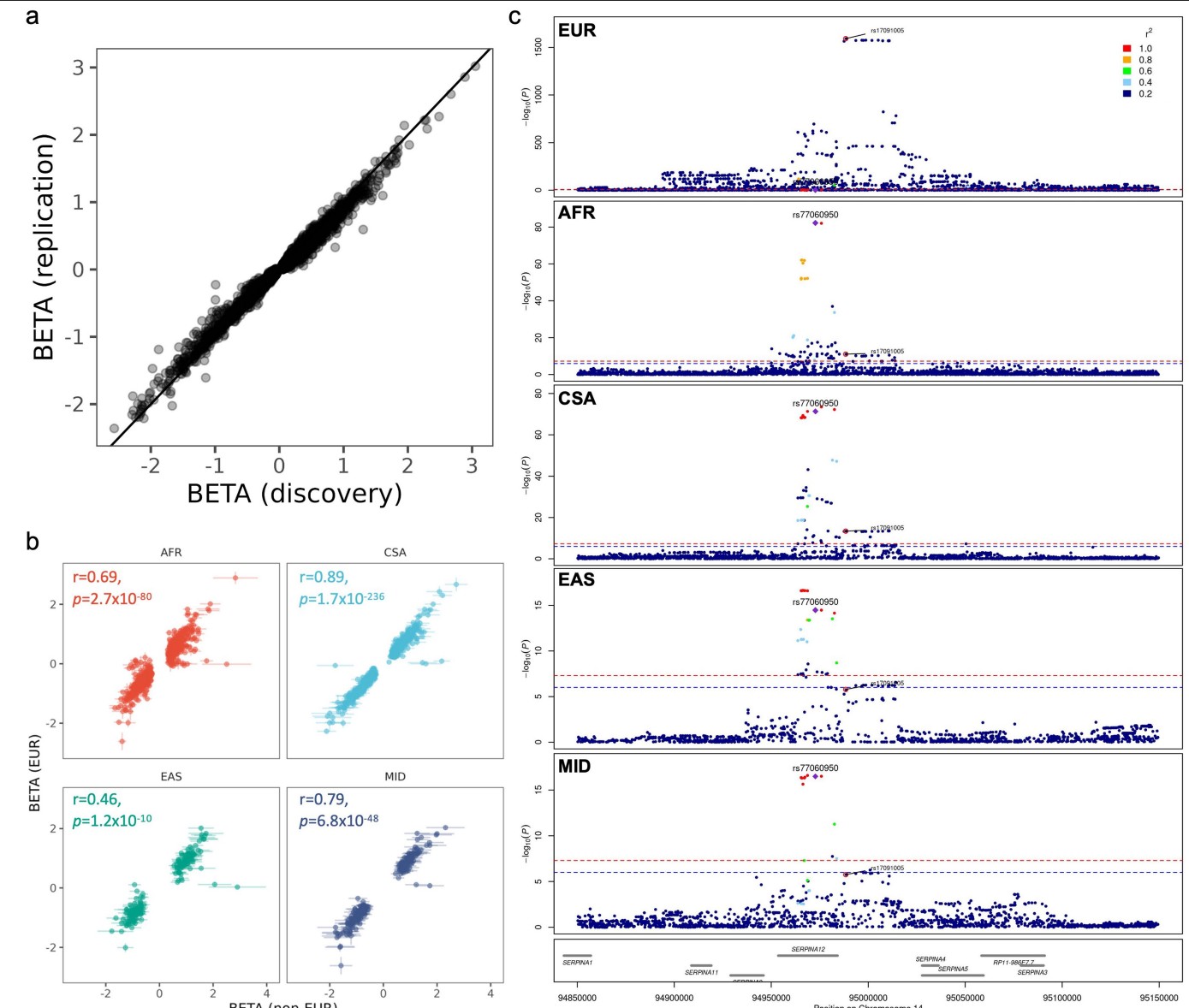

**Extended Data Fig. 5 | (a) Comparison of effect sizes between discovery and replication cohorts.** (b) Comparison of effect sizes between significant non-EUR ancestry specific pQTLs and EUR derived pQTLs. Error bars indicate 99% confidence intervals around the beta estimates. *P*-values (unadjusted) derived from Pearson's correlation test (two-sided) on |beta| over n = 785 (AFR), 732 (CSA), 179 (EAS), 227 (MID) pQTL associations. (c) Regional association plot of the SERPINA12 *cis* association locus across ancestries. *P*-values derived from REGENIE regression GWAS (two-sided, unadjusted).

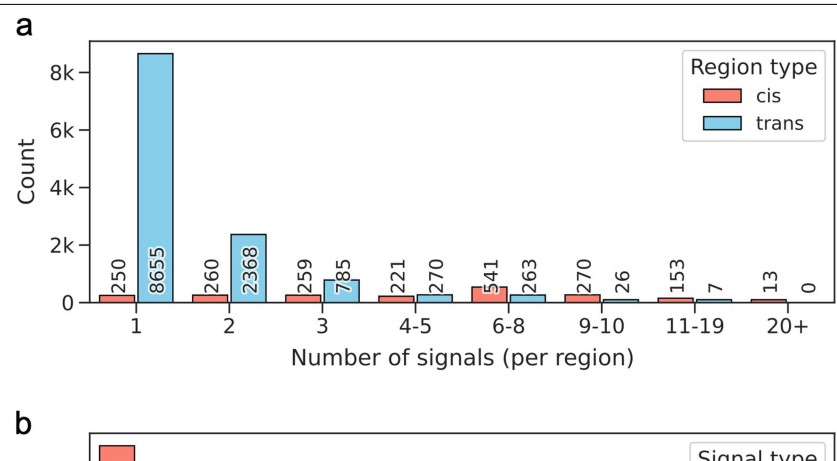

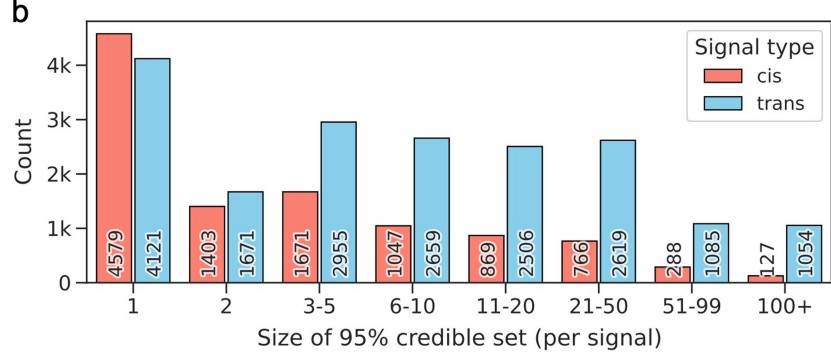

**Extended Data Fig. 6 | Number of independent signals per region (a) and size of 95% credible set per signal (b).** Results are categorized by *cis* (red) and *trans* (blue) associations.

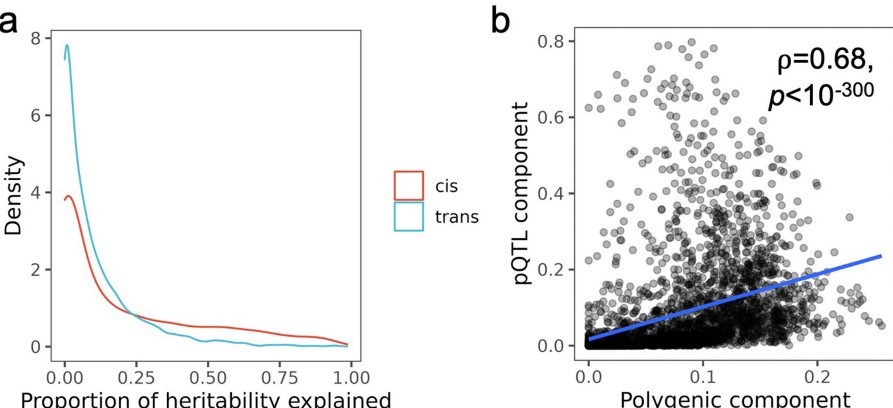

**Extended Data Fig. 7 | (a) Density plot of proportion of total heritability explained by primary *cis* and *trans* associations.** (b) Scatterplot with overlaid regression line of the pQTL component (variance explained by sentinel primary pQTLs) *vs* the polygenic component (genome-wide SNP heritability excluding pQTL regions). *P*-values (unadjusted) for Spearman's correlation test (two-sided). *ρ: Spearman's correlation coefficient.*

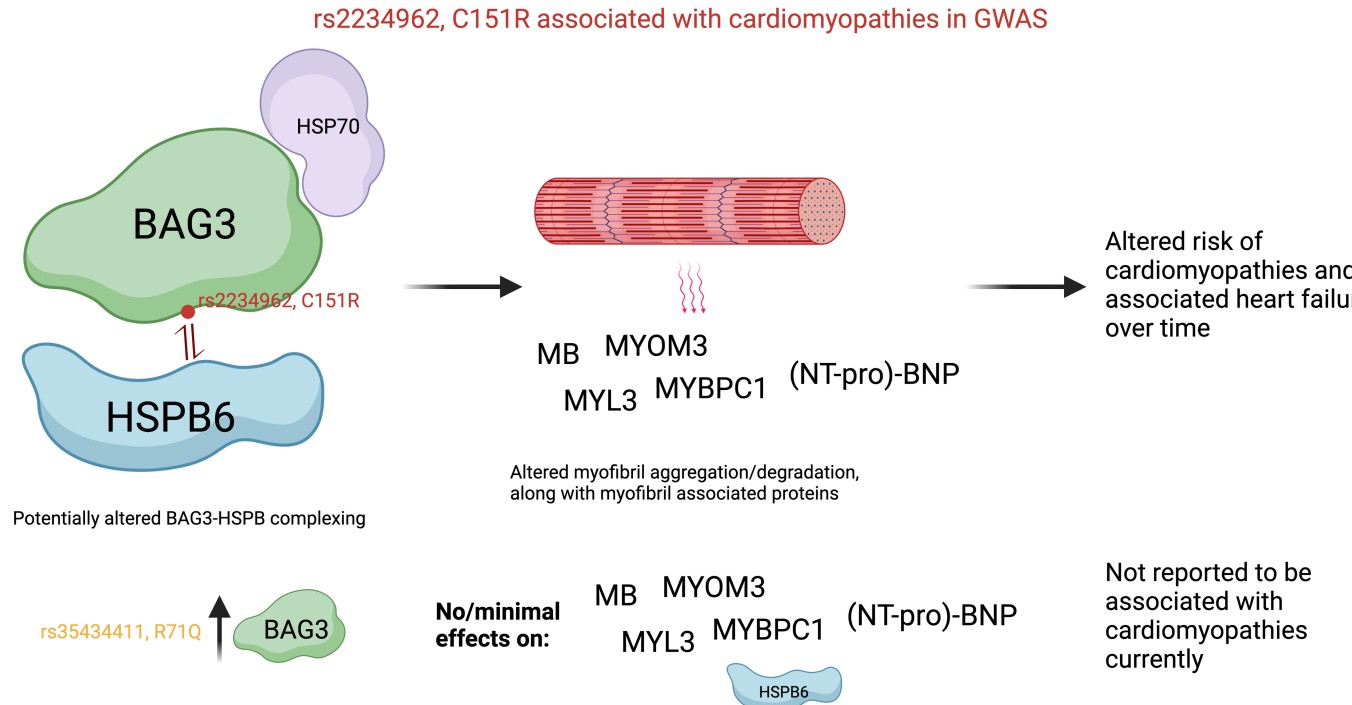

**Extended Data Fig. 8 | Schematic of a potential pathway linking a *BAG3* cardiomyopathy associated missense variant (rs2234962, Cys151Arg) to BAG3-HSBP complexing and downstream effects in cardiac muscle.** Figure created with BioRender.com.

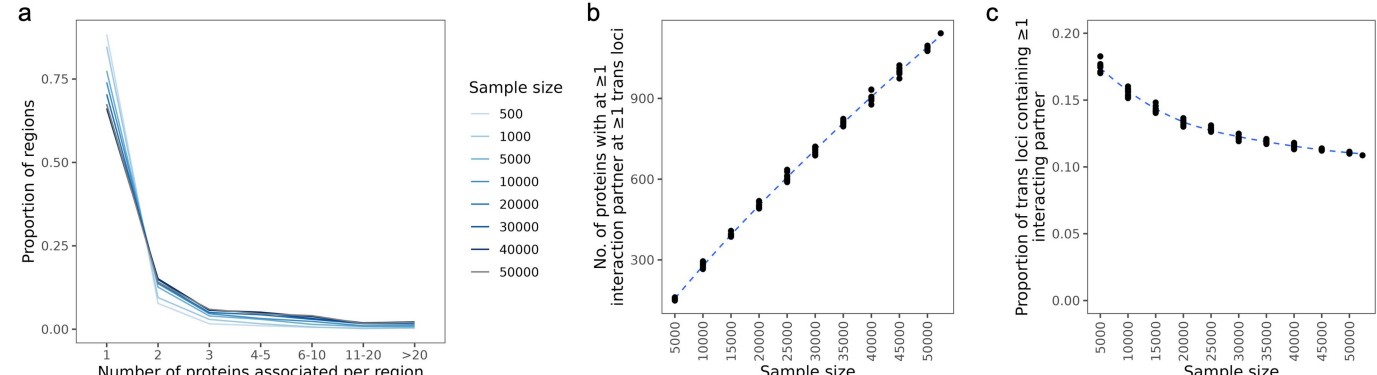

**Extended Data Fig. 9 | (a) Number of proteins associated per genomic region at different sample sizes.** (b) Number of proteins with at least one interaction partner locus (gene product at the *trans* locus that interacts with the protein tested) in at least one of the associated *trans* loci. (c) Proportion of *trans* associations containing at least one interaction partner with the protein tested.

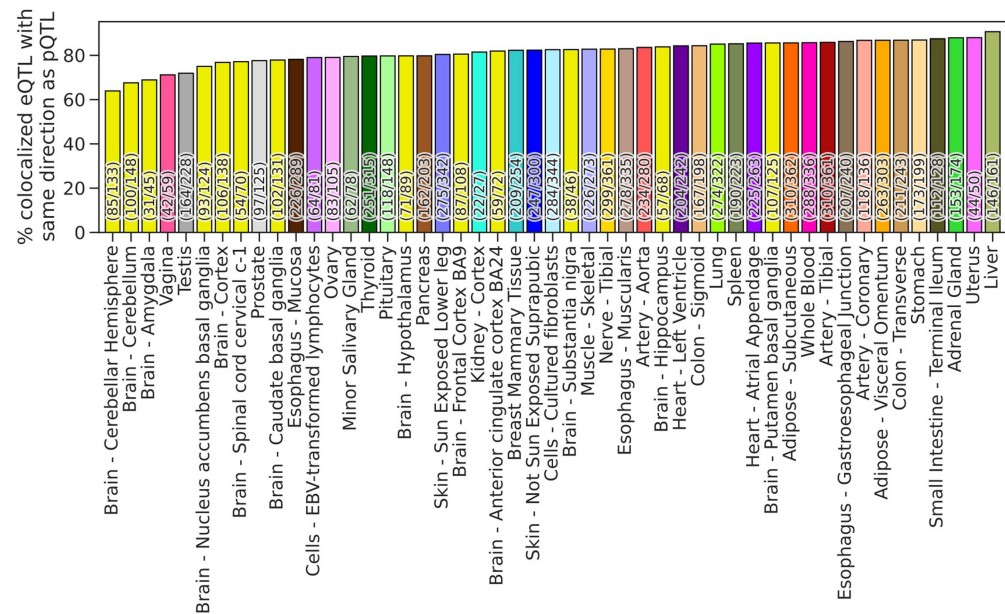

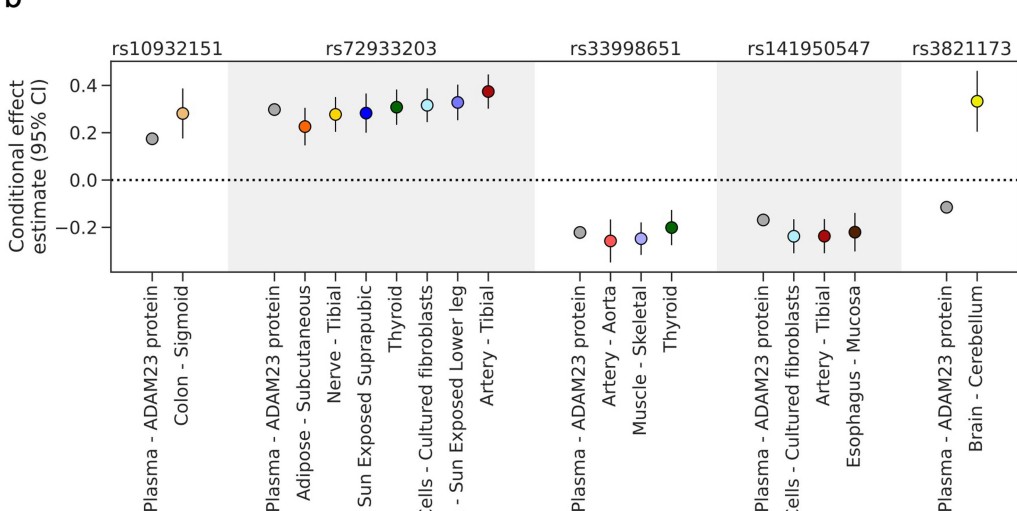

**Extended Data Fig. 10 | Directional concordance of colocalized eQTL signals.**
(a) Percentage of directionally concordant eQTL signals among those colocalized with a pQTL signal, for each GTEx tissue. (b) Conditional effect size estimates (centre point) and 95% confidence intervals (error bars) for top variants of ADAM23 pQTL signals and colocalized eQTL signals (rs33998651 was used as a proxy for rs139001108, which was not tested in GTEx).

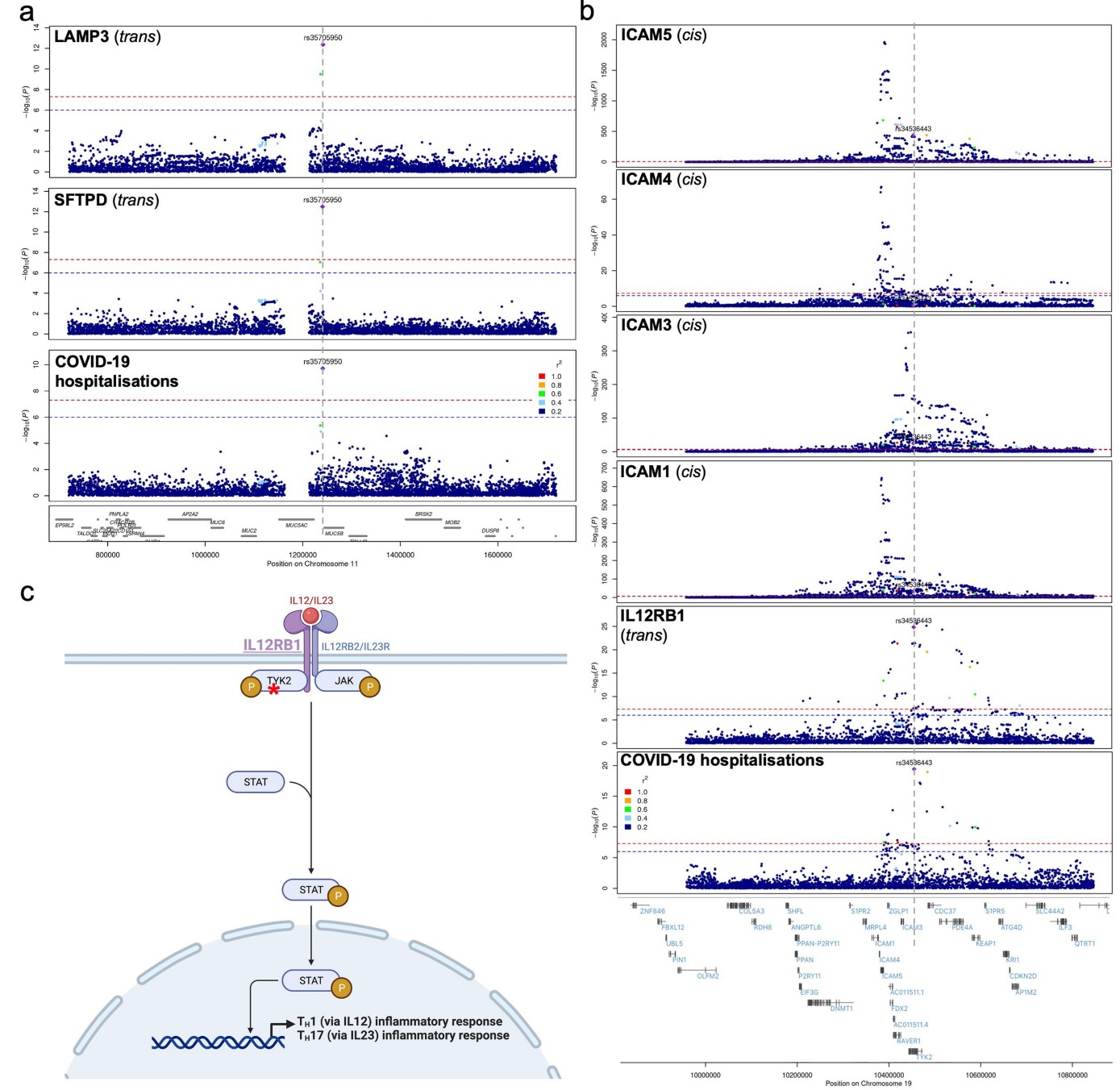

**Extended Data Fig. 11 | Stacked regional association plots between COVID loci and pQTLs.** (a) Regional association between COVID-19 locus at *MUC5B* and SFTPD, LAMP3 *trans* pQTLs (b) Regional association between COVID-19 locus at *TYK2* and colocalized IL12RB1 *trans* pQTL, in addition to the *cis* pQTLs of ICAM-1,3,4 and 5 in close proximity. (a) and (b) *P*-values derived from REGENIE regression GWAS (two-sided, unadjusted). (c) The IL12R-TYK2 inflammatory response signalling schematic with red asterisk indicating the *trans* pQTL for IL12RB1 in *TYK2*. Figure created with BioRender.com.

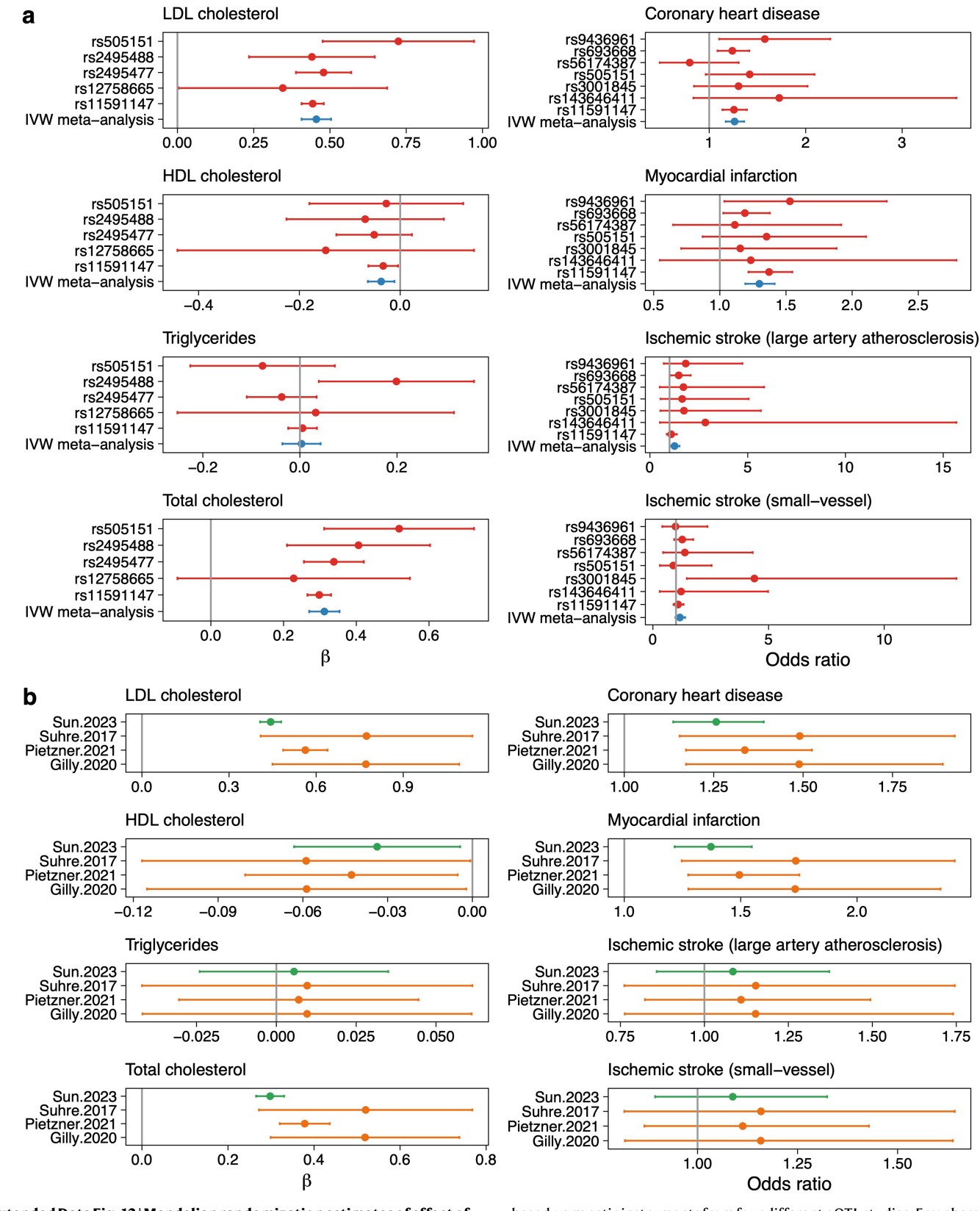

**Extended Data Fig. 12 | Mendelian randomization estimates of effect of increasing levels of *PCSK9* on lipids, cardiovascular diseases and stroke risk.** (a) Effect of PCSK9 plasma protein level on lipids, cardiovascular diseases and stroke risk. (b) Comparison of PCSK9 plasma protein effect estimates based on genetic instruments from four different pQTL studies. Error bars indicate 95% confidence intervals around the effect size estimates. Sample sizes for studies from which summary statistics were derived are detailed in Supplementary Table 30.

Benjamin B. Sun

# Reporting Summary

## Statistics

For all statistical analyses, confirm that the following items are present in the figure legend, table legend, main text, or Methods section.

| n/a | Confirmed | |
|---|---|---|
| ☐ | ☒ | The exact sample size (*n*) for each experimental group/condition, given as a discrete number and unit of measurement |
| ☐ | ☒ | A statement on whether measurements were taken from distinct samples or whether the same sample was measured repeatedly |
| ☐ | ☒ | The statistical test(s) used AND whether they are one- or two-sided<br>*Only common tests should be described solely by name; describe more complex techniques in the Methods section.* |
| ☐ | ☒ | A description of all covariates tested |
| ☐ | ☒ | A description of any assumptions or corrections, such as tests of normality and adjustment for multiple comparisons |
| ☐ | ☒ | A full description of the statistical parameters including central tendency (e.g. means) or other basic estimates (e.g. regression coefficient) AND variation (e.g. standard deviation) or associated estimates of uncertainty (e.g. confidence intervals) |
| ☐ | ☒ | For null hypothesis testing, the test statistic (e.g. *F*, *t*, *r*) with confidence intervals, effect sizes, degrees of freedom and *P* value noted<br>*Give P values as exact values whenever suitable.* |
| ☐ | ☒ | For Bayesian analysis, information on the choice of priors and Markov chain Monte Carlo settings |
| ☒ | ☐ | For hierarchical and complex designs, identification of the appropriate level for tests and full reporting of outcomes |
| ☐ | ☒ | Estimates of effect sizes (e.g. Cohen's *d*, Pearson's *r*), indicating how they were calculated |

*Our web collection on statistics for biologists contains articles on many of the points above.*

## Software and code

Policy information about availability of computer code

| Data collection | R (v3.6.1) |
|---|---|
| Data analysis | R (v3.6.1)<br>REGENIE v2.2.1<br>PLINK v1.9 and v2.0<br>VEP v105<br>ANNOVAR v20211019<br>Ingenuity Pathway Analysis (QIAGEN IPA)<br>WGSA v0.95<br>SuSIE v0.12.6<br>HyPrColoc R package v1.0<br>TissueEnrich R package v1.6.0<br>LiftOver R package 1.20<br>TwoSampleMR R package v0.5.6<br>mrpipeline R package v0.1<br>ieugwasr R package v0.1.5<br>biomaRt R package v2.52.0<br>gwasvcf R package v0.1.0<br>coloc R package v5.1.0 |

For manuscripts utilizing custom algorithms or software that are central to the research but not yet described in published literature, software must be made available to editors and reviewers. We strongly encourage code deposition in a community repository (e.g. GitHub). See the Nature Portfolio guidelines for submitting code & software for further information.

## Data

Policy information about availability of data

All manuscripts must include a data availability statement. This statement should provide the following information, where applicable:

- Accession codes, unique identifiers, or web links for publicly available datasets
- A description of any restrictions on data availability
- For clinical datasets or third party data, please ensure that the statement adheres to our policy

> his is included in the Main Text. An interactive portal for the proteo-genomic results and summary association data are available at https://ukb-ppp.azurewebsites.net/. Underlying proteomics data is available through the UK Biobank Research Access Portal.

# Field-specific reporting

Please select the one below that is the best fit for your research. If you are not sure, read the appropriate sections before making your selection.

☒ Life sciences ☐ Behavioural & social sciences ☐ Ecological, evolutionary & environmental sciences

For a reference copy of the document with all sections, see nature.com/documents/nr-reporting-summary-flat.pdf

# Life sciences study design

All studies must disclose on these points even when the disclosure is negative.

| | |
|---|---|
| Sample size | 54,219 participants; sample size was not predetermined as lower bound of genetic and non-genetic is unknown. |
| Data exclusions | Exclusions as part of QC have been detailed in Supplementary Information and Methods |
| Replication | Internal replication cohort (reported in results) and external replication in independent studies as detailed in the Results and Supplementary Information |
| Randomization | NA - non-interventional |
| Blinding | NA - non-interventional |

# Reporting for specific materials, systems and methods

We require information from authors about some types of materials, experimental systems and methods used in many studies. Here, indicate whether each material, system or method listed is relevant to your study. If you are not sure if a list item applies to your research, read the appropriate section before selecting a response.

### Materials & experimental systems

| n/a | Involved in the study |
|---|---|
| ☐ | ☒ Antibodies |
| ☒ | ☐ Eukaryotic cell lines |
| ☒ | ☐ Palaeontology and archaeology |
| ☒ | ☐ Animals and other organisms |
| ☐ | ☒ Human research participants |
| ☒ | ☐ Clinical data |
| ☒ | ☐ Dual use research of concern |

### Methods

| n/a | Involved in the study |
|---|---|
| ☒ | ☐ ChIP-seq |
| ☒ | ☐ Flow cytometry |
| ☒ | ☐ MRI-based neuroimaging |

## Antibodies

| | |
|---|---|
| Antibodies used | A full list of proteins measured using the antibody-based Olink Explore 3072 is provided on the Olink website: https://olink.com/products-services/explore/ All assays in Olink's panels use antigen affinity-purified polyclonal or monoclonal antibodies (or combinations of both), with the majority being commercially available. |
| Validation | Validation data for the Explore 3072 assay are available on the Olink website: https://olink.com/products-services/explore/ |

# Human research participants

nature portfolio | reporting summary

Policy information about studies involving human research participants

| | |
|---|---|
| Population characteristics | UK Biobank comprises up to 502,650 participants aged between 40 to 69 years at baseline recruited across 22 assessment centres in England, Scotland and Wales. The average age at baseline was 56.52 years (standard deviation, SD 8.09). Of the 502,650 volunteers, 273,468 were women (54.41%), who were on average younger than the men (56.35 years, SD 8.00). Additional details are provided in Hewitt et al. (BMJ Open, 2016).<br><br>A comparison of the UK Biobank with individuals in the general population, conducted by Fry et al. (American Journal of Epidemiology, 2017) found that UKB participants were, "more likely to be older, to be female, and to live in less socioeconomically deprived areas than nonparticipants", suggesting evidence of a selection bias towards healthy volunteers. |
| Recruitment | The recruitment strategy for UK Biobank is described in detail by Bycroft et al (Nature, 2018). Briefly, participants aged 40 to 69 years were recruited across the United Kingdom between the years 2006 and 2010 from the National Health Service (NHS) patient registers. Approximately 9.2 million people living 25 miles (40 km) from one of 22 assessment centers across England, Wales and Scotland were invited to participate, with 5.5% participating in the baseline studies. All participants completed self-report questionnaires detailing their demographic, socioeconomic and health-related characteristics. Participants also underwent several physical assessments (e.g., repeated blood pressure measurements, weight and height). Participants also provided blood, urine and saliva samples, which were then stored in a central storage facility in Stockport, United Kingdom. |
| Ethics oversight | Ethics approval for the UK Biobank study was obtained from the North West Centre for Research Ethics Committee (11/NW/0382). Proteomic profiling of the UK Biobank was approved by the Access Subcommittee of UK Biobank, under Access Management System Application No. 65851. |

Note that full information on the approval of the study protocol must also be provided in the manuscript.

