## [Peer Review File · Nature]

Manuscript Title: Plasma proteomic associations with genetics and health in the UK Biobank

Reviewer Comments & Author Rebuttals

Reviewer Reports on the Initial Version:

Referees' comments:

Referee #1 (Remarks to the Author):

→→In this manuscript by Sun et. al. the authors identify protein quantitative trait loci for 1,463 proteins among 54,306 participants of the UK Biobank. The authors substantially expand the set of variant-protein associations, and highlight examples where genetic variation influences pairs of binding partners and disease susceptibility. Overall, this represents an important effort and provides a resource to facilitate future scientific discovery. However, many of the current conclusions (and the narrow supporting examples like PCSK9 Mendelian randomization) are not novel (eg. the broad conclusions that GWAS of proteins/quantitative traits and Mendelian randomization using pQTLs can recapitulate known drug targets). The manuscript could be improved if the authors were to better contextualize the findings within the current published literature of pQTLs, and demonstrate how the current study expands our understanding of the genetic architecture of the circulating proteome at-large.

Major:

1. The authors chose an antibody-based approach that targeted fewer proteins as compared to aptamer-based approaches that have a higher degree of multiplexing. It would be helpful to the general reader to present the strengths and weakness of each approach and the rationale for choosing the antibody-based approach. This could be done in the Introduction or Discussion.
2. The authors report that 9,098/10,248 of the primary associations were novel, and note in the introduction that ~18,000 variant-protein associations have been previously reported in the literature. This would suggest that many (most?) prior associations were not replicated (at least at the genome-wide significance threshold. Did the authors attempt to replicate previously reported associations and if not, they should?
3. In general the examples cited are interesting but seem to broadly demonstrate the value of pQTL data, which is already well established, and not necessarily what is novel in this analysis. More emphasis on the novel discovery in the current analysis would strengthen the paper.
4. Were fine-mapped causal variants more likely to be coding variants than the lead variant at each locus?
5. In their vignette on the inflammasome, the authors conclude that common genetic variation contributes to inflammasome-mediated immune responses. This finding does not seem surprising given prior GWAS of quantitative traits, inflammatory/immune-mediated diseases, and common-disease GWAS more broadly. Does the current study expand the set of putative genes/proteins that regulate circulating CASP1, IL18, or IL1B (either in comparison to prior pQTL studies, or genes identified in experimental systems?)
6. The association between increased circulating PCSK9 levels and decreased HDL-cholesterol appears to be an order-of-magnitude smaller than the PCSK9-LDL and PCSK9-total cholesterol associations. While the PCSK9-HDL association is nominally significant, it is unlikely to be clinically meaningful, and PCSK9-inhibitors have modest effects on HDL-cholesterol levels in humans (eg. Supplemental Figure 4 from PMID: 30403574). Have the authors performed other sensitivity analyses (eg. colocalization at the PCSK9 locus; PMID: 35452592) to corroborate the PCSK9-HDL association? From a technical standpoint, it would be interesting and important to compare the genetic instruments for PCSK9 derived from the current data with that of previous pQTL datasets with respect to number of variants, F statistics, and percent variance in PCSK9 explained, and to present side by side comparisons of the MR outcomes. This would help demonstrate the value and

novelty of the new pQTL data presented here. From a reporting standpoint, what outcome summary statistics were used for the 2 sample MR?

7. How might the current findings generalize to the broader genetic architecture of the circulating proteome – (eg. this study focused on a relatively narrow subset of proteins in a larger group of individuals than prior studies like Ferkingstad et. al. Nature Genetics 2021, which focused on a larger set of proteins in a slightly smaller sample). Some of this is addressed in Figure 1, but extrapolating to the entire circulating proteome (drawing either quantitative or qualitative conclusions) may be helpful for context. For example, in the Discussion the authors describe the next phase of UKB-PPP, which will include a larger set of proteins – can the authors anticipate the number of cis- and trans- pQTLs they expect to find?

8. Other than the third paragraph, the Discussion is rather underdeveloped. It does little to contextualize the findings in the relevant, broader proteomic and pQTL literature. Rather it seems to either simply make general points about the value of pQTL or the plans of the consortium moving forward.

9. The authors focus on individuals of European ancestry, which is extremely unfortunate for an effort of this magnitude. While the UK Biobank has primarily enrolled participants of European ancestry, the Pan-UK Biobank analysis suggests that >20,000 participants have non-European ancestry. While some recent studies have begun to identify pQTLs in populations of non-European ancestry (Eg. PMID: 35501419), the sample sizes have been small in comparison to pQTL studies among European populations (Eg. PMID: 34857953). Including some (or any) non-European populations from UKB in the discovery effort of the current study would have likely represented the largest pQTL analyses of their kind. It has become customary to conclude that “further study in non-European ancestry populations is warranted,” but the exclusion of non-European populations in this large, privately-funded endeavor deserves further comment. The call that “future investments in population proteomics prioritize genetic diversity in their cohort selection” rings hollow, as the opportunity to make inroads with the current study was not pursued – is this consortium prepared to leverage non-European ancestry samples of UKB to begin addressing this gap?

Minor:

1. In the introduction, the authors list several potential applications of integrating population-scale genomic and proteomic datasets. (eg. improved loss-of-function prediction, biomarker identification, fine-mapping). Most of these approaches do not appear to be employed in the current manuscript and may be better suited for the discussion/future directions.

2. The type of proteomic platform (eg. antibody-based) should be included in the main results in addition to the commercial name of the method (eg. Results paragraph 1 and/or Proteomic data processing and quality control).

3. In the description of baseline/demographic differences between the consortium vs. randomized participants, the authors are encouraged to report the results using a similar format for all associations (for example in the text some associations are described based on absolute differences between groups, some include only a p-value, and some include proportions in each individual group rather than absolute differences in each group).

4. In Figure 1D the authors arbitrarily bin age – can age (as a continuous variable) vs. glycodeclin levels be plotted? While circulating levels clearly decline among women until the age of ~55-60, the authors assert that levels rise among men. However it is not clear from the plot there is any meaningful change across the age bins. In general, given the large sample size, the authors are encouraged to consider the biological (in addition to statistical) meaning of the associations that are highlighted.

5. Line 267 – Figure 1D does not appear to show mean total SNP-based heritability as indicated in the manuscript text. This reference appears to correspond to Extended Data Figure 6.

6. In “Biological enrichment for proteins with multiple trans associations” the authors use CR2 as an example of a gene with multiple trans-associations which are enriched in specific biological pathways. It would be helpful provide some context (eg. to note the CR2 encodes the complement receptor 2) in order for the reader to better interpret the pathway enrichment.

7. Lines 344-354 – the authors suggest that because rs2234962 (located near HSPB6) is

associated with circulating BAG3 and downstream biomarkers of heart failure (in contrast to the primary cis-pQTL rs35434411), HSPB6-BAG3 may be an important complex in the development of heart failure. While this reviewer agrees, the presentation of this logic could be clarified (a diagram may be helpful). Further, the authors note that rs2234962 is a missense variant – does this coding variation affect formation of the HSPB6-BAG3 complex?

8. Line 423 – missing “and” between “eQTLGen” and “GTEx”

9. Line 429-432 – it would be helpful to present the numeric results consistently [eg. % (X/Y)] within the entire colocalization section. It’s not entirely clear what proportion of colocalized proteins colocalized within a single tissue (eg. is it 191/507 = 38%?).

10. Line 441 – the blood-brain barrier explanation would seem to generalize to broader sets of tissues/organs with variable/differential blood permeability.

11. Lines 786, 792 – “multivariate” should be “multivariable” as only a single outcome was used in the regression model.

12. The blood group-FUT2 secretor interaction plot (Figure 5A) and associated manuscript text is confusing as presented. The main point of this plot/text is to demonstrate that secretor status modifies the effect of blood group on protein levels. As presented, the boxplots facilitate comparison of protein levels between secretors and non-secretors within each blood group. However, the authors draw attention to differences across both blood group and secretor status. Visually, this information is very condensed within the current plot, and it may be useful to instead facet each subplot (for a total of 8 subpanels) (or group the x-axis) by secretor status rather than blood group.

Referee #2 (Remarks to the Author):

A. Summary of the key results

Sun et al. present the currently largest genome-wide analysis (GWAS) of protein levels determined in blood plasma using an affinity proteomics approach on > 50000 samples collected by the UK biobank. The work presents a compendium of investigations as a resource. It confirms previous observations of direct genetic regulation of 1500 plasma protein levels (cis) and adds key insights into how distant (trans) genes and networks contribute to regulating protein levels.

B. Originality and significance: if not novel, please include reference

The work is the first major output from a consortium built by partners from the pharmaceutical industry and presents a major expansion of previous proteins GWAS. The authors present that the increased sample size and a newer proteomics approach allowed them to identify new pQTLs. While the discovery of cis-pQTLs plateaus in around 10k subjects, the number of trans-pQTLs and utility of such associations continues to increase with added sample sizes. This finding alone opens the gate for novel investigations of the dynamic architecture of the circulating proteome. It also points to the necessity of conducting even larger plasma proteomics studies, hence following the path of how genetics made it possible to increase our understanding of human biology.

To reach a wider audience, the author should investigate and explain why trans-pQTLs could not be found at the depth before and whether this is driven by only the sample size or could be a consequence of their inclusion criteria for selecting samples. Regarding the highlighted enrichment of SLE, which was only due to 500 samples (1% of the study set), is this a coincidence or a conscious choice? Please clarify if relevant. Other phenotypes were more prevalent in numbers but less strongly enriched. Also, confirm if the stated diagnoses were made at any time point of participation or during the sampling.

C. Data & methodology: validity of approach, quality of data, quality of presentation

The study presents data from one of the world's most exquisite and well-studied population cohorts, the UK biobank. The work appears as the first study to use UKB blood samples for large-scale proteome analysis. Some co-authors and community members have used similar but smaller-sized population UK-based cohorts to conduct protein GWAS. It would be interesting to know if - by coincidence - any among the 50k subjects were also participating in any of the previous studies.

The blood samples were analyzed by the recently established Olink's Explore platform, which is now capable of detecting 1500 circulation proteins. There is an extensive description of how the Olink data batches were processed to perform the presented analyses. What is missing is a global analysis of the data in PCA, UMAP, or similar types of dimensionality reduction approaches. These would help understand if any global pattern exists and if these can be explained by any of the chosen traits, ethnic, center, or technical aspects.

D. Appropriate use of statistics and treatment of uncertainties

My current understanding of the UK biobank setting is that blood samples have been taken at different centers across the country and originate from donors recruited over four years. It is, though, unclear when the blood samples were taken and how these were processed.

The authors should investigate if the sampling centers contribute to any differences in protein levels and if genetics, lifestyles, or demographics can assist to explain the expected center effect. Since the detection of protein levels may be influenced by the time elapsed between sampling and analysis (age of sample), appropriate analyses should be conducted to annotate the protein.

To enrich their resource, it would be helpful to determine the most and least variable proteins across the cohort, age groups, and sexes. Please also investigate if their heterogeneity of variance and if an annotation of their provides hints to explain the genetic variation and environmental or lifestyle effects.

Proteins may be secreted, leaked, or shedded into the blood. Do the authors see any systematic enrichment for these categories or their abundance levels regarding genetic regulation?

When investigating the effect of blood cell counts, please check if the proteins are - exclusively or primarily - expressed in blood cells or other organs. See <https://doi.org/10.1126/science.aax9198>.

The authors focussed on genetic regulation and only briefly touched upon clinically measurable traits such as age, sex, and BMI. Even partly outside the current scope, it would be highly interesting to know if their protein level data can predict age, sex, and BMI. This may help to model these traits solely on experimental data and assist future studies to use the predicted values for trait adjustment. Would it also be possible to predict blood types from such high-dimensional data?

Even though age, sex, and BMI traits are the major drivers of differences in the plasma protein levels, there are obvious reasons why the data has not been studied for disease-specific biomarkers. However, it would be valuable to know if other binary (yes/no) traits collected at samplings, such as smoking, disease diagnosis, or use of (any) medication, can be investigated to enrich the current list and reveal noteworthy proteins.

For the proteomics community, studies of 50k subjects are currently still extremely rare, and increasing sample size may inflate some of the initial observations. Please clarify the approach used to manage proteomics data at that scale.

E. Conclusions: robustness, validity, reliability

The authors discuss conducting more plasma proteome analysis on UK biobank samples to validate the presented observations and findings. A cross-platform comparison may follow up on the presented work to investigate further aspects of the plasma proteome that cannot be measured by the Olink approach. What do the authors expect in terms of validation, and are there any features in the data, such as MAF or effect size, that would make the current findings more robust?

The presented data is "limited" to the Olink approach that measures 1500 proteins. As mentioned by the authors, another 1500 proteins have become available on the platform, and these would be measured in the UK biobank samples. Will the authors implement additional data, or is there a rationale to include the other 1500 proteins in a separate study?

The authors acknowledge the use of a primarily white population, which limits the generalizability of some current observations. However, would ethnicity be detectable in the protein level data?

F. Suggested improvements: experiments, data for possible revision

For the time being, the authors use existing data to confirm their novel aspects. It remains partly unclear how the data could be validated by non-plasma proteomic efforts. Especially a systemic investigation of the trans-regulation of secreted proteins may pose a challenge but create new opportunities and push the boundaries for studying human biology. The author should clarify which experimental routes and possibilities exist to study these cases. In addition, the authors should provide examples of how they would select targets. For example, is a larger effect size a meaningful guide to prioritizing targets of interest and reducing the need for larger sample sizes?

The work represents a fantastic resource and - at this level of importance - would benefit from an accessible portal to host the summary statistics. It would be highly valuable to develop an app or interface in a managed environment to allow the community to access and browse their favorite protein or association of interest and thereby increase the exposure and utility of the data and findings.

The breadth of data provides many opportunities to present different observations, but it also limits the possibilities to focus on core findings. The collection of observations tends to appear as a report rather than a resource that can be used by the community. Some guidance on how to visualize the data and utilize the resource would be suitable. Even though many aspects can be lifted, it remains unclear what the authors deem as the major or novel findings that exceed those made in previous larger-scale proteins GWAS.

Figure 3 presents very appropriate examples of protein regulation. Most proteins are well-known, fairly abundant, and important for blood function. Please investigate protein level data for these example pathways, such as by stratifying for an appropriate genotype and displaying the proteomics consequences downstream.

G. References: appropriate credit to previous work?

The included references were found appropriate.

H. Clarity and context: lucidity of abstract/summary, appropriateness of abstract, introduction and conclusions

The title "Genetic regulation of the human plasma proteome in 54,306 UK Biobank participants" is not very appealing (and connects with previous work from Sun et al in 2018: "Genomic atlas of the human plasma proteome"). Isn't the utility of the plasma proteome architecture or the findings made via the trans-pQTLs worth being highlighted?

Referee #3 (Remarks to the Author):

Thank you very much for the opportunity to review this paper.

Sun et al provide results from a pGWAS of 1,463 proteins measured using the Olink Explore assay in 35,571 European-descent individuals of UKBB, with an additional, independent set of 18,181 UKBB participants (different ethnicities) for replication. This is the first phase of the commercially funded UKB-PPP with results for another 1,500 proteins being imminent, and hence this paper is somewhat preliminary. Overall, this is a very well conducted and written study, and I would like to congratulate specifically the leading but also all authors for steering this multi-partner industry consortium jointly to successful completion in a short timescale.

The discovery arm of this study is of comparable size than the earlier Decode study based on Somalogic's aptamer technology that covers around 5000 protein targets (PMID: 34857953) and therefore not unprecedented. Strengths of the current study include division into an independent discovery and replication set, and conduct of detailed sensitivity, enrichment, and interaction analyses. I like the systematic exploration of the consistency of effect directions for shared expression signals and putative explanations of discordant directions. The study's most important weaknesses include the total omission of a) any systematic evaluation of the downstream clinical consequences of the identified pQTLs to provide biological and translational insights, b) the utilisation of the ethnic diversity that exists within UKBB (and the selected UKBB-PPP samples). Directly addressing point a), and for b) adding results that address the applicability of (at least cis) pQTLs from European-descent participants to other ethnic groups plus an attempt to identify ethnic specific signals, are in my view essential to provide new insights worthy of this large-scale, ambitious effort.

Given this and the numerous previously published pQTL studies (all those considered are helpfully listed in the supplement), it is maybe not surprising that the current effort only identifies 562 new cis pQTLs. This is in line with the author's observation that the number of cis-pQTLs starts to plateau after sample sizes of around 5k. The sample size of the UKBB-PPP therefore largely enhances detectability of new trans pQTLs, many of which are extremely pleiotropic, something that has been obvious in earlier studies. Trans-signals are therefore what most of the reported novel associations can be attributed to (9,309 of all 10,248 associations).

The authors mention the successful exome sequencing and ongoing whole genome sequencing of UKBB. Why were the existing sequencing results not considered, at least for cis variation? This could have provided a more substantial advance in knowledge.

Apart from the important weaknesses above and my detailed questions and suggestions below, I have one other more major technical concern related to the number of reported independent secondary associations based on fine mapping. It is acknowledged that SuSiE tends to select credible sets that are non-overlapping but contain variants in high LD, which is likely to be an artefact of the algorithm rather than representing a true finding (see `run.susie()` documentation of the `coloc` R package). This can also lead to instances in which SuSiE selects a large number of credible sets. I would therefore suggest running SuSiE with an increasing L , i.e. maximum number of credible sets to identify the smallest L that includes distinct credible sets. I would also ask the authors to confirm statistical independence of variants in the credible set using a joint linear regression model that includes all variants at once.

The sensitivity analyses testing the impact of blood cell composition, BMI, season, or fasting time on pQTL effects are useful and show minimal impact on protein levels for most genetic associations, as one might expect. This ought to not be confused with the strong effect of those parameters on protein levels. Since this is the first paper on proteomic measures in UKBB and not all readers will immediately appreciate this, I propose to expand this section to include a general investigation and description in the supplement of the effect of all of these parameters on protein levels. It would be useful if the authors included a figure with the variance explained of all proteins by each of these 'sensitivity' metrics in addition or contrast to the variance explained by other factors (sex, age, ethnicity, deprivation, cis and trans effects, etc).

The initial paragraph on 'biological associations' appears a little superficial and doesn't provide deeper insights. Given the investment of the consortium and findings of existing studies, one would rather hope 'that the proteomic assay can capture physiological effects'. I wonder whether it would help if the authors expanded the exploration of these associations but moved this part to the supplement? I suggest to specifically investigate and add the impact of prevalent diseases and differences in liver and kidney function on protein levels with brief mention of key results in the main text.

The section on the interaction between ABO blood group genotypes and FUT2 secretor status could be strengthened by qualifying the nature and relevance of the reported differences/ interactions for the 38 proteins in more detail (directions and significance). The statement 'We saw that the extent of differences in protein levels between secretors and non-secretors varied depended on the blood group for these proteins' is redundant, i.e. a restatement of the earlier sentence. The following paragraph on gene expression enrichment doesn't follow logically; could the authors please explain this better? I also suggest toning down and referencing the statement '(...) which may underline susceptibility to various FUT2/ABO associated GI conditions'.

The authors state that for 49% of primary cis associations, the index variant was in at least weak LD ($r^2 > 0.01$) with a protein-altering variant. I suggest adding this information for a less permissive LD threshold and compare it to what previous studies and technologies have reported.

The authors provide some details of the generalisability of the selected individuals, which is helpful. Could some more information be provided about the stratified sampling to clarify a) how this led to more efficient sample picking and b) whether or how this was accounted for in the observational and genetic analyses? Could the authors please expand supplementary table 1 to include recruitment centre, time since recruitment/ sampling, deprivation index, drug use (for selected commonly prescribed drugs), and other biological parameters of interest, such as biomarker levels etc.

The section on proteomic insights into COVID-19 associated loci remains rather technical without much biological insight or follow-through. Reference to earlier studies for the reported genes and potential explanations for any potential lack of replication could be discussed in more detail, as could the impact of the heterogeneity of case definitions.

The statement 'highlights the strengths of the antibody based Olink® Explore Assay for pQTL detection and downstream biological discovery' seems somewhat unsubstantiated. I suggest replacing 'antibody based Olink® Explore' with 'broad-capture, affinity-based proteomics'. The value of this study lies in its scale and without direct comparison, i.e. comparative assessment in overlapping samples, inference about strengths or superiority of a given platform would best be avoided.

The discussion highlights the need for orthogonal validation of antibody-based proteomics using aptamer- and mass spectrometry-based assays and the consortium seems to have initiated systematic evaluation of this. It would be reasonable to provide brief reference to earlier work addressing this issue (PMID: 34819519). Given that a 'comparative' preprint based on the UKBB

proteomic data had to be retracted, it would be helpful if the analytical protocol and membership of this new effort was made public in advance, to avoid bias, commercially driven interests, and enhance transparency of process for the benefit of the scientific community. While the investment of the consortium is applaudable and will clearly provide large scientific benefit, the inherent value of the (mainly tax- and Charity-funded) resource itself is substantial, and there is an obligation to be conscious that even small statements can unduly influence the share value of listed proteomic companies.

Other points:

The description of Olink's internal QC assessment in the plasma profiling section of the supplement is relevant (since it was adhered to) but reads a bit like a copy and paste of an Olink's data report. More importantly, UKBB Expand QC parameter look almost too good to be true with extremely few few sample or assay warnings, compared to what others have seen and reported for the same technology. Were any duplicates run, samples or plates?

A two-step normalisation procedure was used to generate NPX (1: across plates, 2: across batches). Please include information on how much this influences protein CVs?

Was there any reason for selecting batch 1 as the reference for 'across batch' normalization? Is there good concordance if another batch (e.g. largest, best) is selected as reference?

Could the authors please perform an analysis of plate effects similar to the one done for investigation of batch effects. It would be helpful to see whether any proteins are particularly affected by plate.

LOD information is helpful but the related Figure S7 is not as informative as it could be due to the y-axis. How many proteins have less than 5% of samples below LOD? Please state this explicitly in the text and redraw the figure either breaking the y-axis or removing proteins with less than 5% below LOD. Please add a table that LOD information for all proteins or at least all with LOD>5%. The authors have focused this part of the % of protein assays with values below LOD per sample. It would be helpful to also know whether any samples have lower detectability than the average.

Language checks:

Refer to men/women not males/females?

Refer to cross-sectional differences as 'higher or lower levels' rather than a 'de-/increase' which implies a time/ longitudinal aspect?

There are some small errors that need correcting throughout, ie pleotropic etc.

Author Rebuttals to Initial Comments:

Referee expertise:

Referee #1: human genetics, omics data

Referee #2: proteomics

Referee #3: genetic epidemiology, omics data

Referees' comments:

Referee #1 (Remarks to the Author):

In this manuscript by Sun et. al. the authors identify protein quantitative trait loci for 1,463 proteins among 54,306 participants of the UK Biobank. The authors substantially expand the set of variant-protein associations, and highlight examples where genetic variation influences pairs of binding partners and disease susceptibility. Overall, this represents an important effort and provides a resource to facilitate future scientific discovery. However, many of the current conclusions (and the narrow supporting examples like PCSK9 Mendelian randomization) are not novel (eg. the broad conclusions that GWAS of proteins/quantitative traits and Mendelian randomization using pQTLs can recapitulate known drug targets). The manuscript could be improved if the authors were to better contextualize the findings within the current published literature of pQTLs, and demonstrate how the current study expands our understanding of the genetic architecture of the circulating proteome at-large.

Major:

1. The authors chose an antibody-based approach that targeted fewer proteins as compared to aptamer-based approaches that have a higher degree of multiplexing. It would be helpful to the general reader to present the strengths and weakness of each approach and the rationale for choosing the antibody-based approach. This could be done in the Introduction or Discussion.

We thank the reviewer for this important recommendation. The relative strengths and weaknesses of aptamer- and antibody-based techniques is a complex and, occasionally, contentious topic. Given the lack of aptamer-based proteomic measurements in UKB facilitating direct comparisons between assays, we originally refrained from discussing the strengths and weaknesses of each approach. However, since the initial submission of this manuscript, several peer-reviewed studies have been published on this topic. We have revised the Discussion to reference two comparative investigations – the first by Katz et al. (*Science Advances*, 2022) and the second by Pietzner et al. (*Nature Communications*, 2021). The Katz study indicates that the most widely applied aptamer-based assay offers greater biological breadth and narrower coefficients of variation, whereas the most widely applied antibody-based assay offers greater target specificity and stronger correlations with a limited set of disease outcomes and targeted immunoassays. The Pietzner paper offers a more nuanced conclusion – indicating that aptamer- and antibody-based techniques capture different components of protein biology and, thus, may offer synergistic insights into human health and disease. We highlight how aptamer-based assays have been applied at much larger scale than antibody-based assays, despite the potential synergistic insights to be gained by applying both approaches. Citing advances in the throughput and breadth of antibody-based assays (now including 2,923 unique protein measurements), we describe how the UK Biobank Pharma Proteomics Project was formed to complement the established body of aptamer-based pQTLs with a comparably sized, open-source collection of antibody-based pQTLs. We hope that this revised Discussion section provides sufficient rationale for selection of the antibody-based approach.

2. The authors report that 9,098/10,248 of the primary associations were novel, and note in the introduction that ~18,000 variant-protein associations have been previously reported in the literature. This would suggest that many (most?) prior associations were not replicated (at least at the genome-wide significance threshold. Did the authors attempt to replicate previously reported associations and if not, they should?

We can confirm that we attempted to replicate previously reported associations and thank the reviewer for highlighting how this was unclear in the original manuscript. In the revised manuscript, we have updated the number of previously identified variant-protein associations from ~18,000 to over 12,000, reflecting the number of independent associations. Of the approximately 12,000 pQTLs identified as part of prior, largely aptamer-based studies, only 2,201 (approximately 17.35%) were replicated in this study, whereas 10,483 pQTLs were novel. This relatively modest degree of replication of prior pQTLs is partly explained by the limited overlap between antibody- and aptamer-based platforms. Only 1,982 proteins are commonly captured by the Olink Explore and Somalogic SomaScan platforms:

When limiting prior pQTL studies only to those conducted using antibody-based methods, the degree of overlap is considerably higher, at 77.7% (599/771):

When limiting previously identified pQTLs to the 1,982 proteins commonly captured by both Olink and Somalogic, the replication rate is approximately 37.5% (1,865/4978). Although this replication rate is

considerably lower than the replication rate for antibody-based pQTLs, it is likely that differences in isotopes, relative affinities for post-translational modifications and other technical and biological variations between aptamer- and antibody-based platforms are contributing towards this issue:

We have updated the Main Text, Supplementary Information and Methods to better describe these findings ('Identification of novel pQTLs in the context of previous pQTLs', p17 and have included a paragraph in the Discussion section (p28) highlighting how replicate rates are partly explained by (1) the expanded coverage of the Olink platform, (2) limited overlap between antibody- and aptamer-based platforms, and (3) technical differences between the two affinity-based techniques:

Our analysis identifies approximately twenty-fold more associations than all prior antibody-based studies. This reflects both the increased sample size of the present study as well as the recently expanded coverage of antibody-based assays. Most previously identified antibody-based pQTLs were replicated in our analysis (77.6%). When we compared our findings to aptamer-based studies, limiting to the set of common proteins between the two platforms, a much smaller percentage of proteins replicated (37.5%). This modest overlap is unsurprising, given that correlations between proteins measured by both platforms are highly variable, and shared genetic signals can be masked by extreme, assay-specific binding affinities. Overall, the largely novel collection of antibody-based pQTLs identified in this study can be combined with the broader body of aptamer-, immunoassay- and mass spectrometry-based pQTLs to interrogate disease mechanisms driven by differences in protein abundance(s), alternative splicing, and other functional alterations less comprehensively captured using only one technique.

3. In general the examples cited are interesting but seem to broadly demonstrate the value of pQTL data, which is already well established, and not necessarily what is novel in this analysis. More emphasis on the novel discovery in the current analysis would strengthen the paper.

We thank the reviewer for highlighting how we should place greater emphasis upon novel findings that have not previously been observed and have moved examples demonstrating the broader value of this population-scale pQTL dataset to the supplement.

In addition to doubling the number of proteins assessed and incorporating ancestry specific analyses, we have restructured our manuscript to emphasize the extent of additional *trans* pQTLs discovered at larger

sample sizes, where, for the first time, we observe extensive networks of *trans* pQTLs regulating target protein levels, either as protein interactions or biological pathways. This is more evident and extended further with the expanded dataset, as seen in the complement component example. Many of the additional *trans* pQTLs would not have been seen at smaller sample sizes, as shown in our sub-sampling analyses (Main Text p23, Supplementary Information p22-23)

Using the expanded dataset of almost 3,000 proteins, we also extend our findings on the long-range *trans* interactions between *ABO* and *FUT2*, with the strongest associations showing enrichment for gastrointestinal (GI) function and GI disease susceptibility. We believe that this is one of the first large-scale discoveries of genetically driven epistatic effects on plasma protein levels, given that *ABO* and *FUT2* are located on different chromosomes. We also illustrate how several genetic associations, previously assumed to be cohort-specific (such as inflammasomes/*NLRP12*) can be detected, albeit at smaller effect sizes, in an independent population. Additionally, in the expanded proteomic dataset, we find novel associations between *NLRC5* missense variants and MHC-class I proteins, such as B2M and BTN3A2. These variants, as *trans* eQTLs, have been shown to be tightly transcriptionally regulated; our findings illustrate how these *trans* eQTL effects are also reflected as *trans* pQTLs. We have moved some of the examples of downstream applications (e.g., multi-trait colocalization and MR, with new insights from our existing approach) into the Supplementary Information (p29-35) to facilitate greater focus on the key results.

4. Were fine-mapped causal variants more likely to be coding variants than the lead variant at each locus?

To address the reviewer's question regarding the prevalence of coding variants among the fine-mapped signals, we have included Supplementary Table 17 in the revised submission and have provided additional descriptive information about protein-altering variants (PAVs) in the Supplementary Information (p20-21). The expanded descriptions include: (1) the number of credible sets containing at least one protein-altering variant (PAVs), (2) whether credible sets for the primary signal at *cis* loci were more likely to contain PAVs compared to those of non-primary signals, and (3) the frequency of *cis* credible sets containing a single PAV. Among *cis* credible sets that contained PAVs, the vast majority only contained a single PAV, which was also frequently the most probable variant.

5. In their vignette on the inflammasome, the authors conclude that common genetic variation contributes to inflammasome-mediated immune responses. This finding does not seem surprising given prior GWAS of quantitative traits, inflammatory/immune-mediated diseases, and common-disease GWAS more broadly. Does the current study expand the set of putative genes/proteins that regulate circulating CASP1, IL18, or IL1B (either in comparison to prior pQTL studies, or genes identified in experimental systems?)

We thank the reviewer for their observations regarding the nature of this vignette. While some of the results of this vignette have been previously identified, some findings are novel (as described below), again highlighting the power of the large scale of this study.

- Replication of previously reported pQTLs between genes known to encode for inflammasome components and plasma IL-18 levels
- Identification of novel pQTLs between genes known to encode for inflammasome components and plasma IL-1B, IL-18, and CASP1 levels
- Identification of novel pQTLs between genes with unclear relationships with inflammasomes and plasma IL-1B, IL-18, and CASP1 levels (note: experimental investigation of the potential relationship between these genes and inflammasome function is beyond the scope of this project)
- Newly added pQTL results linking *NLRC5* missense variants with plasma MHC-class I proteins and BTN3A2.

We have included additional context for this vignette and added fields to Supplementary Table 29 to clarify the nature of these findings. Text describing these results has been moved to Supplementary Information (p33-34).

6. The association between increased circulating PCSK9 levels and decreased HDL-cholesterol appears to be an order-of-magnitude smaller than the PCSK9-LDL and PCSK9-total cholesterol associations. While the PCSK9-HDL association is nominally significant, it is unlikely to be clinically meaningful, and PCSK9-inhibitors have modest effects on HDL-cholesterol levels in humans (eg. Supplemental Figure 4 from PMID: 30403574). Have the authors performed other sensitivity analyses (eg. colocalization at the PCSK9 locus; PMID: 35452592) to corroborate the PCSK9-HDL association? From a technical standpoint, it would be interesting and important to compare the genetic instruments for PCSK9 derived from the current data with that of previous pQTL datasets with respect to number of variants, F statistics, and percent variance in PCSK9 explained, and to present side by side comparisons of the MR outcomes. This would help demonstrate the value and novelty of the new pQTL data presented here. From a reporting standpoint, what outcome summary statistics were used for the 2 sample MR?

We thank the reviewer for this comment. The smaller effects of *PCSK9* protein level on HDL are reflective of the smaller effects seen in clinical trial results, whereby the impact on HDL is also much smaller than LDL and total cholesterol effects. We agree that these effect sizes may not be clinically meaningful; however, they highlight the added power to be leveraged from stronger aggregated instruments resulting from this larger pQTL study. To address the reviewer's request for additional analyses supporting the effect of *PCSK9* protein levels on HDL, we performed colocalization analyses between PCSK9 and all 8 lipid and disease outcomes examined in our original MR analysis. We did not find colocalizations with HDL and ischaemic stroke due to a lack of strong genetic associations with the outcome (since PP1 is high – association with PCSK9 but not the outcome), rather than due to distinct causal mechanism (PP3 in our results were low – independent associations with PCSK9 and the outcome; Supplementary Tables 30). These results alleviate concerns regarding invalid instruments as described in the paper the reviewer suggested (Zuber et al, 2022, PMID 35452592). Notably, we selected our instruments using *cis* PCSK9 variants, as in previous *PCSK9* MR studies, to minimize pleiotropy due to distinct causal variants and non-*PCSK9* pathways. This is corroborated by the strong observed colocalizations for LDL, MI, CHD, TC with high PP4 (colocalization of associations for PCSK9 and the outcome; Supplementary Tables 30).

To address reviewer's suggestion to compare across pQTL datasets, we have added the comparison of the same PCSK9 genetic instrument across four different pQTL studies from 2017 to the current study and performed additional Mendelian randomization analyses to demonstrate the gain in precision of the genetic instrument and the power of MR analysis with this much larger pQTL library. This is clearly demonstrated by the evidently smaller standard errors and confidence intervals in our current MR effect estimates compared with previous studies for the same lead functional variant (Extended Data Figure 12b, Supplementary Table 30).

Finally, we now include the data sources for the outcome GWAS used in the PCSK9 MR analysis (IEU openGWAS project; <https://gwas.mrcieu.ac.uk/datasets/>; GWAS ID and reference included in Supplementary Tables 30). We have revised the text in Supplementary Information (p35), Extended Data Figure 12, and added Supplementary Tables 30 to include these new results.

7. How might the current findings generalize to the broader genetic architecture of the circulating proteome – (eg. this study focused on a relatively narrow subset of proteins in a larger group of individuals than prior studies like Ferkingstad et. al. Nature Genetics 2021, which focused on a larger set of proteins in a slightly smaller sample). Some of this is addressed in Figure 1, but extrapolating to the entire circulating proteome (drawing either quantitative or qualitative conclusions) may be helpful for context. For example, in the Discussion the authors describe the next phase of UKB-PPP, which will

include a larger set of proteins – can the authors anticipate the number of cis- and trans- pQTLs they expect to find?

We have now integrated data from the second phase of UKB-PPP, incorporating 2,923 unique proteins, in total. The general trends in this expanded collection of protein measurements roughly fall in line with our initial anticipations. We also place the number of findings from ancestry specific analyses in the context of similar European predominant sample sizes and find that each ancestry's total pQTL findings fall in-line (Main text Figure 2e). We have revised our subsampling analyses to provide additional insights:

For context, it is critical to recognize that the selection of content evaluated on affinity-based proteomics platforms tends to prioritize proteins known to be secreted with higher circulating levels. That intentional bias diminishes as platforms expand and add content with lower or less frequent circulation levels.

1. For broader proteomic coverages, anticipated in future expansions of antibody and aptamer-based technologies, we subsampled proteins to account for reduced protein abundances as assays expand. This analysis shows slower gains in pQTL yield with increased proteome coverage; however, there are still considerable additional gains to be made before detectability of proteins present in plasma becomes a key bottleneck.
2. We updated the 'variance explained' component to account for independent associations from SUSIE (European based LD) – leading to higher but more accurate variance estimates.

We have expanded the Discussion (p29) and Supplementary Information (p22-23) section to describe how we expect pQTL associations to scale with sample size and protein coverage. More generally, we emphasize the natural constraint of proteins that are detectable in the human plasma and added insights on how reduced abundance of proteins influences the number of pQTLs observable in plasma.

We believe that this revised analysis complements and expands upon the work described by Ferkingstad *et al.* – offering a large-scale, open-access resource based upon data generated using an independent technology. The revised manuscript includes updated sub-sampling analyses, offering more in-depth insights of how pQTL architecture changes with sample size and proteome coverage at reducing abundances, putting our study, previous studies and future studies in context for the first time (Supplementary Information, p22-23).

Notably, the research team involved in the Ferkingstad *et al.* paper has directly compared findings from their Somalogic-based dataset to findings gleaned from our Olink-based dataset as part of an independent investigation and manuscript (in preparation); this analysis was beyond the scope of the present study.

8. Other than the third paragraph, the Discussion is rather underdeveloped. It does little to contextualize the findings in the relevant, broader proteomic and pQTL literature. Rather it seems to either simply make general points about the value of pQTL or the plans of the consortium moving forward.

We have revised and expanded the Discussion to include greater contextualization of our findings in the relevant proteomic and pQTL literature and have removed or de-emphasized prior statements regarding the value of pQTLs or the plans of the consortium moving forward. The revised Discussion focuses on five topics, including: (1) The relative strengths and weaknesses of aptamer- and antibody-based assays and our rationale for selecting an antibody-based assay, (2) contextualization of our findings within the broader pQTL / proteomics literature, (3) potential use cases for *trans* pQTLs and pQTL networks identified in the present study, (4) predicted scaling of pQTL detections with increased sample sizes and higher multiplexing,

and (5) initial insights into ancestry specific pQTLs and the value of conducting larger-scale, non-European pQTL mapping.

9. The authors focus on individuals of European ancestry, which is extremely unfortunate for an effort of this magnitude. While the UK Biobank has primarily enrolled participants of European ancestry, the Pan-UK Biobank analysis suggests that >20,000 participants have non-European ancestry. While some recent studies have begun to identify pQTLs in populations of non-European ancestry (Eg. PMID: 35501419), the sample sizes have been small in comparison to pQTL studies among European populations (Eg. PMID: 34857953). Including some (or any) non-European populations from UKB in the discovery effort of the current study would have likely represented the largest pQTL analyses of their kind. It has become customary to conclude that “further study in non-European ancestry populations is warranted,” but the exclusion of non-European populations in this large, privately-funded endeavor deserves further comment. The call that “future investments in population proteomics prioritize genetic diversity in their cohort selection” rings hollow, as the opportunity to make inroads with the current study was not pursued – is this consortium prepared to leverage non-European ancestry samples of UKB to begin addressing this gap?

We thank the reviewer for this valid criticism. Despite the limited sample sizes of non-European participants in UK Biobank, this is a highly important issue that we should at least begin to address, as smaller, analyses of non-European participants could still facilitate future meta-analyses and downstream analyses requiring ancestry specific LD and summary data (e.g., Coloc and MR).

In addition to the European discovery/replication analysis and the full cohort analysis (the latter of which was included in the first draft, but described in greater detail as part of this revision), we have:

- Included non-European, ancestry-specific pQTL analyses based on the Pan-UKBB defined ancestry groups, and
- Summarised the key results from non-European ancestry analyses in a dedicated section of the Results (p14-15)

Given the much smaller sample sizes, most of the associations were, unsurprisingly, already observed in the European analysis. However, we identified a small collection of pQTLs that appear to be enriched in populations of non-European ancestries due to the allele being more enriched in non-European ancestries/near absent in European ancestries. Some examples are discussed in more detail in the added results (p14-15). This includes examples of pQTLs that are depleted in frequency in European ancestry and hence not found.

Effect sizes were well-aligned and the number of detected pQTLs was in-line with expectations for comparable European analyses of similar sub-sample sizes, suggesting that no specific ancestry systematically out-performed the other. We also found that MAF of primary pQTLs in African ancestry was enriched against European not seen for Central/South Asian vs Europeans despite similar sample sizes, suggesting African ancestry potentially provides additional signals over Europeans compared to Central/South Asian ancestries.

Although we found that there is limited gain in pQTLs from non-EUR ancestries due to small samples sizes, we observed indications of potential gains in pQTLs at expanded samples for variants that are enriched in non-European ancestries. Thus, there is likely added value in conducting large-scale pQTL mapping in non-European populations.

The added ancestry-based analyses and results are now in Main text (p14-15), Supplementary Table 11, Figure 2e inset, Extended Data Figure 5b and c)

Minor:

1. In the introduction, the authors list several potential applications of integrating population-scale genomic and proteomic datasets. (eg. improved loss-of-function prediction, biomarker identification, fine-mapping). Most of these approaches do not appear to be employed in the current manuscript and may be better suited for the discussion/future directions.

We have revised the manuscript and focused on key findings as suggested. We have removed this line from the introduction. The applications of the UKB-PPP dataset will extend beyond the scope of the present study – as evident at the 2022 ASHG conference and the publication of several complementary pre-prints (e.g., doi: <https://doi.org/10.1101/2022.10.09.511476>).

2. The type of proteomic platform (eg. antibody-based) should be included in the main results in addition to the commercial name of the method (eg. Results paragraph 1 and/or Proteomic data processing and quality control).

We have updated the first paragraph of the results section (p5) and the renamed ‘Data processing, quality control and orthogonal assay comparisons’ section (p6) with the type of proteomic platform (antibody-based) and commercial name of the method (proximity extension assay).

3. In the description of baseline/demographic differences between the consortium vs. randomized participants, the authors are encouraged to report the results using a similar format for all associations (for example in the text some associations are described based on absolute differences between groups, some include only a p-value, and some include proportions in each individual group rather than absolute differences in each group).

This confusion is partly driven by differences in data types (proportions vs quantitative) being compared. These data are much clearer in ST1 rather than in verbose text. We have simplified the reporting to include only significant findings and refer readers to ST1 (which has been expanded in light of reviewer feedback) for more details.

4. In Figure 1D the authors arbitrarily bin age – can age (as a continuous variable) vs. glycodeclin levels be plotted? While circulating levels clearly decline among women until the age of ~55-60, the authors assert that levels rise among men. However it is not clear from the plot there is any meaningful change across the age bins. In general, given the large sample size, the authors are encouraged to consider the biological (in addition to statistical) meaning of the associations that are highlighted.

We have replotted Figure 1d with age as a continuous variable (with FSHB added from the expanded analyses as an expected finding from known physiology). The observed increase in glycodeclin levels in males is indeed small, but statistically significant, owing to the large sample size. We have emphasized the changes in males are “minimally increased with age” (p8), and have added the linear effects for males to the plot. Throughout the manuscript, we have indicated OR and effect sizes where biological effects should be emphasized.

5. Line 267 – Figure 1D does not appear to show mean total SNP-based heritability as indicated in the manuscript text. This reference appears to correspond to Extended Data Figure 6.

We have corrected this in the text (we now refer to Extended Data Figure 7 in the revised submission).

6. In “Biological enrichment for proteins with multiple trans associations” the authors use CR2 as an

example of a gene with multiple trans-associations which are enriched in specific biological pathways. It would be helpful provide some context (eg. to note the CR2 encodes the complement receptor 2) in order for the reader to better interpret the pathway enrichment.

We have added the additional context in parentheses: “CR2 (complement receptor 2, expressed on B-lymphocytes)” with a line at the end to indicate the role of CR2 in B-lymphocyte signaling.

7. Lines 344-354 – the authors suggest that because rs2234962 (located near HSPB6) is associated with circulating BAG3 and downstream biomarkers of heart failure (in contrast to the primary cis-pQTL rs35434411), HSPB6-BAG3 may be an important complex in the development of heart failure. While this reviewer agrees, the presentation of this logic could be clarified (a diagram may be helpful). Further, the authors note that rs2234962 is a missense variant – does this coding variation affect formation of the HSPB6-BAG3 complex?

We have revised and extended the text around the BAG3 (Main text, p20-21), incorporating the expanded proteomic dataset, which reinforces the evidence supporting HSPB6-BAG3 as an important complex for the development of heart failure. To aid interpretation, we have added a schematic diagram (Extended Data Figure 8) to outline the putative effects of rs2234962 (associated with cardiomyopathies) vs the primary rs35434411 pQTL variant. We have also noted the close proximity of the missense variant to the HSPB binding IPV motifs in *BAG3*, highlighting how it affects interactions with other HSP/HSBPs in a recent, albeit small, *in vitro* study. Further large-scale functional experiments are needed to confirm how the missense variant leads to altered interaction of BAG3 with HSPB6 and other HSPBs (not measured here).

Notably, in addition to the HSPB6 trans pQTL at the BAG3 locus (rs2234962; Cys151Arg), we found trans associations for MB, MYOM3, MYBPC1, MYL3, proBNP, and NT-proBNP. BAG3 functions through BAG3-HSP70-HSPB complexes, which play an important role in heart failure and cardiomyopathies⁴⁶, including the same BAG3 signal (rs2234962) in previous GWAS of cardiomyopathies^{47,48}. ProBNP and NT-proBNP are established biomarkers of heart failure and cardiac damage⁴⁹ whilst MB, MYOM3, MYBPC1 and MYL3 are all myocyte (MB)/myofibrillar proteins. The rs2234962 pQTL is an independent secondary cis pQTL for BAG3 levels from the primary cis pQTL (rs35434411; Arg71Gln, Supplementary Table 16), for which we did not find significant evidence of association with the aforementioned proteins after adjusting for the number of proteins tested ($p < 1.5 \times 10^{-5}$). The rs2234962 missense variant sits in between two conserved IPV motifs which are essential for HSPB6/8 binding⁵⁰ and may potentially modulate interactions with HSP/HSBPs *in vitro*⁵¹. Taken together, these results provide evidence of different mechanisms of effect driven by different variants in BAG3 (Extended Data Figure 8), with the rs2234962 missense variant potentially affecting both BAG3 levels and BAG3-HSPB6 complexing, leading to specific downstream perturbations in cardiac muscle proteins, downstream blood biomarkers of heart failure and potentially risk of cardiomyopathies.

8. Line 423 – missing “and” between “eQTLGen” and “GTEx”

In our revised manuscript, we have updated our methods for performing colocalization to account for regions that contain more than one causal variant (SuSiE+ coloc framework). Because we do not have a suitable LD reference panel for eQTLGen, we have replaced this analysis with whole blood from GTEx and removed all mentions to eQTLGen in the text. The key findings remained consistent.

9. Line 429-432 – it would be helpful to present the numeric results consistently [eg. % (X/Y)] within the entire colocalization section. It’s not entirely clear what proportion of colocalized proteins colocalized within a single tissue (eg. is it $191/507 = 38\%$?).

We have added denominators to all percentages in the eQTL colocalization section to be consistent with the rest of the manuscript. For clarity, we have removed the statement about colocalization within a single tissue in the revised manuscript.

10. Line 441 – the blood-brain barrier explanation would seem to generalize to broader sets of tissues/organs with variable/differential blood permeability.

We have updated Extended Data Figure 10 to better display the directional concordance rate between pQTL and eQTLs across different GTEx tissues. We have also updated the text to better generalize the blood brain barrier explanation. Main Text (p25):

We observed the lowest directional concordance rates in tissues from the brain (Extended Data Figure 10a), including the cerebellar hemisphere (64%; 85/133) and cerebellum (68%; 100/148), and the highest in the liver (90%; 146/141), which could potentially be explained by factors affecting access to circulation such as the blood brain barrier.

11. Lines 786, 792 – “multivariate” should be “multivariable” as only a single outcome was used in the regression model.

This has been corrected throughout the text.

12. The blood group-FUT2 secretor interaction plot (Figure 5A) and associated manuscript text is confusing as presented. The main point of this plot/text is to demonstrate that secretor status modifies the effect of blood group on protein levels. As presented, the boxplots facilitate comparison of protein levels between secretors and non-secretors within each blood group. However, the authors draw attention to differences across both blood group and secretor status. Visually, this information is very condensed within the current plot, and it may be useful to instead facet each subplot (for a total of 8 subpanels) (or group the x-axis) by secretor status rather than blood group.

We apologize for the confusion caused here. We have revised this section with insights from the expanded dataset, improving wording and adding contextualization for the top associations (findings for MUC2, FAM3D and ALPI are all from expanded data). The evidence of GI enrichments and involvement are reinforced further with the expanded data. We have replotted with the axis grouped by secretor status (Figure 4a) which indeed makes the message much clearer along with the text changes. The revised and expanded text is now in Supplementary Information (p29-30).

Referee #2 (Remarks to the Author):

A. Summary of the key results

Sun et al. present the currently largest genome-wide analysis (GWAS) of protein levels determined in blood plasma using an affinity proteomics approach on > 50000 samples collected by the UK biobank. The work presents a compendium of investigations as a resource. It confirms previous observations of direct genetic regulation of 1500 plasma protein levels (cis) and adds key insights into how distant (trans) genes and networks contribute to regulating protein levels.

B. Originality and significance: if not novel, please include reference

The work is the first major output from a consortium built by partners from the pharmaceutical industry and presents a major expansion of previous proteins GWAS. The authors present that the increased sample size and a newer proteomics approach allowed them to identify new pQTLs. While the discovery of cis-pQTLs plateaus in around 10k subjects, the number of trans-pQTLs and utility of such associations continues to increase with added sample sizes. This finding alone opens the gate for novel investigations of the dynamic architecture of the circulating proteome. It also points to the necessity of conducting even larger plasma proteomics studies, hence following the path of how genetics made it possible to increase our understanding of human biology.

To reach a wider audience, the author should investigate and explain why trans-pQTLs could not be found at the depth before and whether this is driven by only the sample size or could be a consequence of their inclusion criteria for selecting samples.

Many thanks to the reviewer for this helpful suggestion. In the revised manuscript, we have added ancestry-specific analyses and revised our subsampling analyses; these analyses show that the observed gain in *trans* pQTLs is mostly due to sample size increases, since (1) we saw no global systematic shifts in pQTL detections in for other ancestries; (2) our discovery cohort (EUR, randomised baseline samples of UKB) and combined cohort analyses (all UKB-PPP, including non-random sample selections) are very consistent; and (3) the variance explained slowly but steadily increasing with sample size beyond 10k samples shows how larger sample sizes can help us detect more *trans* associations with smaller effect sizes; we also show in one of the deep-dive examples (IL15) that many of the functionally linked *trans* pQTLs are only detected at large sample sizes (Supplementary Information, p22).

We have revised the *trans* associations with sample scaling in the results (Main text, p23-24), Supplementary Information (p22-23), and added the following to the revised Discussion (p29):

We found that the discovery of cis pQTLs is saturated to the number of proteins tested after ~10,000 samples. Although trans association discoveries continue to increase, the variance explained by trans loci increased at a slower rate beyond 10,000 samples. We anticipate most gains from future, larger-scale studies to be driven by the detection of trans associations with smaller polygenic effects, rare associations, and associations with proteins not previously tested. We also note that the natural upper constraint of detectable proteins will vary between samples and tissue matrices; non-blood based proteomic studies are needed to explore this in greater detail.

The added ancestry-based analyses and results are now in Main text (p14-15), Supplementary Table 11, Figure 2e inset, Extended Data Figure 5b and c)

Regarding the highlighted enrichment of SLE, which was only due to 500 samples (1% of the study set), is this a coincidence or a conscious choice? Please clarify if relevant. Other phenotypes were more prevalent in numbers but less strongly enriched. Also, confirm if the stated diagnoses were made at any time point of participation or during the sampling.

The highlighted SLE enrichment in UKB-PPP was driven by consortium selection. Each of the 13 consortium members pre-selected approximately 500 participants. Therefore, different diseases or potential characteristics may be enriched to varying extents compared with the full UKB cohort. Due to the reasonably small number of samples the consortium could pre-select, any specific selection for diseases/other traits will impact rare diseases/traits proportionally more than common ones.

Hypothetically, for a ‘disease X’, with 1% prevalence in 500k (5,000 total cases in UKB: ~500 cases in ~50,000 UKB-PPP randomly sampled participants), if 1,000 consortium-selected samples were for ‘disease

X', then there would be 1,500 'disease X' cases in UKB-PPP (assuming no sampling overlap), representing an enrichment by $\sim 3\times$ ($p=1.5e-250$). On the other hand, for a common 'disease Y', with 20% prevalence, there would be 100,000 cases in all of UKB, with $\sim 10,000$ cases in UKB-PPP if randomly selected. Thus, if 1,000 consortium selected samples were for 'disease Y', there would be 11,000 cases, representing a much smaller enrichment $\sim 1.1\times$ ($p=7.0e-26$).

Unfortunately, we were unable to comprehensively assess why certain samples were pre-selected at certain companies. Therefore, we performed our discovery pQTL analysis using the randomized baseline portion (excluding consortium pre-selected samples) to retain representativeness. Diagnoses were from any time since enrollment in the UK Biobank study; we have added this detail to the Methods.

C. Data & methodology: validity of approach, quality of data, quality of presentation

The study presents data from one of the world's most exquisite and well-studied population cohorts, the UK biobank. The work appears as the first study to use UKB blood samples for large-scale proteome analysis. Some co-authors and community members have used similar but smaller-sized population UK-based cohorts to conduct protein GWAS. It would be interesting to know if - by coincidence - any among the 50k subjects were also participating in any of the previous studies.

Data on the overlap between this UKB-PPP study and comparable UKB-based proteomic studies are not available due to confidentiality within and between independent cohorts. Rough estimates suggest any overlap between studies would be marginal with little no meaningful impact on results. The UKB-PPP generated data for approximately 0.2% of the UK population aged 40-70 at time of enrollment (2% enrolled in UKB, 10% of which included in UKB-PPP). The exact amount of overlap between studies would be influenced by inclusion/exclusion criteria, outreach strategies, individuals' interests in participation, and other factors which are difficult to estimate. Nevertheless, it is reasonable to expect the level of overlap to be very small. All analyses described in this manuscript (excluding ancestry specific analyses) were based on very large sample sizes and thus, were unlikely to be impacted by potential overlap with other UK-based cohorts.

The blood samples were analyzed by the recently established Olink's Explore platform, which is now capable of detecting 1500 circulation proteins. There is an extensive description of how the Olink data batches were processed to perform the presented analyses. What is missing is a global analysis of the data in PCA, UMAP, or similar types of dimensionality reduction approaches. These would help understand if any global pattern exists and if these can be explained by any of the chosen traits, ethnic, center, or technical aspects.

We thank the reviewer for these helpful comments. In the revised manuscript, we have now included a global analysis of the data using Principal Component Analysis (PCA). Regarding age of sample, because sample age is a continuous variable, we examined potential "sample age effect" using an approach like batch effect; that is, we fit a simple random-effects model on each protein and calculated the percentage of variability attributable to age of sample. Please note that in the updated analysis, we incorporated the additional proteins measured in the expanded data. The details of the analyses along with corresponding figures are added to Supplementary Information (p14-16). In brief, we did not observe obvious clustering/outliers in the global results for sex, ancestry, centre, or sample age. Batch and plate effects are examined in Supplementary Information (p9-12). We also performed additional analyses to detect potential place effects as requested by another reviewer. Overall, we did not detect batch or plate effects.

D. Appropriate use of statistics and treatment of uncertainties

My current understanding of the UK biobank setting is that blood samples have been taken at different centers across the country and originate from donors recruited over four years. It is, though, unclear when the blood samples were taken and how these were processed.

The authors should investigate if the sampling centers contribute to any differences in protein levels and if genetics, lifestyles, or demographics can assist to explain the expected center effect. Since the detection of protein levels may be influenced by the time elapsed between sampling and analysis (age of sample), appropriate analyses should be conducted to annotate the protein.

We did not detect systematic sampling center effects as shown in our response to the previous question; we examined the distribution of PC1 and PC2 of each of the sampling centers and did not find a particular center that was out of range of the others. We did not detect a global effect of sampling age, either, as demonstrated in our response to the previous question, in which we calculated for each protein the percentage of variability attributable to sample age and all except about 2 dozen proteins had percentages greater than 10%. We have added to ST3 the annotation for each protein the effects of sample age. As a precaution, we still incorporated sampling centers and sample ages in all downstream analyses to account for any potential confounding effect(s) from these two variables.

To enrich their resource, it would be helpful to determine the most and least variable proteins across the cohort, age groups, and sexes. Please also investigate if their heterogeneity of variance and if an annotation of their provides hints to explain the genetic variation and environmental or lifestyle effects.

We have determined the most and least variable proteins across age, sex and the additional factors requested by the reviewer. Based on these analyses, we believe that the cohort differences are also influenced by these factors mentioned above which are more interpretable, thus we did not do any additional cohort differences due to large numbers of unmeasured/potential confounder and colliders.

In the second part of this request, we believe that the reviewer may be suggesting that we leverage variance pQTLs to detect epistasis and environmental effects, as outlined by *Brown, A.A. et al. (2014) Genetic interactions affecting human gene expression identified by variance association mapping. Elife, 3, e01381*. Although we believe that this is an interesting topic, it is out of the scope of the present study. In the revised manuscript, we outlined several key non-genetic analyses – including proteomic associations with prevalent diseases and liver and kidney functions – providing confidence in the biological validity of the proteomic measurements. We also conduct several technical validations, including correlation analyses with immunoturbidimetric assays. We believe that an investigation of variance pQTLs, systematically exploring potential gene-by-environment effects, would be sufficiently detailed to warrant its own independent, self-contained study.

Proteins may be secreted, leaked, or shedded into the blood. Do the authors see any systematic enrichment for these categories or their abundance levels regarding genetic regulation?

We have now tested all proteins with *cis* pQTLs for enrichment/depletion against all tested Olink proteins as background (to avoid confounding, given that the assay is not a random sample of the human proteome). Indeed, as the reviewer anticipates, we see enrichment for a higher proportion of secreted proteins with pQTLs. We have also examined enrichments for membrane/cytoplasm/nuclear and other (miscellaneous) compartments. We have added these results to the Main text (p. 12).

For proteins with *cis* pQTLs, we found significant enrichments ($p < 0.01$, correcting for 5 categories) in the proportions of proteins that are secreted (odds ratio [OR]=1.52, $p = 1.1 \times 10^{-11}$), and depletions

in cytoplasm (OR=0.76, $p=1.6 \times 10^{-5}$) and nuclear proteins (OR=0.55, $p=2.2 \times 10^{-12}$) compared against the assay background. These enrichments and depletions were attenuated when considering proteins with any pQTL, but enrichment for secreted proteins (OR=1.22, $p=6.5 \times 10^{-4}$) and depletion for nuclear proteins (OR=0.80, $p=0.0032$) remained..

When investigating the effect of blood cell counts, please check if the proteins are - exclusively or primarily - expressed in blood cells or other organs. See <https://doi.org/10.1126/science.aax9198>.
[doi.org]

We have performed these analyses, updated our Supplementary Table (ST24), and added a table in the Supplementary Information to incorporate this along with the following text in the Supplementary Information (p24):

We cross-referenced the 2,415 proteins with pQTLs for enrichment in various tissues including blood reported by Uhlen et.al¹² (Supplementary Table 24). Within the list of 2,415 proteins, 3 proteins (CLEC4C, IFNL1 and MAP1LC3B2) were enriched in blood and 34 other proteins were enriched in more than one tissue including blood (Table S3). The list of these proteins along with the tissues and the specific distributions are detailed in Supplementary Table 24.

The authors focused on genetic regulation and only briefly touched upon clinically measurable traits such as age, sex, and BMI. Even partly outside the current scope, it would be highly interesting to know if their protein level data can predict age, sex, and BMI. This may help to model these traits solely on experimental data and assist future studies to use the predicted values for trait adjustment. Would it also be possible to predict blood types from such high-dimensional data?

We thank the reviewer for this suggestion. Such an analysis would complement our results by validating established or widely reported biological associations from the proteomic data. In the revised manuscript, we incorporated a cross-validated LASSO prediction model and assessed its performance in a held-out dataset for age, sex, BMI, liver function (ALT, AST) and renal function (eGFR) in addition to ABO blood groups. Indeed, protein levels can predict the aforementioned traits to a considerable degree. We observed good predictive performances for O, A, B blood groups ($F1 > 0.9$), but not AB blood group ($F1 = 0.44$) – this is not unexpected due to the severe case imbalances and low prevalence for AB blood group (3.6%). Much more accessible and accurate genotyping data can be used for blood group typing if possible. The results are in updated Main text (p10-11), Extended Data Figure 3c and Supplementary Table 8).

Even though age, sex, and BMI traits are the major drivers of differences in the plasma protein levels, there are obvious reasons why the data has not been studied for disease-specific biomarkers. However, it would be valuable to know if other binary (yes/no) traits collected at samplings, such as smoking, disease diagnosis, or use of (any) medication, can be investigated to enrich the current list and reveal noteworthy proteins.

We thank the reviewer for acknowledging the myriad reasons why the data have not been systematically investigated for disease-specific biomarkers. We have included the proteomic associations with the traits suggested and included associations with the twenty most prevalent ICD10 conditions, as outlined in Supplementary Table 2. We identify many intuitive biological associations, as highlighted in the revised version of the main text. We believe that the addition of these proteomic associations with health states and diseases further validate the value of this dataset. However, more systematic, in-depth interrogation of all significant proteins is beyond the scope of this paper and is best left to the broader UKB research community and individual industry partners as part of follow-up studies and manuscripts.

The updated results and text can be found in Main text (p8-9) and Supplementary Table 7.

For the proteomics community, studies of 50k subjects are currently still extremely rare, and increasing sample size may inflate some of the initial observations. Please clarify the approach used to manage proteomics data at that scale.

We thank the reviewer for these comments, although the mechanism by which increased sample size could inflate initial observations from smaller datasets is somewhat unclear. Usually for true positives, significance will increase with larger sample sizes, while false positive will regress to the null. We show both in genetic and non-genetic analyses a wide range of biologically sensible results, and replicate many well-known associations. Our choice of significance thresholds for multiple correction on the total number of proteins tested is likely over-cautious and over-penalizes our number of significant associations, since we ignore correlations and non-independence between proteins. Since the full summary results will be made available, we leave it to the users to determine alternative significance thresholds, if appropriate. The present study's large sample size affords us robust statistical power, facilitating higher-confidence interrogation of prior results. As the field moves towards applying proteomics at larger sample sizes and integrating broader-capture, high-throughput technologies, several broadly applied statistical practices should address potential false-positive findings – most notably, Bonferroni adjustment for multiple comparisons, which has been applied in the present study for both the genetic analyses ($p = 0.05$ divided by the approximate number of genetic loci tests) and the cross-sectional proteomic analyses ($p = 0.05$ divided by the number of proteins tested). Although the sample size is large, management of data generated using the Olink proximity extension assay should be relatively straightforward as opposed to genetic, imaging or other mass spectrometry data, as the proteomic data generated is essentially a matrix of ~55k rows by 3k columns that can be imported into most statistical analysis packages (e.g. R, Stata) without issues. The consortium is currently collaborating with UK Biobank to upload these data to their DNANexus platform, which will provide users with an accessible graphic user environment in which to interrogate the dataset.

E. Conclusions: robustness, validity, reliability

The authors discuss conducting more plasma proteome analysis on UK biobank samples to validate the presented observations and findings. A cross-platform comparison may follow up on the presented work to investigate further aspects of the plasma proteome that cannot be measured by the Olink approach. What do the authors expect in terms of validation, and are there any features in the data, such as MAF or effect size, that would make the current findings more robust?

The planned cross-platform comparison will likely involve a much smaller relative sample size (~2500 samples) compared to the present study. Thus, it is unlikely that we can employ pQTLs as validation tools across the platforms, given that genetic association studies of the protein concentrations would likely be under-powered. Minor allele frequency and effect size can be misleading features to use for validation. We believe that orthogonal comparisons of different proteomic assay methods in the same samples are needed to decipher complementarities and uniqueness of different approaches and also decipher cohort specific effects vs effects translatable across cohorts especially for trans associations.

We have exercised caution in ensuring robust summary results by excluding rare variants, including a replication cohort, and maintaining a tight Bonferroni significance threshold.

However, at this stage, the cross-platform study has not commenced. Hence, to avoid confusion and conflicts, we have removed references to this comparative study in the revised manuscript discussion.

The presented data is “limited” to the Olink approach that measures 1500 proteins. As mentioned by the authors, another 1500 proteins have become available on the platform, and these would be measured in the UK biobank samples. Will the authors implement additional data, or is there a rationale to include the other 1500 proteins in a separate study?

We thank the reviewer for highlighting the value of including the additional 1500 proteins in this manuscript. These data were not generated or readily available at the time of initial manuscript submission; however, they have since been generated and made accessible to all 13 consortium members. Thus, the revised manuscript now includes results from an analysis of the complete set of protein measures – comprising the original ~1500 analytes from Phase 1 of the UKB-PPP project in addition to ~1500 analytes from Phase 2. We believe that it is most valuable for the scientific community to gain access to all summary results as early as possible.

The authors acknowledge the use of a primarily white population, which limits the generalizability of some current observations. However, would ethnicity be detectable in the protein level data?

We have expanded the main pQTL analyses to other ancestries. Due to the severe case imbalances and confounding of other ancestries through potential disease/trait selection/prevalence differences, we cannot reliably predict ethnicity from the proteomic data. We believe integration of these data, given their open access nature, with other non-European cohorts would be better powered to explore ancestry specific proteomic signatures.

F. Suggested improvements: experiments, data for possible revision

For the time being, the authors use existing data to confirm their novel aspects. It remains partly unclear how the data could be validated by non-plasma proteomic efforts. Especially a systemic investigation of the trans-regulation of secreted proteins may pose a challenge but create new opportunities and push the boundaries for studying human biology. The author should clarify which experimental routes and possibilities exist to study these cases. In addition, the authors should provide examples of how they would select targets. For example, is a larger effect size a meaningful guide to prioritizing targets of interest and reducing the need for larger sample sizes?

The reviewer makes a good suggestion to conduct functional follow-up of *trans* pQTLs for novel protein interaction/coupling discovery and to assess the value of non-plasma efforts in disentangling plasma proteomic effects vs *trans* effects active in other sample matrices. We have added the following language to the discussion to provide some guidance on *trans* pQTL features for downstream follow-up:

We highlight several novel *trans* pQTL effects on protein interaction partners and *trans* pQTLs with large effect sizes, providing coding variant candidates for downstream functional experimentation, investigating genetic effects on protein conformation(s), complexing, and interactions. We also describe reciprocal and bidirectional protein interactions with *trans* pQTLs, providing additional candidates for experimental deep dives. The proportion of *trans* pQTL effects observed in blood plasma that may be reflected across other tissues or sample matrices remains to be elucidated – emphasizing the need to perform population-scale proteomics beyond blood⁶⁸. We hypothesize that many of the *trans* pQTLs affecting pathways active in blood, such as complement or haematological pathways, may not be seen in non-blood based proteomic studies, whereas non-blood based studies may still detect *trans* pQTLs with larger effect sizes for proteins that are present in those matrices⁶⁸.

The work represents a fantastic resource and - at this level of importance - would benefit from an accessible portal to host the summary statistics. It would be highly valuable to develop an app or interface in a managed environment to allow the community to access and browse their favorite protein or association of interest and thereby increase the exposure and utility of the data and findings.

We thank the reviewer for this helpful suggestion. We have now implemented a detailed web interface for fast query, tabulation and visualization of our results. In addition, we have provided bulk download options for those who may be interested in high-throughput integration and deep-dive systematic analyses.

The breadth of data provides many opportunities to present different observations, but it also limits the possibilities to focus on core findings. The collection of observations tends to appear as a report rather than a resource that can be used by the community. Some guidance on how to visualize the data and utilize the resource would be suitable. Even though many aspects can be lifted, it remains unclear what the authors deem as the major or novel findings that exceed those made in previous larger-scale proteins GWAS.

We have revised and restructured our manuscript to highlight the key findings revolving around *trans* pQTL networks. We highlight the added value of our resource in improving estimates from common downstream approaches, including wider and stronger genetic instruments for MR, facilitating multi-protein/multi-trait colocalization, insights on the inflammasome pathway in context of previous pQTL findings, and long-range genetic epistasis ABO-FUT2. These vignettes have been revised with more detail and added data, and many have been moved to the supplementary information for conciseness. With results of this scale, we believe that a more dynamic, interactive interface, as the reviewer suggested above, will be more valuable than a limited set of static figures provided in the manuscript.

Figure 3 presents very appropriate examples of protein regulation. Most proteins are well-known, fairly abundant, and important for blood function. Please investigate protein level data for these example pathways, such as by stratifying for an appropriate genotype and displaying the proteomics consequences downstream.

Given the large network of associations illustrated in Figure 3d (complement pathway), we could not feasibly include genotype effects for all associations in the complement cascade. To address the reviewer's request, we have added Figure 3b (IL15 signalling) and Figure 3c (*cis* and *trans* effects for the bidirectional *trans* pQTL pair) with proteomic levels stratified by genotype. We have also created an additional Supplementary Table (ST22) that highlights reciprocal protein associations, illustrating these in a more interpretable format than verbose text.

G. References: appropriate credit to previous work?

The included references were found appropriate.

H. Clarity and context: lucidity of abstract/summary, appropriateness of abstract, introduction and conclusions

The title "Genetic regulation of the human plasma proteome in 54,306 UK Biobank participants" is not very appealing (and connects with previous work from Sun et al in 2018: "Genomic atlas of the human plasma proteome"). Isn't the utility of the plasma proteome architecture or the findings made via the

trans-pQTLs worth being highlighted?

We appreciate the reviewer's recommendation for a more appealing title, reflecting both non-genetic and *trans* pQTL findings. The new title of the manuscript is: "**Plasma proteomic profiling of 54,219 UK Biobank participants reveals novel insights into genetics, health, and disease**".

Referee #3 (Remarks to the Author):

Thank you very much for the opportunity to review this paper.

Sun et al provide results from a pGWAS of 1,463 proteins measured using the Olink Explore assay in 35,571 European-descent individuals of UKBB, with an additional, independent set of 18,181 UKBB participants (different ethnicities) for replication. This is the first phase of the commercially funded UKB-PPP with results for another 1,500 proteins being imminent, and hence this paper is somewhat preliminary. Overall, this is a very well conducted and written study, and I would like to congratulate specifically the leading but also all authors for steering this multi-partner industry consortium jointly to successful completion in a short timescale.

The discovery arm of this study is of comparable size than the earlier Decode study based on Somalogic's aptamer technology that covers around 5000 protein targets (PMID: 34857953) and therefore not unprecedented. Strengths of the current study include division into an independent discovery and replication set, and conduct of detailed sensitivity, enrichment, and interaction analyses. I like the systematic exploration of the consistency of effect directions for shared expression signals and putative explanations of discordant directions. The study's most important weaknesses include the total omission of a) any systematic evaluation of the downstream clinical consequences of the identified pQTLs to provide biological and translational insights,

b) the utilisation of the ethnic diversity that exists within UKBB (and the selected UKBB-PPP samples).

Directly addressing point a), and for b) adding results that address the applicability of (at least *cis*) pQTLs from European-descent participants to other ethnic groups plus an attempt to identify ethnic specific signals, are in my view essential to provide new insights worthy of this large-scale, ambitious effort.

We thank the reviewer for their kind words and constructive feedback.

To address point (a):

We have now incorporated data from Phase 2 of UKB-PPP, effectively doubling the number of proteins analyzed. We appreciate that the omission of systematic Mendelian randomization or colocalization analyses limits downstream clinical insights. As this was a pre-competitive effort, the companies chose to focus our efforts on biologic relevance and relationships to disease with proteomics, rather than focus on methods for target discovery like Mendelian Randomisation. Systematic colocalization and Mendelian Randomization are also vast analytical undertakings, and individual consortium members and the wider scientific community are better placed to conduct these downstream analyses.

Considering comments from other reviewers, we have focused our key findings on the *trans* pQTL protein networks that are beginning to emerge from our large-scale analyses, providing examples where *trans* network effects are not detected at smaller sample sizes from our sub-sampling analyses. We have revised the manuscript to place greater focus on the broadened set of *trans* pQTL networks identified via the expanded Olink dataset, which have, to date, only been limited to small sets of receptor-ligand pairs in the

published literature. We provide novel mechanistic insights into the NLRP3 inflammasome and complement cascade and expand examples of how *trans* pQTLs can inform specific disease risks. For example, we provide examples of extensive epistatic effects on protein levels in the form of ABO-FUT2 interactions; the expanded dataset reinforces the relevance of this effect on key gastrointestinal proteins (MUC2, FAM3D, ALPI), which may explain some of the pathways underlining GI pathologies linked to ABO/FUT2.

We have moved examples of how the scientific community can leverage our summary results for downstream (multi-trait) colocalisations and MR to the Supplementary Information. We believe that the expanded set of biological insights gleaned from *trans* pQTL networks, the inclusion of new, non-genetic analyses (including proteomic associations with prevalent diseases and proteomic prediction modeling), and the construction of an open-access browser for full interrogation of our findings should sufficiently highlight the translational potential of this large-scale proteomic dataset.

For point (b):

We thank the reviewer for highlighting the importance of including participants of non-European ancestries. Despite the limited sample sizes in non-European ancestries, we should at least begin to address this gap and help facilitate future meta-analyses and downstream analyses requiring ancestry specific LD and summary data.

In addition to European ancestry results and the full cohort results (which we included in the first draft, but have made more direct references to in the main text of the revision), we have:

- Included non-European, ancestry-specific pQTL analyses based on the Pan-UKB defined ancestry groups, and
- Summarised the key results from non-European ancestry analyses in a dedicated section of the Results (p14-15)

Given the much smaller sample sizes, most of the associations were, unsurprisingly, already observed in the European analysis. However, we identified a small collection of pQTLs that appear to be enriched in populations of non-European ancestries due to the allele being more enriched in non-Europeans/near absent in EURs. Some examples are discussed in more detail in the added results (p14-15). This includes examples of pQTLs that are depleted in frequency in Europeans and hence not found.

Effect sizes were well-aligned and the number of detected pQTLs was in-line with expectations for comparable European analyses of similar sub-sample sizes, suggesting that no specific ancestry systematically out-performed the other. We also found that MAF of primary pQTLs in African ancestry was enriched against European not seen for Central/South Asian vs Europeans despite similar sample sizes, suggesting African ancestry potentially provides additional signals over Europeans compared to Central/South Asian ancestries.

Although we found that there is limited gain in pQTLs from non-EUR ancestries due to small samples sizes, we observed indications of potential gains in pQTLs at expanded samples for variants that are enriched in non-European ancestries. Thus, there is likely added value in conducting large-scale pQTL mapping in non-European populations.

The added ancestry-based analyses and results are now in Main text (p14-15), Supplementary Table 11, Figure 2e inset, Extended Data Figure 5b and c)

We appreciate the reviewer's comment regarding the comparable sample sizes of our discovery pGWAS and the sample size of deCODE's most recently published analysis. However, we should emphasize that

the deCODE study did not implement a discovery/replication design, and is more comparable to our combined cohort analysis, for which we have an approximately ~50% larger sample size. Our reporting is more conservative in terms of significance thresholds. We also compare our findings with previous pQTL studies, including the deCODE study.

Given this and the numerous previously published pQTL studies (all those considered are helpfully listed in the supplement), it is maybe not surprising that the current effort only identifies 562 new *cis* pQTLs. This is in line with the author's observation that the number of *cis*-pQTLs starts to plateau after sample sizes of around 5k. The sample size of the UKBB-PPP therefore largely enhances detectability of new *trans* pQTLs, many of which are extremely pleiotropic, something that has been obvious in earlier studies. *Trans*-signals are therefore what most of the reported novel associations can be attributed to (9,309 of all 10,248 associations).

We believe our novel *cis* and *trans* pQTL findings still present a considerable gain in the field, as we show in our subsampling analyses. Whilst *trans* pQTLs are increasingly detected and replicated in the literature, investigations of *trans* pQTL networks have been limited in previous studies, especially in the form of pathways (such as IL15 and complement proteins). We agree with this reviewer and other reviewers' comments regarding the key insights to be gained from *trans* pQTL associations and have refocused and extended our manuscript to focus on these *trans* pQTL characteristics.

The authors mention the successful exome sequencing and ongoing whole genome sequencing of UKBB. Why were the existing sequencing results not considered, at least for *cis* variation? This could have provided a more substantial advance in knowledge.

We have decided to focus on a systematic source of pQTLs using the imputed data for this reference resource. Imputation covers MAF very well down to ~0.1%, as seen here; the addition of exome sequencing adds mostly to the rarer MAFs, which this study would be underpowered to detect in its replication stage, and most likely susceptible to false positives. Currently, different sample sizes are covered by WES and WGS studies - typically less than the imputed data, thus making downstream integration and interpretation problematic, with the risk of introducing artefactual associations when combined with imputation results directly.

We have conducted previous studies in similarly sized sub-cohorts of UKB, whereby WES single variant analyses yielded limited added contributions compared to imputation (Genetic map of regional sulcal morphology in the human brain from UK biobank data, <https://www.nature.com/articles/s41467-022-33829-1>). Therefore, rare variant analyses may be best implemented using aggregation methods like burden analyses, leveraging the full UKB-PPP cohort to maximise power, at the expense of a replication cohort. Indeed, the UKB exome sequencing dataset was recently used to study rare variant effects on the plasma proteome in the full UKB-PPP cohort (<https://www.biorxiv.org/content/10.1101/2022.10.09.511476v1>). We believe that this study was sufficiently detailed and complementary to this study to warrant its own self-contained manuscript.

Apart from the important weaknesses above and my detailed questions and suggestions below, I have one other more major technical concern related to the number of reported independent secondary associations based on fine mapping. It is acknowledged that SuSiE tends to select credible sets that are non-overlapping but contain variants in high LD, which is likely to be an artefact of the algorithm rather than representing a true finding (see `run.susie()` documentation of the `coloc` R package). This can also lead to instances in which SuSiE selects a large number of credible sets. I would therefore suggest running SuSiE with an increasing *L*, i.e. maximum number of credible sets to identify the smallest *L* that includes distinct credible sets. I would also ask the authors to confirm statistical independence of variants in the

credible set using a joint linear regression model that includes all variants at once.

We apologize for the misunderstanding. As the reviewer correctly points out, SuSiE with summary statistics and an LD reference is prone to convergence issues and can potentially report non-overlapping, highly linked credible sets, often with a single variant in the credible set. However, we used the SuSiE regression method with individual-level genotypes and phenotypes which, in our hands, was less prone to artifactual signals. We have re-written the results and method sections to clarify that we are using the individual-level version of SuSiE with genotype and phenotype residuals adjusted for the same covariates as the marginal association analysis. We have also added an additional heuristic for removing credible sets that are highly linked ($r^2 > 0.8$).

Main text, p36: “We used the Sum of Single Effects regression (SuSiE, version 0.12.6) to identify and fine map independent signals using individual-level genotypes and protein level measurements from discovery set participants.”

Main text, p37: “We applied a post-hoc filter to remove credible sets in high LD with another credible set in the same region (lead variants $r^2 > 0.8$).”

In our previous submission, we were already incrementing the maximum number of credible sets to find the smallest L with distinct credible sets, starting with the default (L=10). Starting with a smaller initial L than the default (L=5) substantially increased the runtime, but did not decrease the total number of credible sets identified.

Main text, p37: “For test regions where SuSiE found the maximum number of credible sets, which was initially set at L=10, we incremented L by 1 until no additional credible sets were detected.”

We acknowledge the importance of confirming statistical independence of the credible sets, and have added exact conditional effects, standard errors, and p-values using multiple linear regression models of the most probable variants from each credible set for each region. We have modified Supplemental Table 16 to include these conditional results for regions with multiple independent signals, as well as the linkage (r^2) between each signal.

Main text, p37: “For regions with multiple credible sets, we assessed statistical independence by performing multiple linear regression using the most probable variants for each credible set and the same genotype and phenotype residuals.”

The sensitivity analyses testing the impact of blood cell composition, BMI, season, or fasting time on pQTL effects are useful and show minimal impact on protein levels for most genetic associations, as one might expect. This ought to not be confused with the strong effect of those parameters on protein levels. Since this is the first paper on proteomic measures in UKBB and not all readers will immediately appreciate this, I propose to expand this section to include a general investigation and description in the supplement of the effect of all of these parameters on protein levels. It would be useful if the authors included a figure with the variance explained of all proteins by each of these ‘sensitivity’ metrics in addition or contrast to the variance explained by other factors (sex, age, ethnicity, deprivation, cis and trans effects, etc).

We thank the reviewer for suggesting distinguishing the non-genetic effects (blood counts, BMI, season and fasting time) on protein levels from confounding effects on pQTLs. To address this question, we first created a reference model regressing the demographic covariates used in GWAS (age, gender, age*gender, age², age2*gender, ancestry and 20 genetic PCs) on protein levels. We then calculated the difference in the variance explained by the reference model and a multivariate model including the base covariates and each of the non-genetic factors separately to evaluate the percentage of variance in protein levels explained by the non-genetic factors alone. We have added detailed summaries of variance explained by each of these models and difference with respect to the based model as a supplementary table (ST25) and a figure showing these associations as a supplementary figure (also copied below for reference). On average, demographic covariates explain 3.6% of the variance in plasma protein levels, while blood cell covariates, BMI, fasting time, and season explain 4%, 1.5%, 0.06%, and 0.03% respectively. The summary of results are added to Supplementary Information (p25, Figure S14 see below) and Supplementary Table 25.

The initial paragraph on ‘biological associations’ appears a little superficial and doesn’t provide deeper insights. Given the investment of the consortium and findings of existing studies, one would rather hope ‘that the proteomic assay can capture physiological effects’. I wonder whether it would help if the authors expanded the exploration of these associations but moved this part to the supplement? I suggest to specifically investigate and add the impact of prevalent diseases and differences in liver and kidney function on protein levels with brief mention of key results in the main text.

We have integrated the reviewer’s suggestions, exploring the impacts of the physiological factors they have suggested, and have revised the text, tables and figures accordingly – Main text (p8-11), Supplementary Tables (ST7-8), Extended Data Figure 3. Along with a similar comment from another reviewer, we have trained and tested proteomic prediction models for key demographics, renal function (through derived eGFR estimates using the CKD-EPI equation) and liver function proxied by the liver function enzymes (ALT and AST) measured on the biochemistry assay in UKB.

The observed physiological effects of glycodelin (and now, from the expanded dataset, FSH) largely serve as positive controls; indeed, we believe this is the first time that such a non-linear sex-specific effect varying with age corresponding to female physiology is described in a populational proteomics context. Here, the

longitudinal effects across time can also be captured to some extent by cross-sectional protein measurements at scale.

The section on the interaction between ABO blood group genotypes and FUT2 secretor status could be strengthened by qualifying the nature and relevance of the reported differences/ interactions for the 38 proteins in more detail (directions and significance). The statement ‘We saw that the extent of differences in protein levels between secretors and non-secretors varied depended on the blood group for these proteins’ is redundant, i.e. a restatement of the earlier sentence. The following paragraph on gene expression enrichment doesn’t follow logically; could the authors please explain this better? I also suggest toning down and referencing the statement ‘(...) which may underline susceptibility to various FUT2/ABO associated GI conditions’.

In light of the expanded protein data, we have bolstered the evidence, expanded and revised the text and moved the section to the Supplementary Information to improve flow (p29-30). We have focused our contextualizations on the top associations from the expanded set of proteins (MUC2, FAM3D and ALPI), which show clear link to GI pathophysiology. We believe that the added data – demonstrating the roles of MUC2 and APLI in GI conditions, and the strong FUT2 FAM3D co-expression in Human Protein Atlas – provide additional insights and context, justifying the final summary sentence, which we have slightly toned down. As a result of relevance of FUT2 and the GI related proteins, it now follows more naturally that we test for enrichment of ABO-FUT2 interaction proteins for their expression in GI tissue (along with other tissues); our expanded data have led to even stronger enrichments for GI tissues across both humans and mouse.

The authors state that for 49% of primary cis associations, the index variant was in at least weak LD ($r^2 > 0.01$) with a protein-altering variant. I suggest adding this information for a less permissive LD threshold and compare it to what previous studies and technologies have reported.

We thank the reviewer for the comment and now state (Main text, p16) that of the cis-pQTLs, 512 (26%) were protein-altering variants or were in high LD ($r^2 > 0.8$) with a PAV. We note that these findings are in line with recent large scale proteomics publications.

The authors provide some details of the generalisability of the selected individuals, which is helpful. Could some more information be provided about the stratified sampling to clarify a) how this led to more efficient sample picking and b) whether or how this was accounted for in the observational and genetic analyses? Could the authors please expand supplementary table 1 to include recruitment centre, time since recruitment/ sampling, deprivation index, drug use (for selected commonly prescribed drugs), and other biological parameters of interest, such as biomarker levels etc.

The stratified sampling process in increasing sample picking efficiencies have been added in the Supplementary Information (p2-3) with reference (Allen et al 2021). The optimization steps to reduce number of plates for sample picking were not additionally adjusted for as any confounders were deemed randomly distributed across the optimization steps. The sample picking process for the randomised component is employed in all other UKB published and ongoing projects that required sampling, and we follow the same protocols as those in the added citation. Any potential stratifications/confounding based on age, sex, recruitment centres or sub-cohort selection were adjusted for in our primary analyses.

As the reviewer recommends, we have added the following to Supplementary Table 1:

- i) Recruitment centre (we were unable to provide a detailed breakdown by centre since some centres contained <25 samples, leading to identifiability issues as per UK Biobank guidance),

- ii) Time since recruitment in years,
- iii) Deprivation index,
- iv) Number of medications and major medication/supplement groups, based on UKB fields 137, 6153, 6154, 6155, 6179, 2492, which covers commonly used medications,
- v) Biomarkers – split between (1) haematology measures for routine clinically measured full blood count variables from 100081 and (2) Biochemistry from UKB category 17518.

We additionally have categorized ST1 into demographic, medications, haematology and biochemistry biomarker sections to improve readability.

The section on proteomic insights into COVID-19 associated loci remains rather technical without much biological insight or follow-through. Reference to earlier studies for the reported genes and potential explanations for any potential lack of replication could be discussed in more detail, as could the impact of the heterogeneity of case definitions.

We have moved the COVID-19 section to the Supplement Information (p31-32), as explained above, and have expanded the COVID-19 text to incorporate additional biological contextualization and references to related literature and orthogonal data. For TYK2, we have added an Extended Data Figure panel (EDF11c) to help aid interpretation with the (revised) text. We have also added a small section at the end to address limitations and impact of heterogenous case definitions.

The statement ‘highlights the strengths of the antibody based Olink® Explore Assay for pQTL detection and downstream biological discovery’ seems somewhat unsubstantiated. I suggest replacing ‘antibody based Olink® Explore’ with ‘broad-capture, affinity-based proteomics’. The value of this study lies in its scale and without direct comparison, i.e. comparative assessment in overlapping samples, inference about strengths or superiority of a given platform would best be avoided.

The reviewer’s point is well-taken. The text has been updated to read, “While our study highlights the strengths of broad-capture, affinity-based assays for pQTL detection and downstream biological discovery, further technological advances will enable more comprehensive population-scale investigations incorporating protein isoforms, proteoforms generated by post-translational modifications, and single-cell proteomic resolution.” In the revised text, this paragraph has also been expanded to include a broader discussion of aptamer- and antibody-based proteomics platforms, acknowledging that direct comparison between the platforms is strongly recommended, but beyond the scope of the present study.

The discussion highlights the need for orthogonal validation of antibody-based proteomics using aptamer- and mass spectrometry-based assays and the consortium seems to have initiated systematic evaluation of this. It would be reasonable to provide brief reference to earlier work addressing this issue (PMID: 34819519). Given that a ‘comparative’ preprint based on the UKBB proteomic data had to be retracted, it would be helpful if the analytical protocol and membership of this new effort was made public in advance, to avoid bias, commercially driven interests, and enhance transparency of process for the benefit of the scientific community. While the investment of the consortium is applaudable and will clearly provide large scientific benefit, the inherent value of the (mainly tax- and Charity-funded) resource itself is substantial, and there is an obligation to be conscious that even small statements can unduly influence the share value of listed proteomic companies.

We thank the reviewer for these helpful comments. Given the potential implications the reviewer has raised, we have removed the reference to the potential comparative study in UKB and have rephrased the language to focus on the scientific merits a generic study of this sort may add value to, with added reference to Pietzner et al. (*Nature Communications*, 2021), which provides an excellent overview of the benefits both proteomics platforms offer for scientific discovery, in addition to a recent paper by Katz et al. (*Science*

Advances, 2022), which provides additional perspectives on the issue. We agree that the planned multi-platform comparison study should provide a transparent analytical framework *a priori* to avoid biases/conflicts of interest. However, at this stage, the study has not commenced. Hence, to avoid confusion and conflicts, we have removed references to this comparative study in the revised manuscript discussion.

The retraction of the ‘comparative’ pre-print occurred as it was uploaded prematurely in error by the authors. A full explanation is provided on the BioRxiv pre-print server, which reads as follows:

The authors have withdrawn this manuscript because this paper was posted prematurely in advance of a UK Biobank Pharma Proteomics Project consortium effort. Therefore, the authors do not wish this work to be cited as reference for the project. If you have any questions, please contact the corresponding author.

All scientists involved in development of the analytical protocol, data analysis and downstream interpretation of results for the present study are named as co-authors on our manuscript. Individuals who provided key logistic and legal support are acknowledged in the Acknowledgements section. Thus, membership of this effort is already provided with full transparency.

Other points:

The description of Olink’s internal QC assessment in the plasma profiling section of the supplement is relevant (since it was adhered to) but reads a bit like a copy and paste of an Olink’s data report. More importantly, UKBB Expand QC parameter look almost too good to be true with extremely few sample or assay warnings, compared to what others have seen and reported for the same technology. Were any duplicates run, samples or plates?

We have provided details of the full sample processing and analysis protocols for transparency. The Olink internal processing pipeline largely follows from the standard protocols published by Wik et al ([https://www.mcponline.org/article/S1535-9476\(21\)00140-7/fulltext](https://www.mcponline.org/article/S1535-9476(21)00140-7/fulltext)), and thus remains similar to data reports and proteomic processing descriptions from prior Olink studies. There have been minor changes specific to the UKB-PPP project, in terms of the expanded set of proteins, sample set-up differences in UKB, and calibration of normalizations, which have been detailed in the Supplementary Information.

We appreciate the reviewer’s comment on the QC parameters. We have revised and simplified Table S1 in the Supplementary Information in terms of the percentage of data affected, rather than samples (since some assay/sample warnings affect a subset of proteins, and are therefore not reflected in the original over-optimistic tabulations) to give a more intuitive tabulation. This approach is in-line with prior studies, which usually do not report Olink warning metrics (e.g., Gyllensten et al, <https://www.mdpi.com/2072-6694/14/7/1757>). Our study followed the Olink QC procedures outlined in the above-cited protocol and in other Olink Explore studies. If over 1>/6 of samples failed per block/panel, Olink assessed the quality of the run and determined if a rerun was required.

We have taken several approaches to ensure data quality; these are now expanded throughout the revised manuscript. The additional technical validations, biological validations including comparisons with independent studies and genetic *cis* pQTLs all pointing to robustness of the data. Additionally, our median CV (6.4%) is slightly better in this dataset compared to a comparable dataset described in Katz et al. (<https://www.science.org/doi/10.1126/sciadv.abm5164>), and is in-line with the technical protocol paper (based on the 1.4k protein data). We have added the proportion of samples post QC filtering in

Supplementary Table 3 for reference, and have excluded one protein (GLIPR1) with outlying high proportion of samples failing QC.

The plot below summarizes intra- and inter-plate CV of Olink control samples and CVs of blind duplicate samples. It is important to note that Olink was not aware of the location of the biological sample when running the PEA technology. The CVs of blind duplicates (dark blue line) and Olink control samples (yellow line) are very similar if one takes into consideration that Olink's samples are located in wells A12 and B12, while the biological replicates are positioned randomly into the plate.

We have also expanded our comparison of the same proteins measured using independent approaches from the UKB biochemistry panel, which show strong correlations overall, boosting confidence in the assay processing and QC pipelines. We have also expanded our age, sex, and BMI association comparisons with other Olink/SomaScan studies, showing concordances despite differences in cohort, sample processing and analytical variations.

Each sample was run in singleton. We included blind duplicate samples (biological rather than the same technical samples run twice) from randomly selected individuals spread randomly across the plate as part of the QC pipeline. A small number of overlapping samples were used to calibrate normalization performances across batches. These have been described in the processing pipeline in the Supplementary Information.

Overall, with the additional checks, we are confident in the QC pipelines implemented in our study and do not believe the data quality is significantly different to other studies using the Olink Explore assay. Surprisingly, we could not find details on data assay warning proportions in previous Olink Explore based studies; it may be the case that these studies observed similarly low rates of assay warnings to those observed in the present study.

A two-step normalisation procedure was used to generate NPX (1: across plates, 2: across batches). Please include information on how much this influences protein CVs?

The within-batch normalization procedure centers data at NPX=0 by subtracting the plate-specific median per assay from all samples and assays in the same plate. Across batches, normalization computes a set of

adjustment factors as the difference of the assay-specific median NPX values of each batch. The first step accounts for plate-to-plate variation within batch and the second step accounts for batch-to-batch variations within the study. Both steps are shifts of an assay-specific fixed factor in NPX scale; namely, plate median in the first step and difference of assay-specific medians between batches in the second step. Normalization steps do not affect intra-plate CVs as the exact same factors are applied in both steps. Inter-plate CVs after the first step of normalization improve within-batch CV, while inter-plate CVs after the two-step normalization improve CV for the full dataset. We have added this to the Supplementary Information text on page 6-7.

The plot below summarises the above and shows the CV for each step of the normalization: step 0 - pc normalized - light blue; step 1 - within batch normalized - red; step 2 - across batches normalized - dark blue. The bottom pane shows intra-plate CV, which remain unaffected from normalization steps. In the upper pane, for each batch blue bars are equal or lower than light blue and red in each batch. This shows that within-batch normalization works when looking at one batch at a time. However, within batch normalized data have higher CV as batch-to-batch variation has not been removed (see red bar for the full dataset - rightmost groups of bars) - when across batches normalization is applied, CV are lower.

Was there any reason for selecting batch 1 as the reference for ‘across batch’ normalization? Is there good concordance if another batch (e.g. largest, best) is selected as reference?

The selection of the reference batch does not play any role as the across-batches normalization uses differences of assay-specific median NPX values between batches. These are fixed factors in NPX scale that aim at harmonizing median NPX values of assays between batches. We have added this to the Supplementary Information (p7). Batch 1 was the first to run chronologically and has undergone extensive investigation on randomization and assay performance; thus, it was selected as the reference batch. The plot below compares CVs from using batch 1 and batch 5 (randomly selected) as the reference set, respectively, and explicitly clarifies that CVs are not altered due to the selected reference set.

Could the authors please perform an analysis of plate effects similar to the one done for investigation of batch effects. It would be helpful to see whether any proteins are particularly affected by plate.

We have included this analysis by creating a term representing plate random effects instead of batch effect, as the reviewer suggested. We have added the results to Supplementary Information (p10-12). All proteins had <30% and 2 proteins (Olink protein ID = OID20796 or OID21305) had >20% of the variability attributable to plate, and all except those labeled (16 in total) had the percentage less 10%. Therefore, we do not believe that there was a global plate effect.

LOD information is helpful but the related Figure S7 is not as informative as it could be due to the y-axis. How many proteins have less than 5% of samples below LOD? Please state this explicitly in the text and redraw the figure either breaking the y-axis or removing proteins with less than 5% below LOD. Please add a table that LOD information for all proteins or at least all with LOD>5%. The authors have focused this part of the % of protein assays with values below LOD per sample. It would be helpful to also know whether any samples have lower detectability than the average.

We thank the reviewer for the constructive feedback. We replaced the old Figure S7 with a revised Figure (Figure S8) in the Supplementary Information (p13), showing the histogram of proportion below LOD by panel, resembling the prior plot but removing proteins that had less than 5% sample below LOD. The specific percentage of samples below LOD for each protein is available in Supplementary Table 3.

Overall, 1894 proteins out of 2941 had less than 5% of samples below LOD. However, this proportion differed by panel - as 90.8%, 82.6%, 77.9%, and 81.5% of proteins in Cardiometabolic, Inflammation, Neurology and Oncology panels had <5% samples with NPX measurements below LOD; the corresponding figures for Cardiovascular_II, Inflammation_II, Neurology_II, and Oncology_II panels were 51.2%, 64.4%, 34.2%, and 32.3%, respectively.

Language checks:

Refer to men/women not males/females?

We have changed “males/females” to “men/women” throughout the text.

Refer to cross-sectional differences as ‘higher or lower levels’ rather than a ‘de-/increase’ which implies a time/ longitudinal aspect?

Where applicable, we have changed this throughout the text.

There are some small errors that need correcting throughout, ie pleotropic etc.

We have revised the manuscript and corrected any errors that we could identify.

Reviewer Reports on the First Revision:

Referees' comments:

Referee #1 (Remarks to the Author):

The authors are applauded for the thoroughness with which they have addressed the critiques. The paper is much improved. It is a true conceptual and analytic tour de force and at the same time is presented in clear, concise, and easy to follow language. I really enjoyed reading it. I have the following minor critiques that mostly require textual clarification or recrafting of language/messaging:

1. The inclusion of individuals not from the White British population strengthens the work. The authors should take care, however, accurately describe the approach to categorizing populations as this has been a significant topic of recent editorials and a book from the US National Academy of Science (<https://nap.nationalacademies.org/catalog/26902/using-population-descriptors-in-genetics-and-genomics-research-a-new>) . The way in which they are referenced should be in accordance with how the population subgroups were determined and the intended use of that determination. For example, they use the term "ethnic background" on lines 134-135 whereas elsewhere they use "ancestry" (line 268, 351). It may be that these are correct and precise as written but this should be verified and addressed. It is recommended they follow the NAS guidelines referenced above.
2. When describing the differences between the randomly chosen, consortium chosen, and COVID-19 cohorts, the text on lines 127-135 could be made clearer by including information on the direction of the smoking prevalence differences.
3. Figure 1A is confusing. It is not clear why there are two columns of panels that differ in that one lists 369 Cardiometabolic and the other 267 Cardiometabolic proteins.
4. The sentence on lines 311-313 is likely missing a few words as it appears jumbled.
5. There is a typo on line 356 with a few duplicated words: "Given the small AMR sample size for AMR".
6. We appreciate the author's use of colocalization to further explore the PCSK9-HDLc Mendelian randomization analysis. The authors find that for HDLc there is strong evidence ($PP1 > \sim 0.9$) to support the presence of an association with only PCSK9. These colocalization results suggest that the PCSK9-HDLc MR findings are an artifact of the precisely-estimated genetic association with circulating PCSK9, without a meaningful signal for HDLc at the PCSK9 locus. We strongly recommend the authors contextualize the effect sizes of both the MR finding (eg. the effect size is miniscule compared to LDLc or total cholesterol) and the results of RCTs (eg. PCSK9i has <10% effect on HDLc, in comparison to ~50% for LDLc). Several secondary mechanisms by which PCSK9 might influence HDLc have been proposed (PMID: 28826253), and based on the pattern of positive MR (with small effect size) with negative colocalization it is important to highlight this possibility.
7. Figure 1D – could the text be moved to a section of the figure that doesn't overlap with the plotted points? This is already difficult to read within the context of a small subpanel.
8. Citation #32 appears "INVALID"

Referee #2 (Remarks to the Author):

Sun et al. have submitted a revised and improved manuscript is now titled "Plasma proteomic

profiling of 54,219 UK Biobank participants reveals novel associations with genetics, health, and disease". The authors have done an excellent job addressing my first round of comments.

Undoubtedly, the presented data set, now that it is also accessible to the community, will impact research using plasma proteomics data in many different ways. The presented "first analysis" of the data could have been done in many different ways, and I applaud the authors for touching upon as many aspects without diluting the overall message. In fact, it might be impossible to extract THE single most important finding from such a unique data set. Bringing in associations to major clinical traits and using proteomics for their predictions has shown the utility and quality of the data. This transparent reporting will make this study a reference for future investigations. Connecting the trans-pQTLs with interaction and regulation has provided validity and reference to our current knowledge.

Based on the authors' response and improvements made to the manuscript, just a few minor questions have remained. My focus will remain on the proteomics side of the study.

1) Diagnosis and sampling - As the authors pointed out, the proteome provides insights into health at the time points of sampling. In some diseases, the effects on the molecular level remain short-lived (e.g., infections), while elevated or decreased levels may also persist for extended periods (e.g., diabetes). Disease severity may also be another non-genetic component influencing how the plasma proteome adapts. The COVID study could be used to investigate the time between sampling after diagnosis to exemplify such effects.

2) Global analysis - The authors added PCAs and related analyses to the revised versions and did not observe effects on PCs and box plots (Fig S9-11). However, for this reviewer, center #19 might be different from the others. Please add how many samples originated from the respective centers and support the visual assessments using basic statistical tests to define the existence or absence of the effects. Also, please describe the degree of processing for the data used for the PCA. Despite their lower robustness, our experience with UMAP provided more refined insights into hidden clusters that were not visual by PCA. This may be due to the non-linear relationship between proteins. Please present such an analysis using NPX data adjusted for age, sex, and BMI but without correcting further for pre-analytical factors and PCs. The scope should be identifying possible sub-clusters that influence the global data structure.

3) Data processing and filtration - The new data set starts with nearly 3000 proteins. Please clarify how many of these were chosen for the final analyses or if different numbers of proteins were used per sample. A flowchart (quick guide) in the supplementary could be helpful. Please confirm that one or several data processing schemes were applied to the NPX data for the different analyses. For example, if I understood correctly, the proteomics data were adjusted by the first 20 PCs - which is unusual for this data type. A concern could be that extensive pre-processing could overcompensate "true" effects and, thus, make it difficult to confirm the findings in smaller validation studies (which may suffer from the influence of factors that could only be accounted for in large-scale studies).

4) Sample age - While I agree that there is no uniform "sample age effect," the increased frequency of inflammatory proteins being labeled as sample age affected could reflect the nature of the short-lived reactive factors. A final analysis and 2-3 sentences about other commonalities between the indicated proteins (red labels in Fig S12) would be informative. If possible, the authors could check if data from this sample-age-susceptible protein group are any less reliable than other sample-age-stable proteins (e.g., frequency of sample values below LOD). The susceptibility to storage time may likely increase the variance of the protein data and influence their association ranking. A closing statement that each protein has its unique stability will be helpful in understanding (some) differences between research on proteins and DNA.

5) Association and prediction - It was very reassuring to see the utility of the proteomics data in

Fig S3. Again, the type and degree of pre-processing of the NPX values were not accessible. Please add details if these differ between the analyses.

6) Effects of sample size - The authors responded to the rebuttal with an interesting quote: "Usually for true positives, significance will increase with larger sample sizes, while false positive will regress to the null." Similar to the pQTL analysis, would it be possible to simulate the trends of the effects or p-values for clinical and pre-analytical variables with increasing sample sizes? This could be done for center, sample age, sex, BMI, and other preference variables. If the statement holds, the influence of the less critical variables (such as sample age, center) will decrease with sample size. In contrast, deterministic variables (such as BMI, age, sex) will continue to increase in importance. It will again be essential to describe the degree of NPX data processing.

7) Resampling - Efforts to resample individuals are vital for future studies to evaluate the consistency of observations made at this scale. Since it is deemed unlikely that any other sample of a UKB participant has been analyzed in other cohort studies, the next round should focus on resampling and longitudinal studies. These will reveal the stability of the marker in terms of analytical performance and consistency of personal health signatures. Please add this to the discussion.

It will be interesting to follow how the community will expand on this study and perform functional investigations to confirm new knowledge.

Referee #3 (Remarks to the Author):

Thank you for the opportunity to review this this revised version.

The authors have attempted to carefully address each of my comments, which is much appreciated.

The revisions have led to substantial changes, most of which will, in my view, benefit the readers and have improved the manuscript. The authors now include results for all 3k proteins, the discussion of platform comparability is now more balanced, information on the influence of a range of parameters on the plasma proteome has been provided, ethnic specific analyses and comparisons have been included, and the new version has now more details on QC, normalisation, and data quality in the supplement.

As a result, the length seems to have increased substantially with a lot of technical information in the main text. In my view there is substantial scope to streamline and shorten several sections (e.g. observational analyses, sex-age etc interactions) in addition to moving some of these parts to the supplement. It takes a long time to get to the 'meat'... I propose to do this to make room for the exciting elements of this paper, i.e. focus on the genetic analyses and insights into disease, biology and drug targets, the main focus of this work. This is the part that will be of most interest to Nature readers and is currently extremely short in the main text (11150-62). More of this should be promoted from the supplement. Otherwise, this work is at danger of reading like a somewhat lengthy counting and listing exercise, which doesn't do the enormous amount of work that went into it justice.

Some of the included examples are very interesting and well described. For BAFF, it would be helpful to replace the generic 'larger studies will uncover...' with a comment on any potential implications for SLE and BAFF inhibition.

It is extremely useful that the consortium has now included analyses of the expanded set of proteins; this is provided that results will be accessible for all 2,922 proteins at publication through

the newly developed webserver.

One important element I take issue with is the extremely lax (nominal) threshold and strategy used to define replication. The cost of reporting false positives and many non-specific trans associations is too high and not justified. The replication rate is non-conservative. I propose to change the order to what is a more accepted standard for good reason a) discovery, b) replication (directionally concordant and genome-wide significant), then c) detailed follow-up analyses for replicated pQTLs. I see little reason to include full-cohort pQTLs – the marginal value (i.e. 26 more cis pQTLs without replication) is too small to justify this section. If the authors insist on keeping it, it should at least go to the supplement.

Previous work should be referenced much earlier on in the introduction, specifically studies of the same proteins using the same technology and included to after the replication stage to define novelty for cis and trans – this appears to not have been done systematically. Table S2 is incomplete, this needs to be updated to not overestimate novelty.

The main text (rather than supplement) would benefit from some more specific details on QC parameters (brief) by panel and clarification whether results refer to those before and after exclusions.

Table S1 should be more clearly summarised in the main text and the denominator for each column clarified in the legend. The plate effect appears substantial for quite a number of proteins, too much of some of important QC messages are buried in the supplement, including % of samples with measures below LOD or influence of sample age.

Small points:

Description of generalisability (of subsets) should include differences in means/ proportions, not simply p-values. At this size, meaningless, very small differences may be significant without any bearing on the results. Showing disease enrichment of disease-specific samples is circular, I don't see the point. People were not randomised but randomly selected.

Many sentences lack references and for some statements of earlier work appear selected.

L152 – studies not databases. Pls clarify what you mean by 'proprietary', this seems to not quite be the wrong term.

Figure S5. Odd choice of scale for the y-axis. Please label highest proteins to make it useful. Same is true (y-axis scale and no labelling) for a number of figures in the supplement, this could easily be changed to make these more useful/ readable.

Author Rebuttals to First Revision:

Referees' comments:

Referee #1 (Remarks to the Author):

The authors are applauded for the thoroughness with which they have addressed the critiques. The paper is much improved. It is a true conceptual and analytic tour de force and at the same time is presented in clear, concise, and easy to follow language. I really enjoyed reading it. I have the following minor critiques that mostly require textual clarification or recrafting of language/messaging:

We thank the reviewer for their generous feedback on the manuscript.

1. The inclusion of individuals not from the White British population strengthens the work. The authors should take care, however, accurately describe the approach to categorizing populations as this has been a significant topic of recent editorials and a book from the US National Academy of Science (<https://nap.nationalacademies.org/catalog/26902/using-population-descriptors-in-genetics-and-genomics-research-a-new>). The way in which they are referenced should be in accordance with how the population subgroups were determined and the intended use of that determination. For example, they use the term “ethnic background” on lines 134-135 whereas elsewhere they use “ancestry” (line 268, 351). It may be that these are correct and precise as written but this should be verified and addressed. It is recommended they follow the NAS guidelines referenced above.

The reviewer raises a crucial point. We thank them for providing a link to the National Academies book. We have reviewed the main text and can verify that all uses of “ethnic background” specifically refer to terminology used in UK Biobank’s touchscreen questionnaire, outlined in full here: <https://biobank.ctsu.ox.ac.uk/crystal/field.cgi?id=21000>. Importantly, this question was dropped from the touchscreen protocol on the 24th of October, 2016, possibly as a result of the important emerging discussions on such terminology referred to by the reviewer, above. We have retained instances of “ethnic background” in the main text only where they apply to the touchscreen questionnaire and have included a note regarding this terminology on page 3 of the Supplementary Information.

All other references to population subgroups in both the main text and supplementary information use the term “ancestry”, referring to genetic ancestry, which we believe is most consistent with guidelines from the National Academies guidelines and with recent editorials (e.g., <https://www.nature.com/articles/s41588-021-00952-6>). We clarify that “ancestry” refers to genetically determined ancestry in the Methods section when describing the Pan UKBB definitions.

2. When describing the differences between the randomly chosen, consortium chosen, and COVID-19 cohorts, the text on lines 127-135 could be made clearer by including information on the direction of the smoking prevalence differences.

We thank the reviewer for highlighting how this was unclear and have amended the text to include the direction of smoking prevalence differences.

3. Figure 1A is confusing. It is not clear why there are two columns of panels that differ in that one lists 369 Cardiometabolic and the other 267 Cardiometabolic proteins.

We apologise for the confusion. The Olink Explore 3072 platform contains two sets of each panel (i.e., “Cardiometabolic I” and “Cardiometabolic II”, etc.). Text referring to version II was truncated in our previous submission. This has been fixed in the revised submission.

4. The sentence on lines 311-313 is likely missing a few words as it appears jumbled.

We have rephrased the sentence as follows: “Of the trans pQTLs located on the same chromosome as the gene encoding the protein, all but two were located >2Mb away from their corresponding genes (93% were >5Mb, 81% were >10Mb away).”

5. There is a typo on line 356 with a few duplicated words: “Given the small AMR sample size for AMR”.

We have corrected this typo.

6. We appreciate the author’s use of colocalization to further explore the PCSK9-HDLc Mendelian randomization analysis. The authors find that for HDLc there is strong evidence ($PP1 > \sim 0.9$) to support the presence of an association with only PCSK9. These colocalization results suggest that the PCSK9-HDLc MR findings are an artifact of the precisely-estimated genetic association with circulating PCSK9, without a meaningful signal for HDLc at the PCSK9 locus. We strongly recommend the authors contextualize the effect sizes of both the MR finding (eg. the effect size is miniscule compared to LDLc or total cholesterol) and the results of RCTs (eg. PCSK9i has <10% effect on HDLc, in comparison to ~50% for LDLc). Several secondary mechanisms by which PCSK9 might influence HDLc have been proposed (PMID: 28826253), and based on the pattern of positive MR (with small effect size) with negative colocalization it is important to highlight this possibility.

We thank the reviewer for this suggestion and have added the effect size contextualization with published clinical trials. We have also described the potential secondary mechanism, citing the suggested reference (p44 in Supplementary Information).

7. Figure 1D – could the text be moved to a section of the figure that doesn’t overlap with the plotted points? This is already difficult to read within the context of a small subpanel.

Figure 1d has been modified now with the regression summary text now above the subpanels.

8. Citation #32 appears “INVALID”

This has been corrected.

Referee #2 (Remarks to the Author):

Sun et al. have submitted a revised and improved manuscript is now titled “Plasma proteomic profiling of 54,219 UK Biobank participants reveals novel associations with genetics, health, and disease”. The authors have done an excellent job addressing my first round of comments.

Undoubtedly, the presented data set, now that it is also accessible to the community, will impact research using plasma proteomics data in many different ways. The presented “first analysis” of the data could have been done in many different ways, and I applaud the authors for touching upon as many aspects without diluting the overall message. In fact, it might be impossible to extract THE single most important finding from such a unique data set. Bringing in associations to major clinical traits and using proteomics for their predictions has shown the utility and quality of the data. This transparent reporting will make this

study a reference for future investigations. Connecting the trans-pQTLs with interaction and regulation has provided validity and reference to our current knowledge.

Based on the authors' response and improvements made to the manuscript, just a few minor questions have remained. My focus will remain on the proteomics side of the study.

1) Diagnosis and sampling - As the authors pointed out, the proteome provides insights into health at the time points of sampling. In some diseases, the effects on the molecular level remain short-lived (e.g., infections), while elevated or decreased levels may also persist for extended periods (e.g., diabetes). Disease severity may also be another non-genetic component influencing how the plasma proteome adapts. The COVID study could be used to investigate the time between sampling after diagnosis to exemplify such effects.

We appreciate the reviewer's comments regarding the effects of disease severity and the time point of sampling upon plasma protein levels. Given the heterogeneity of UKB and the vast number of diseases potentially influencing protein concentrations, we believe that this topic is best addressed as an independent, dedicated manuscript to retain the focus and length of this manuscript. Other reviewers have indicated that the manuscript needs to be streamlined; thus, we have discussed with the editor and concluded that it would be beyond the current scope to include complex new analyses in an already overlong manuscript. Although the reviewer makes an excellent suggestion to explore temporal and disease severity-related effects on proteins using data from the COVID-19 sub-study, that study is only limited to two time points in a healthier and younger population (as seen in ST1) capable of attending repeat imaging visits; thus, it could lead to biased conclusions best addressed as part of a standalone manuscript.

We also appreciate the reviewer's comments regarding longitudinal effects of proteins in disease mortality and risk, which is being extensively studied as part of several independent studies using the UKB-PPP data, e.g.:

1. <https://www.medrxiv.org/content/10.1101/2023.05.01.23288879v1> - "Blood protein levels predict leading incident diseases and mortality in UK Biobank".
2. <https://assets.researchsquare.com/files/rs-2626017/v1/8257e2d4133fc456e79f95eb.pdf?c=1678950869> - "Plasma Proteomic Determinants of Common Causes of Mortality"
3. <https://www.medrxiv.org/content/10.1101/2023.02.27.23286529v1> - "Plasma neurofilament light levels show elevation prior to diagnosis of sporadic motor neuron disease in the UK Biobank cohort"

We believe that the effects of longitudinal sampling and disease severity are best investigated by these dedicated independent studies and have added language to our summaries of disease associations (Supplementary Information) acknowledging the potential impact(s) of these variables.

2) Global analysis - The authors added PCAs and related analyses to the revised versions and did not observe effects on PCs and box plots (Fig S9-11). However, for this reviewer, center #19 might be different from the others. Please add how many samples originated from the respective centers and support the visual assessments using basic statistical tests to define the existence or absence of the effects. Also, please describe the degree of processing for the data used for the PCA. Despite their lower robustness, our experience with UMAP provided more refined insights into hidden clusters that were not visual by PCA. This may be due to the non-linear relationship between proteins. Please present such an analysis using NPX data adjusted for age, sex, and BMI but without correcting further for pre-analytical

factors and PCs. The scope should be identifying possible sub-clusters that influence the global data structure.

We have added the sample numbers for each center in the revised Figure S11. The data used for PCA were the NPX data excluding: (1) control samples, (2) those who withdrew from the study/data not processed, (3) data points missing NPX values, (4) those missing in covariates such as sex and sampling center. This has been added to the Supplementary Information (p8).

To detect whether there is a global center effect, tests such as t-test or Mood's median test may not be ideal, because (1) magnitude of differences in a PC are not readily interpretable, and (2) the results of these tests, in particular p-values, are a function of data size; in other words, the larger the data, the more likely $p < 0.05$. With over 50,000 samples, mostly likely the null hypothesis (i.e., all means/medians are equal) will be rejected, even though there may not be that much different in terms of effect – hence we took a comparable approach to the one used to assess batch effects, i.e., fit a simple random-effects model to examine the proportion of variation attributable to sampling center (added to SI p16):

$PC_i = b_0 + u_j + e_{jk}$, where

- $i = 1$ or 2 , indicating standardized principal components 1 (PC1) or 2 (PC2)
- b_0 = global mean
- u_j = sampling center random effects
- e_{jk} = error terms
- Percentage of variability attributable to sampling center = $\frac{VAR(u_j)}{VAR(u_j + e_{jk})}$

For PC1, the proportion of variability attributable to sampling center was 3.5%, and the corresponding figure for PC2 was 1.0%. Because the proportion of variability attributable to sampling center was low, we do not believe there was strong evidence indicating the existence of a sampling center effect.

We understand that in Figure S11 the distribution of standardized PC1 of center #19 looked somewhat higher than others. However, as the mixed-effects model showed, the variability attributable to sampling center was low (3.5%), and the much smaller sample size of centre #19 may have contributed to its boxplot looking slightly different to the others. Thus, we do not believe that there is strong evidence indicating the existence of sampling center effect. Nonetheless, we have adjusted for UKB centres in our analyses, given that protein-by-protein the effects may vary.

We have also added the UMAP analyses, now alongside the PCA plot (now revised Figures S9 and S10). We did not observe obvious subclusters within the UMAP plot. As the reviewer highlighted, UMAP is good for broad visualization to find non-linear topological clusters but is not as stable as deterministic PCA approaches.

3) Data processing and filtration - The new data set starts with nearly 3000 proteins. Please clarify how many of these were chosen for the final analyses or if different numbers of proteins were used per sample. A flowchart (quick guide) in the supplementary could be helpful. Please confirm that one or several data processing schemes were applied to the NPX data for the different analyses. For example, if I understood correctly, the proteomics data were adjusted by the first 20 PCs - which is unusual for this data type. A concern could be that extensive pre-processing could overcompensate “true” effects and, thus, make it

difficult to confirm the findings in smaller validation studies (which may suffer from the influence of factors that could only be accounted for in large-scale studies).

Analyses described in this manuscript included all proteins that passed rigorous quality control. We have reported the key CV and QC performance metrics in Supplementary Table 3 for transparency, facilitating user defined filtering. After sample filtering, data pre-processing and QC, we did not perform further NPX processing; we now clarify this on page 8 in the SI and Methods (p25 main text). We employed uniform processing on NPX measures by rank normalizing each protein separately prior to analyses – this has been clarified in the Methods (p25, “Proteomic measurement, processing and quality control”). The sample level and genetic QC can lead to different number of samples analysed for each protein-variant combination; this information is included in the full summary statistics upon download.

We excluded only one protein (GLIPR1) due to an outlying 99.4% of data failing QC – also mentioned in the Supplementary Information. These details have been added to Methods (p25). We did not adjust proteomic data by proteomic PCs, for the reasons the reviewer outlined above. We did, however, adjust for 20 *genetic* PCs to minimize coarse population stratification. We have checked the text in the Methods (p28) to confirm that this refers to correction for genetic PCs.

4) Sample age - While I agree that there is no uniform “sample age effect,” the increased frequency of inflammatory proteins being labeled as sample age affected could reflect the nature of the short-lived reactive factors. A final analysis and 2-3 sentences about other commonalities between the indicated proteins (red labels in Fig S12) would be informative. If possible, the authors could check if data from this sample-age-susceptible protein group are any less reliable than other sample-age-stable proteins (e.g., frequency of sample values below LOD). The susceptibility to storage time may likely increase the variance of the protein data and influence their association ranking. A closing statement that each protein has its unique stability will be helpful in understanding (some) differences between research on proteins and DNA.

We have now included an analysis investigating the indicated proteins for enrichment in GO process and molecular function pathways with text describing the results in Supplementary Information p18. We added (with two review references) the relevance of the enriched MAPK and ubiquitin pathways in cellular stress. We have also implemented an analysis investigating sample-age variability explained against proportion of samples below LOD and failing QC – and did not find that the sample-age susceptible protein group was less reliable. We have also added a closing statement as the reviewer recommended to the last part of this section in Supplementary Information p18.

5) Association and prediction - It was very reassuring to see the utility of the proteomics data in Fig S3. Again, the type and degree of pre-processing of the NPX values were not accessible. Please add details if these differ between the analyses.

In response to comment (3), we have clarified details of pre-processing of the NPX values in the Methods and SI.

6) Effects of sample size - The authors responded to the rebuttal with an interesting quote: “Usually for true positives, significance will increase with larger sample sizes, while false positive will regress to the null.” Similar to the pQTL analysis, would it be possible to simulate the trends of the effects or p-values for clinical and pre-analytical variables with increasing sample sizes? This could be done for center, sample age, sex, BMI, and other preference variables. If the statement holds, the influence of the less critical variables (such as sample age, center) will decrease with sample size. In contrast, deterministic

variables (such as BMI, age, sex) will continue to increase in importance. It will again be essential to describe the degree of NPX data processing.

We appreciate the reviewer's comments, which relate to the previous comment on how large sample sizes may inflate initial observations. We apologize for the slight confusion that our response raised. We agree with the reviewer's view on this limitation. We meant to clarify that false positives will regress to the null in independent study cohorts, which would still be the gold standard. This is a different phenomenon than what we assessed via the relationship of sample size and pQTL discovery rates (Figure 2e-g).

The reviewer raised another important point about the impact of "less critical" factors (e.g., sample age, collection center). These factors can be sources of technical variability that do NOT regress to NULL with increased samples sizes.

We have added the reviewer's comment that increasing sample sizes potentially may inflate some of the initial observations in the Discussion (p23) since this applies to other large-scale population studies and raise this limitation/caution more widely:

As sample sizes continue to increase, modest proteomic associations may become inflated; thus, independent replication in diverse cohorts remains critical for robust, translational results.

7) Resampling - Efforts to resample individuals are vital for future studies to evaluate the consistency of observations made at this scale. Since it is deemed unlikely that any other sample of a UKB participant has been analyzed in other cohort studies, the next round should focus on resampling and longitudinal studies. These will reveal the stability of the marker in terms of analytical performance and consistency of personal health signatures. Please add this to the discussion.

We thank the reviewer for this suggestion and have added the below text to the discussion (p23):

Future studies should also prioritize resampling initiatives and longitudinal analyses, facilitating more systematic evaluations of assay analytical performance(s), consistencies of personal health signatures, and the effects of disease incidence, prevalence and severity on biomarker stability and pQTL detection.

It will be interesting to follow how the community will expand on this study and perform functional investigations to confirm new knowledge.

Referee #3 (Remarks to the Author):

Thank you for the opportunity to review this this revised version.

The authors have attempted to carefully address each of my comments, which is much appreciated.

The revisions have led to substantial changes, most of which will, in my view, benefit the readers and have improved the manuscript. The authors now include results for all 3k proteins, the discussion of platform comparability is now more balanced, information on the influence of a range of parameters on the plasma proteome has been provided, ethnic specific analyses and comparisons have been included, and the new version has now more details on QC, normalisation, and data quality in the supplement.

As a result, the length seems to have increased substantially with a lot of technical information in the main text. In my view there is substantial scope to streamline and shorten several sections (e.g. observational analyses, sex-age etc interactions) in addition to moving some of these parts to the supplement. It takes a long time to get to the ‘meat’... I propose to do this to make room for the exciting elements of this paper, i.e. focus on the genetic analyses and insights into disease, biology and drug targets, the main focus of this work. This is the part that will be of most interest to Nature readers and is currently extremely short in the main text (11150-62). More of this should be promoted from the supplement. Otherwise, this work is at danger of reading like a somewhat lengthy counting and listing exercise, which doesn’t do the enormous amount of work that went into it justice.

We appreciate the reviewer’s comments regarding the length of the manuscript. In the first manuscript revision, we aimed to fulfil previous requested from reviewers to shorten the main text by moving several results to the Supplementary Information, focusing on the key findings and important technical details, while incorporating several requested new analyses. Given the extent of these new analyses, the manuscript’s word count was considerably over the 4,500-word limit recommended for an eight-page article.

In the new revision, we have further streamlined the main text – delegating observational analyses and some technical sections to the Supplementary Information as the reviewer recommended. Since this will be the first manuscript introducing the UKB-PPP study, we have retained key background information on the cohort and validations as the reviewers previously recommended.

We note that the while these changes have heavily streamlined the main text, it still slightly exceeds the recommended word count. Thus, we have refrained from promoting additional materials from the Supplement. While we are happy to include additional insights into disease biology and drug targets in the main text, the final composition of the main text may best be determined via editorial guidance, incorporating feedback from the three expert reviewers.

Some of the included examples are very interesting and well described. For BAFF, it would be helpful to replace the generic ‘larger studies will uncover...’ with a comment on any potential implications for SLE and BAFF inhibition.

We thank the reviewer for their comment.

This vignette is a proof-of-concept that demonstrates the potential impact of trans pQTL pairs to identify and characterize biological networks.

BAFF inhibition is an established treatment option for SLE patients*. Therefore, we included these findings as an example of how they can be relevant to health traits, disease biology, and drug targets. We have adjusted this text to clarify this intent.

These results are not expected to have any impact on the existing BAFF/SLE therapies.

*In 2011, belimumab received the first approval for a biologic treatment of LN/SLE in more than 50 years. It has received several additional approvals, including a recent one for pediatric patients (<https://us.gsk.com/en-us/media/press-releases/gsk-announces-us-fda-approval-of-benlysta-belimumab-for-pediatric-patients-with-active-lupus-nephritis/>)

It is extremely useful that the consortium has now included analyses of the expanded set of proteins; this is provided that results will be accessible for all 2,922 proteins at publication through the newly developed webserver.

Results for all 2,922 proteins can be fully perused via the newly developed webserver. The consortium has released full results for the first 1,463 proteins to the public, accessible at <https://www.synapse.org/#!/Synapse:syn51364943/wiki/622119>, and is committed to releasing results for the remaining proteins upon publication.

One important element I take issue with is the extremely lax (nominal) threshold and strategy used to define replication. The cost of reporting false positives and many non-specific trans associations is too high and not justified. The replication rate is non-conservative. I propose to change the order to what is a more accepted standard for good reason a) discovery, b) replication (directionally concordant and genome-wide significant), then c) detailed follow-up analyses for replicated pQTLs. I see little reason to include full-cohort pQTLs – the marginal value (i.e. 26 more cis pQTLs without replication) is too small to justify this section. If the authors insist on keeping it, it should at least go to the supplement.

We appreciate the author's comments on replication thresholds. We previously included both a more stringent replication threshold and a more lenient threshold for completeness. The more stringent threshold of $p < 1.3e-5$ accounts for the number of unique associated genetic regions, whereas the more lenient nominal significance threshold of $p < 0.05$ is in line with reporting of such a threshold in several previous pQTL studies (Pietzner 2021, Katz 2022, Yang 2021, Gujonsson 2022). Nonetheless, the reviewer's point regarding potential reporting of false positives at $p < 0.05$ is well-taken. Thus, we have removed the nominal significance threshold in the text to avoid confusion and have kept the more stringent threshold.

We believe that the more stringent threshold of $p < 1.3e-5$ is the most appropriate for replication, as it adjusts for the number of associated regions in the discovery analysis while accounting for linkage disequilibrium between closely positioned sentinel variants and correlations between proteins at a locus. We have conducted empirical permutation analyses (permuting sample-wise to retain the genetic and proteomic correlation structure, $n=100$ times) to check whether false positives are maintained at a low level. At our threshold of $p < 1.3e-5$, we expect on average 19 of the 9,775 (0.19%) pQTLs deemed replicated to be false positives, and at most 30 (0.31%) false positives across 100 permutation runs even without stipulating consistent effect directions. Therefore, our choice for replication threshold adequately controls for the number of false positives in replications. As a comparison, false positive rates at $p < 0.05$ are (1179/9775) 12.1% and (202/9775) 2.1% disregarding and accounting for consistent directions respectively.

We have removed the full-cohort pQTLs section from the text, retaining a single, brief reference to the analysis for completeness (Main text, p10).

Previous work should be referenced much earlier on in the introduction, specifically studies of the same proteins using the same technology and included to after the replication stage to define novelty for cis and trans – this appears to not have been done systematically. Table S2 is incomplete, this needs to be updated to not overestimate novelty.

We thank the reviewer for these comments. Given that the Olink Explore 3072 technology is a relatively new platform, the only published study that has conducted genome-wide genetic analysis of all 2,923 proteins measured via the assay is from Koprulu et al. (*Nature Metabolism*, 2023). This is a very recent study, published three months ago, shortly before our resubmission. In the revised introduction, we now

include a citation to Koprulu et al alongside a selection of other key proteogenomics studies examples; a more extensive review is available in Table S2 in the Supplementary Information.

We believe that our cross-referencing with prior protein genome-wide association studies represents a more systematic and extensive effort than any others published to date; however, we acknowledge that the comprehensiveness of this approach may have been unclear in the prior revision. Thus, we have added a clarification below and have expanded the supplementary information to provide more details of our methods ('List of prior pQTL studies').

We used two primary resources to ascertain previously published pQTL studies: the first is the GWAS catalog (<https://www.ebi.ac.uk/gwas/>) and the second is a list of published GWAS with proteomics compiled by Prof. Karsten Suhre at metabolomix.com (<http://www.metabolomix.com/a-table-of-all-published-gwas-with-proteomics/>). From these two sources, we originally included any publications available on or before March 1, 2023. For this revision, we also have included the very recent study by Koprulu et al mentioned above (<https://www.nature.com/articles/s42255-023-00753-7>).

Metabolomix.com contains an extensive list of studies with genetics and proteomics/proteomics adjacent data, and we manually curated these for suitability of inclusion. In order to be included, a prior publication needed to meet all of the following criteria:

- (1) study was a peer-reviewed publication; we excluded studies that were pre-prints and abstracts.
- (2) study was conducted genome wide. Sources of genetic data included were genome-wide array (generally with imputation), exome sequencing, or whole genome sequencing.
- (3) Results reported in publication were genome-wide. Studies were excluded if publication only tested and/or reported SNPs in candidate genes and/or only tested and/or reported SNPs in cis-genes.
- (4) Appropriate p-value correction (GW threshold) was included and reported. Studies were excluded if FDR correction was the only p-value used/reported.
- (5) Phenotypes tested had to be truly protein based: studies from flow cytometry or other cell-imaging modalities were excluded.
- (6) Studies were required to report all proteins in a panel to be included. A study that generated proteomics data on a platform but only reported selected proteins were not included.

The majority of pQTL studies have been conducted in plasma or serum; some studies also include proteins measured in CSF or brain which were also considered in our cross-referencing effort. We believe this is the most comprehensive comparison of previous human pQTL studies to date, in view of comparisons done in previously published pQTL studies.

The main text (rather than supplement) would benefit from some more specific details on QC parameters (brief) by panel and clarification whether results refer to those before and after exclusions.

Given the length of the main text and excess of technical details that the reviewer alludes to in the first comment (with which we fully agree), we believe that QC parameters by panel are best kept self-contained in the detailed Supplementary Information to keep focus on the main results. We have now

clarified that all results are reported after exclusions at the end of the first paragraph in “Data processing, quality control and orthogonal assay comparisons” (Main text, p6).

Table S1 should be more clearly summarised in the main text and the denominator for each column clarified in the legend. The plate effect appears substantial for quite a number of proteins, too much of some of important QC messages are buried in the supplement, including % of samples with measures below LOD or influence of sample age.

We have added the denominator n for each column in the legend for ST1. Considering the additional factors analyzed in the previous revision, alongside the manuscript length constraints discussed above, there were too many denominators to describe in the main text (Main text p5). The key message of this paragraph is that the randomly selected cohort is representative of the full UKB cohort, and that the other sub-cohorts differed across various factors described in ST1. Readers can assess these factors in more detail in the supplementary materials and make necessary adjustments when working with those subcohorts. We have added brief details to the main text (p7) on the proportion of proteins with >10% variability attributable to batch and plate and refer to the detailed supplementary information in the main text. We have also added brief details on % of samples with measures below LOD. Per our comments to Reviewer 2, we have now conducted additional analyses investigating sample-age variability explained against proportion of samples below LOD and failing QC, and did not find that the sample-age susceptible protein group was less reliable. We have added these additional details to Pages 20-24 of the Supplementary Information.

Small points:

Description of generalisability (of subsets) should include differences in means/ proportions, not simply p -values. At this size, meaningless, very small differences may be significant without any bearing on the results. Showing disease enrichment of disease-specific samples is circular, I don't see the point. People were not randomised but randomly selected.

We appreciate the reviewer's comment and have now included the differences in means/proportions in the comparisons of cohort subsets alongside the p -values, where appropriate. We have also contextualized our Mendelian randomization analysis of PCSK9 with results from real-world clinical trials, in accordance with comments from Referee 1. Furthermore, we have updated our descriptions of proteomic associations with disease subsets (now relegated to the supplementary information) to include beta values alongside p -values.

All instances of “randomised” have been updated to, “randomly selected” in the text.

We have illustrated enrichments of certain diseases within the pre-selected samples since we did not know, *a-priori*, which diseases would be enriched (samples were not necessarily selected specifically for disease status). Thus, not all diseases were enriched uniformly – some, such as cataracts, were not enriched. Providing this information gives readers and prospective users of the underlying dataset full context.

Many sentences lack references and for some statements of earlier work appear selected.

We have comprehensively reviewed the manuscript for sentences lacking references or statements without comprehensive citations to prior work and have amended all instances accordingly. In terms of previous pQTL studies, the main text cites several examples, with the complete list provided in Table S2.

L152 – studies not databases. Pls clarify what you mean by ‘proprietary’, this seems to not quite be the wrong term.

We have revised this sentence in line with the reviewer’s comment, replacing “databases” with “studies” and “proprietary” with “...lacking subject-level access or linkage to deep phenotyping”.

Figure S5. Odd choice of scale for the y-axis. Please label highest proteins to make it useful. Same is true (y-axis scale and no labelling) for a number of figures in the supplement, this could easily be changed to make these more useful/ readable.

We thank the reviewer for this suggestion. We have revised Figure S5, S7 and S12 to improve the y-axis scaling, labelling the proteins, now ordered by descending percentage of attributed variability rather than ordering by protein ID. We also divide by protein panels to make the figures more informative.

Reviewer Reports on the Second Revision:

Referees' comments:

Referee #2 (Remarks to the Author):

The authors have further improved the manuscript and streamlined its flow (as agreed with the editors). All my concerns have been resolved, and I don't have any more comments that need to be addressed. Minor details, such as the duplication of references 23 and 61, will for sure be resolved. This work presents an important contribution to the field of proteomics.

Referee #3 (Remarks to the Author):

Thank you for this revised submission. The authors have done a stellar job at responding to and addressing our comments and I look forward to seeing this version out soon.

Thank you for this great work and amazing data!

A noticed few very small points that may merit checking/ correction:

- Line 318/319: 'multivariable' instead of 'multiple' linear regression models (and elsewhere) I assume?
- Might be worth adding a comment comment on the potential sample overlap between UKB instruments for PCSK9 and CVD outcomes in the MR analysis section?
- Figure 1c: please add odds ratios of enrichment not just significance.
- In the analysis for Supp F9, if the effect of sex has been regressed out before PCA, it should not be used to colour, which would be meaningless, the values should have been used in PCA and UMAP without adjusting for anything.
- The number of PAVs among fine-mapped variants is substantial.
- I couldn't find legends for suppl figures, and some appear to be low resolution/ blurry.

Author Rebuttals to Second Revision:

Responses to outstanding referee's comments:

Referee #2 (Remarks to the Author):

The authors have further improved the manuscript and streamlined its flow (as agreed with the editors). All my concerns have been resolved, and I don't have any more comments that need to be addressed. Minor details, such as the duplication of references 23 and 61, will for sure be resolved. This work presents an important contribution to the field of proteomics.

We thank the reviewer for the comments and suggestions. The citation duplication mentioned has been fixed.

Referee #3 (Remarks to the Author):

Thank you for this revised submission. The authors have done a stellar job at responding to and addressing our comments and I look forward to seeing this version out soon.

Thank you for this great work and amazing data!

We thank the reviewer for the comments and suggestions, and for their positive feedback.

A noticed few very small points that may merit checking/ correction:

- Line 318/319: 'multivariable' instead of 'multiple' linear regression models (and elsewhere) I assume?

We have resolved this.

- Might be worth adding a comment on the potential sample overlap between UKB instruments for PCSK9 and CVD outcomes in the MR analysis section?

We have added this to the section in the Supplementary Information (p45).

- Figure 1c: please add odds ratios of enrichment not just significance.

We have added this as the size of the points.

- In the analysis for Supp F9, if the effect of sex has been regressed out before PCA, it should not be used to colour, which would be meaningless, the values should have been used in PCA and UMAP without adjusting for anything.

We have updated the SF9 plots without adjustments.

- The number of PAVs among fine-mapped variants is substantial.

We have noted this.

- I couldn't find legends for suppl figures, and some appear to be low resolution/ blurry.

We have updated the SF figures and legends as needed.